# CEDAR-GPP: spatiotemporally upscaled estimates of gross primary productivity incorporating $CO_2$ fertilization

Yanghui Kang[1,2,3], Maoya Bassiouni[1,2], Max Gaber[1,4], Xinchen Lu[1,2], Trevor F. Keenan[1,2]

[1] Department of Environmental Science, Policy, and Management, University of California, Berkeley, Berkely, CA 94720, USA

[2] Climate and Ecosystem Sciences Division, Lawrence Berkeley National Laboratory, Berkeley, CA 94720, USA

[3] Department of Biological Systems Engineering, Virginia Tech, Blacksburg, VA 24061, USA

[4] Department of Geosciences and Natural Resource Management, University of Copenhagen, Copenhagen, 1350, Denmark.

Correspondence: Yanghui Kang (yanghuikang@vt.edu)
Trevor Keenan (trevorkeenan@berkeley.edu)

**Abstract**: Gross primary productivity (GPP) is the largest carbon flux in the Earth system, playing a crucial role in removing atmospheric carbon dioxide and providing carbohydrates needed for ecosystem metabolism. Despite the importance of GPP, however, existing estimates present significant uncertainties and discrepancies. A key issue is the underrepresentation of the $CO_2$ fertilization effect, a major factor contributing to the increased terrestrial carbon sink over recent decades. This omission could potentially bias our understanding of ecosystem responses to climate change.

Here, we introduce CEDAR-GPP, the first global machine-learning-upscaled GPP product that incorporates the direct $CO_2$ fertilization effect on photosynthesis. Our product is comprised of monthly GPP estimates and their uncertainty at 0.05° resolution from 1982 to 2020, generated using a comprehensive set of eddy covariance measurements, multi-source satellite observations, climate variables, and machine learning models. Importantly, we used both theoretical and data-driven approaches to incorporate the direct $CO_2$ effects. Our machine learning models effectively predict monthly GPP ($R^2 \sim 0.72$), the mean seasonal cycles ($R^2 \sim 0.77$), and spatial variabilities ($R^2 \sim 0.63$) based on cross-validation at flux sites. After incorporating the direct $CO_2$ effects, predicted long-term

GPP trend across global flux towers substantially increases from 3.1 gC $m^{-2}$ year$^{-1}$ to 4.5 – 5.4 gC $m^{-2}$
year$^{-1}$, which aligns more closely with the 7.7 gC $m^{-2}$ year$^{-1}$ trend detected from eddy covariance data.
While the global patterns of annual mean GPP, seasonality, and interannual variability generally align
with existing satellite-based products, CEDAR-GPP demonstrates higher long-term trends globally
after incorporating $CO_2$ fertilization and reflected a strong temperature control on direct $CO_2$ effects.
The estimated global GPP trend is 0.57 – 0.76 PgC year$^{-1}$ from 2001 to 2018 and 0.32 – 0.34 PgC year$^{-1}$
from 1982 to 2018. Estimating and validating GPP trends in data-scarce regions, such as the tropics,
remains challenging, underscoring the importance of ongoing ground-based monitoring and
advancements in modeling techniques. CEDAR-GPP offers a comprehensive representation of GPP
temporal and spatial dynamics, providing valuable insights into ecosystem-climate interactions. The
CEDAR-GPP product is available at https://zenodo.org/doi/10.5281/zenodo.8212706 (Kang et al.,

2024).

# 1. Introduction

Terrestrial ecosystem photosynthesis, known as Gross Primary Productivity (GPP), is the primary source of food and energy for the Earth system and human society (Keenan and Williams, 2018). Through photosynthesis, terrestrial ecosystems also mitigate climate change, by removing thirty percent of anthropogenic carbon emissions from the atmosphere each year (Friedlingstein et al., 2023). However, due to the lack of direct measurements at the global scale, our understanding of photosynthesis and its spatiotemporal dynamics is limited, leading to considerable disagreements among various GPP estimates (Anav et al., 2015; O'Sullivan et al., 2020; Smith et al., 2016; Yang et al., 2022). Addressing these uncertainties is crucial for improving the predictability of ecosystem dynamics under climate change (Friedlingstein et al., 2014).

Over the past three decades, global networks of eddy covariance flux towers collected *in situ* carbon flux measurements that allow for accurate estimates of GPP, providing valuable insights into photosynthesis dynamics under various environmental conditions (Baldocchi, 2020; Beer et al., 2010). To quantify and understand GPP at scales and locations beyond the ~ 1km$^2$ flux tower footprints, machine learning has been employed with gridded satellite and climate datasets to upscale site-based measurements and produce wall-to-wall GPP maps (Dannenberg et al., 2023; Joiner and Yoshida, 2020; Jung et al., 2011; Tramontana et al., 2016; Xiao et al., 2008; Yang et al., 2007; Zeng et al., 2020). This "upscaling" approach provides data-driven and observation-based quantifications without prescribed functional relations between GPP and its climatic or environmental drivers. It offers unique empirical constraints of ecosystem carbon dynamics, complementing those derived from process-based and semi-process-based approaches such as terrestrial biosphere models or the Light Use Efficiency (LUE) models (Beer et al., 2010; Gampe et al., 2021; Jung et al., 2017; Schwalm et al., 2017). In recent years, the growth of global and regional flux networks, coupled with increasing efforts in data standardization, has offered new opportunities for the advancement of upscaling frameworks, enabling comprehensive quantifications of terrestrial photosynthesis (Joiner and Yoshida, 2020; Nelson et al., 2024; Pastorello et al., 2020).

Effective machine learning upscaling depends on a complete set of input predictors that fully explain GPP dynamics. Upscaled datasets have primarily relied on satellite-observed greenness indicators, such as vegetation indices, Leaf Area Index (LAI), the fraction of absorbed photosynthetically active radiation (fAPAR), which effectively capture canopy-level GPP dynamics related to leaf area changes (Joiner and Yoshida, 2020; Ryu et al., 2019; Tramontana et al., 2016).

However, important aspects of leaf-level physiology, such as those controlled by climate factors, are
often omitted in major upscaled datasets, preventing accurate characterization of GPP responses to
climate change (Bloomfield et al., 2023; Stocker et al., 2019). In particular, none of the previous
upscaled datasets have considered the direct effect of atmospheric $CO_2$ on leaf-level photosynthesis,
which is a key factor contributing to at least half of the enhanced land carbon sink observed over the
past decades (Keenan et al., 2016, 2023; Ruehr et al., 2023; Walker et al., 2021). This omission can lead
to incorrect inferences regarding long-term trends in various components of the terrestrial carbon
cycle (De Kauwe et al., 2016).
Multiple independent lines of evidence from the atmospheric inversion (Wenzel et al., 2016),
atmospheric $^{13}C/^{12}C$ measurements (Keeling et al., 2017), ice core records of carbonyl sulfide
(Campbell et al., 2017), glucose isotopomers (Ehlers et al., 2015), as well as free-air $CO_2$ enrichment
experiments (FACE) (Walker et al., 2021), suggest a widespread positive effect of elevated atmospheric
$CO_2$ on GPP from site to global scales. Increasing atmospheric $CO_2$ *directly* stimulates the biochemical
rate or the light use efficiency (LUE) of leaf-level photosynthesis, known as the direct $CO_2$ fertilization
effect (CFE). Enhanced photosynthesis could lead to greater net carbon assimilation, contributing to
an increase in total leaf area. This expansion, contributing to a higher light interception, further
enhances canopy-level photosynthesis (i.e. GPP), which is referred to as the indirect CFE. The direct
CFE has been found to dominate GPP responses to $CO_2$ compared to the indirect effect, from both
theoretical and observational analyses (Chen et al., 2022; Haverd et al., 2020; Keenan et al., 2023).
Satellite-based estimates have shown an increasing global GPP trend in the past few decades
largely attributable to $CO_2$-induced increases in LAI (Chen et al., 2019; De Kauwe et al., 2016; Piao et
al., 2020; Zhu et al., 2016). However, previous upscaled GPP datasets, as well as most LUE models
such as the MODIS GPP product, have failed to consider the direct $CO_2$ effects on leaf-level
biochemical processes (Jung et al., 2020; Zheng et al., 2020). Consequently, these products likely
underestimated the long-term trend of global GPP, leading to large discrepancies when compared to
process-based models, which typically consider both direct and indirect $CO_2$ effects (Anav et al., 2015;
De Kauwe et al., 2016; Keenan et al., 2023; O'Sullivan et al., 2020). Notably, recent improvements in
LUE models have included the $CO_2$ response and show improved long-term changes in GPP globally
(Zheng et al., 2020), yet, this important mechanism is still missing in GPP products upscaled from *in*
*situ* eddy covariance flux measurements based on machine learning models.

To improve the quantification of GPP spatial and temporal dynamics and provide a robust

representation of long-term dynamics in global photosynthesis, we developed the CEDAR-GPP[1] data
product. CEDAR-GPP was upscaled from global eddy covariance carbon flux measurements using
machine learning along with a broad range of multi-source satellite observations and climate variables.
In addition to incorporating direct $CO_2$ fertilization effects on photosynthesis, we also account for
indirect effects via greenness indicators and include novel satellite datasets such as solar-induced
fluorescence (SIF), Land Surface Temperature (LST) and soil moisture to explain variability under
environmental stresses. We provide monthly GPP estimations and associated uncertainties at 0.05°
resolution derived from ten model setups. These setups differ by the temporal range depending on
satellite data availability, the method for incorporating the direct $CO_2$ fertilization effects, and the
partitioning approach used to derive GPP from eddy covariance measurements. Short-term model
setups are primarily based on data derived from MODIS satellites generating GPP estimates from
2001 to 2020, while long-term estimates span 1982 to 2020 using combined Advanced Very High
Resolution Radiometer (AVHRR) and MODIS data. We used two approaches to incorporate the
direct $CO_2$ fertilization effects, including direct prescription with eco-evolutionary theory and machine
learning inference from the eddy-covariance data. Additionally, we provide a baseline configuration
that did not incorporate the direct $CO_2$ effects. Uncertainties in GPP estimation were quantified using
bootstrapped model ensembles. We evaluated the machine learning models' skills in predicting
monthly GPP, seasonality, interannual variability, and trend against eddy covariance measurements,
and compared the CEDAR-GPP spatial and temporal variability to existing satellite-based GPP
estimates.

## 2. Data and Methods

### 2.1 Eddy covariance data

We obtained monthly eddy covariance GPP measurements from 2001 to 2020 from the

FLUXNET2015    (Pastorello    et    al.,    2020),    AmeriFlux    FLUXNET
(https://ameriflux.lbl.gov/data/flux-data-products/), and ICOS Warm Winter 2020 (Warm Winter
2020 Team, 2022) datasets. All data were processed with the ONEFLUX pipeline (Pastorello et al.,
2020). Following previous upscaling efforts (Tramontana et al., 2016), we selected monthly GPP data

---

[1] CEDAR stands for upsCaling Ecosystem Dynamics with ARtificial intelligence

with at least 80% of high-quality hourly or half-hourly data for temporal aggregation. High-quality
data refers to GPP derived from measured or high-quality gap-filled Net Ecosystem Exchange (NEE)
data. We further excluded large negative GPP values, setting a cutoff of -1 gC m$^{-2}$ day$^{-1}$. We utilized
GPP estimates from both the night-time (GPP_REF_NT_VUT) and day-time
(GPP_REF_DT_VUT) partitioning approaches. We classified flux tower sites according to the C3
and C4 plant categories reported in metadata and related publications when available and used a C4
plant percentage map (Still et al., 2003) otherwise. This classification information is included in
Supplementary Text S1. Our analysis encompassed 233 sites, predominantly located in North America,
Western Europe, and Australia (Figure 1). A list of the sites is provided in Appendix A. Despite their
uneven geographical distribution, these sites effectively cover a diverse range of climatic conditions
and are representative of global biomes (Figure 1c, 1d). In total, our dataset included over 18000 site-
months. Note that we did not include eddy covariance data before 2001, since it was limited to only a
few sites with only four sites containing data before 1996. This scarcity might introduce biases in the
machine learning models, particularly in the relationship between GPP and CO$_2$, leading to unreliable
extrapolations across space and time in the long-term predictions.

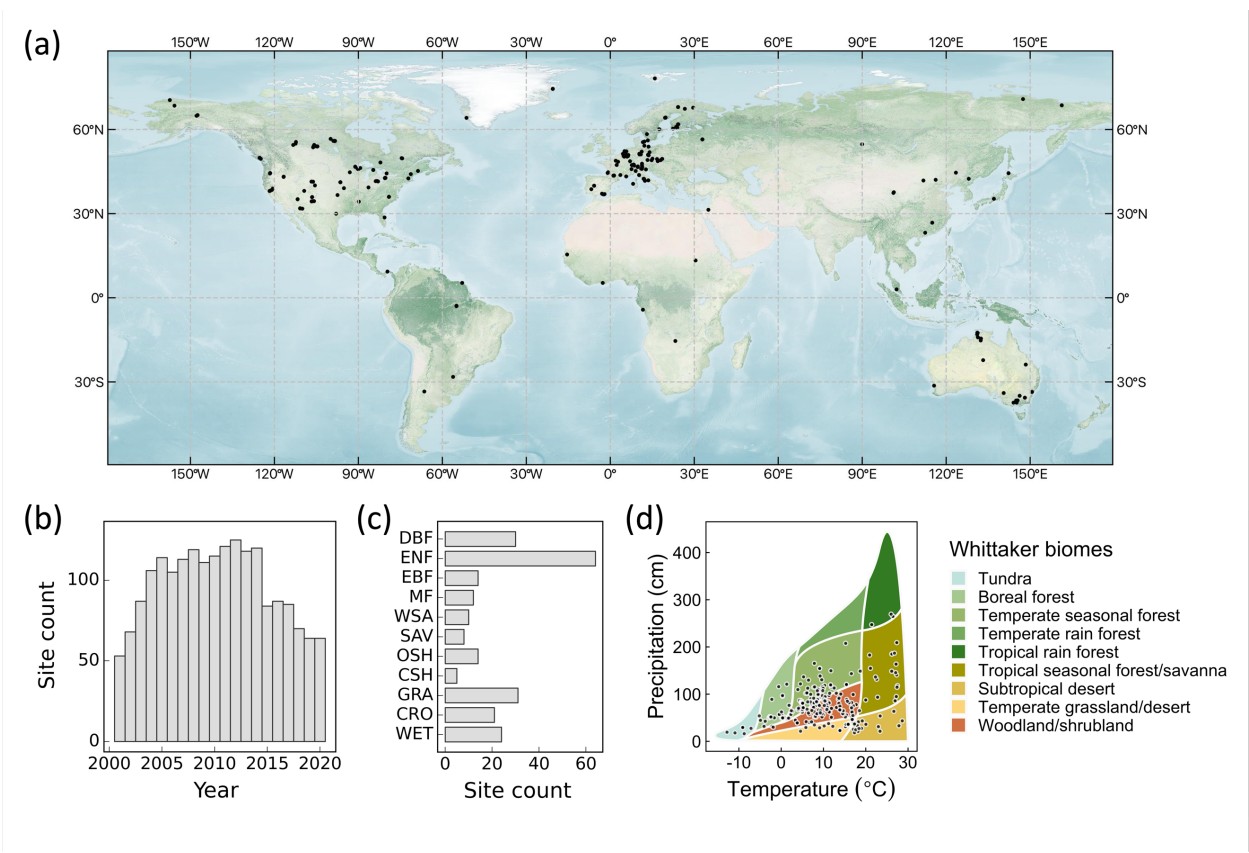


Figure 1. (a) Spatial distribution of eddy covariance sites used to generate the CEDAR-GPP product. (b) Annual site counts. (c) Site counts by biomes. ENF: evergreen needleleaf forests, EBF: evergreen broadleaf forests, DBF: deciduous broadleaf forests, MF: mixed forests, WSA: woody savannas, SAV: savannas, OSH: open shrublands, CSH: closed shrublands, GRA: grasslands, CRO: croplands, WET: wetlands. (d) Sites distributions in the annual temperature and precipitation space. Whittaker biome classification is shown as a reference of natural vegetation based on long-term climatic conditions. It does not directly indicate the actual biome associated with each site. The base map in (a) was obtained from the NASA Earth Observatory map by Joshua Stevens using data from NASA's MODIS Land Cover, the Shuttle Radar Topography Mission (SRTM), the General Bathymetric Chart of the Oceans (GEBCO), and Natural Earth boundaries. Whittaker biomes were plotted using the "plotbiomes" R package (Ștefan and Levin, 2018).

## 2.2 Global input datasets

We compiled an extensive set of covariates from gridded climate reanalysis data, multi-source satellite datasets including optical, thermal, and microwave observations, as well as categorical information on land cover, climate zone, and C3/C4 classification. The datasets that we compiled offer comprehensive information about GPP dynamics and its responses to climatic variabilities and stresses. Table 1 lists the datasets and associated variables used to generate CEDAR-GPP.

Table 1. Datasets used in different model setups to generate the CEDAR GPP product. Refer to Table S1 for a list of specific variables from each dataset.

| Category | Dataset | Temporal coverage | Spatial resolution | Temporal resolution | Usage in model setups | | Reference |
|---|---|---|---|---|---|---|---|
| | | | | | Short-term | Long-term | |
| Climate | ERA5-Land | 1950 – present | 0.1° | Monthly | ✓ | ✓ | (Sabater, 2019) |
| | ESRA Global Monitoring Laboratory Atmospheric Carbon Dioxide | 1976 – present | - | Monthly | ✓ (only in CFE-ML and CFE-Hybird setups) | ✓ (only in CFE-Hybrid setup) | (Thoning et al., 2021) |
| Satellite-based datasets | MODIS Nadir BRDF-adjusted reflectance (MCD43C4v006) | 2000 – present | 0.05° | Daily | ✓ | | (Schaaf and Wang, 2015) |
| | MODIS Terra and Aqua LAI/fPAR (MCD15A3H, MOD15A2H, v006) | 2000 – present | 500m | 4-day, 8-day | ✓ | | (Myneni et al., 2015a, b) |
| | MODIS Terra and Aqua LST (MYD11A1, MOD11A1, v006) | 2000 – present | 1 km | Daily | ✓ | | (Wan et al., 2015b, a) |
| | BESS_Rad | 2000 – 2020 | 0.05° | Daily | ✓ | | (Ryu et al., 2018) |
| | Continuous-SIF (from OCO-2 and MODIS) | 2000 – 2020 | 0.05° | 4-day | ✓ | | (Zhang, 2021) |
| | ESA CCI Soil Moisture Combined Passive and Active v06.1 | 1979 – 2021 | 0.25° | Daily | ✓ | | (Gruber et al., 2019) |
| | GIMMS LAI4g | 1982 – 2021 | 0.0833° | Half-month | | ✓ | (Cao et al., 2023) |
| | GIMMS NDVI4g | 1982 – 2021 | 0.0833 ° | Half-month | | ✓ | (Li et al., 2023b) |
| Static categorical datasets | MODIS Land Cover (MCD12Q1v006) | Average status used between 2001 and 2020 | 500m | - | ✓ | ✓ | (Friedl and Sulla-Menashe, 2019) |
| | Koppen-Geiger Climate Classification | present | 1 km | - | ✓ | ✓ | (Beck et al., 2018) |
| | C4 percentage map | present | 1° | - | ✓ | ✓ | (Still et al., 2003, 2009) |



2.2.1   Climate variables

We obtained air temperature, vapor pressure deficit, precipitation, potential evapotranspiration,

and skin temperature from the EAR5-Land reanalysis dataset (Sabater, 2019) (Table 1; Table S1). We
applied a three-month lag to precipitation, to represent the root zone water availability. Averaged
monthly atmospheric $CO_2$ concentrations were calculated as an average of records from the Mauna
Loa Observatory and South Pole Observation stations, retrieved from NOAA's Earth System
Research Laboratory (Thoning et al., 2021).
2.2.2   Satellite datasets

We assembled a broad collection of satellite-based observations of vegetation greenness and

structure, LST, solar radiation, solar-induced fluorescence (SIF), and soil moisture (Table 1, Table S1).

We used three MODIS version 6 products: surface reflectance, LAI/fAPAR, and LST. Surface

reflectance from optical to infrared bands (band 1 to 7) was sourced from the MODIS Nadir BRDF-
adjusted reflectance (NBAR) daily dataset (MCD43C4) (Schaaf and Wang, 2015). From these data, we
derived vegetation indices, including NIRv (Badgley et al., 2019), kNDVI (Camps-Valls et al., 2021),
NDVI, Enhanced Vegetation Index (EVI), Normalized Difference Water Index (NDWI) (Gao, 1996),
and the green chlorophyll index (CIgreen) (Gitelson, 2003). We also used snow percentages from the
NBAR dataset. We used the 4-day LAI and fPAR composite derived from Terra and Aqua satellites
(MCD15A3H) (Myneni et al., 2015a; Yan et al., 2016a, b) from July 2002 onwards and the MODIS 8-
day LAI and fPAR dataset from Terra only (MOD15A2H) prior to July 2002 (Myneni et al., 2015b).
We used day-time and night-time LST from the Aqua satellite (MYD11A1) (Wan et al., 2015b), with
the Terra-based LST product (MOD11A1) used after July 2002 (Wan et al., 2015a). Terra LST was
bias-corrected with the differences in the mean seasonal cycles between Aqua and Terra following
Walther et al. (2022).

We used the PKU GIMMS NDVI4g dataset (Li et al., 2023b) and PKU GIMMS LAI4g (Cao

et al., 2023) datasets available from 1982 to 2020. PKU GIMMS NDVI4g is a harmonized time series
that includes AVHRR-based NDVI from 1982 to 2003 (with biases and corrections mitigated through
inter-calibration with Landsat surface reflectance images) and MODIS NDVI from 2004 onward.
PKU GIMMS LAI4g consisted of consolidated AVHRR-based LAI from 1982 to 2003 (generated
using machine learning models trained with Landsat-based LAI data and NDVI4g) and reprocessed
MODIS LAI (Yuan et al., 2011) from 2004 onwards.
We utilized photosynthetically active radiation (PAR), diffusive PAR, and shortwave
downwelling radiation from the BESS_Rad dataset (Ryu et al., 2018). We obtained the continuous-
SIF (CSIF) dataset (Zhang, 2021; Zhang et al., 2018) produced by a machine learning algorithm trained
using OCO-2 SIF observations and MODIS surface reflectance. We used surface soil moisture from
the ESA CCI soil moisture combined passive and active product (version 6.1) (Dorigo et al., 2017;
Gruber et al., 2019).
2.2.3   Other categorical datasets
We used plant functional type (PFT) information derived from the MODIS Land Cover product
(MCD12Q1) (Friedl and Sulla-Menashe, 2019). We followed the International Geosphere-Biosphere
Program classification scheme but merged several similar categories to maximize the amount of eddy
covariance sites/observations available for each category. Closed shrublands and open shrublands are
combined into a shrubland category. Woody savannas and savannas are combined into savannas. We
generated a static PFT map by taking the mode of the MODIS land cover time series between 2001
– 2020 at each pixel to mitigate uncertainties from misclassification in the MODIS dataset.
Nevertheless, changes in vegetation structure induced by land use and land cover change are reflected
in the dynamics surface reflectance and LAI/fAPAR datasets we used. We used the Koppen-Geiger
main climate groups (tropical, arid, temperate, cold, and polar) (Beck et al., 2018). We also utilized a
C4 plant percentage map to account for different photosynthetic pathways when incorporating $CO_2$
fertilization (Still et al., 2003, 2009). The C4 percentage dataset was constant over time.
2.2.4   Data preprocessing
We implemented a three-step preprocessing strategy for the satellite datasets: 1) quality control,
2) gap-filling, and 3) spatial and temporal aggregation. Firstly, we selected high-quality data based on
the quality control flags of the satellite products when available. For the MODIS NBAR dataset
(MCD43C3), we used data with 75% or more high-resolution NBAR pixels retrieved with full
inversions for each band. For MODIS LST, we selected the best quality data from the quality control
bitmask as well as data where retrieved values had an average emissivity error of no more than 0.02.
For MODIS LAI/fAPAR, we used retrievals from the main algorithm with or without saturation. We
used all available data in ESA-CCI soil moisture due to the presence of substantial data gaps. In the
gap-filling step, missing values in satellite datasets were temporally filled at the native temporal
resolution, following a two-step protocol adapted from Walther et al (2021). Short temporal gaps were
first filled with medians from a moving window, and the remaining gaps were filled with the mean
seasonal cycle. For datasets with a high temporal resolution, including MODIS NBAR (daily),
LAI/fPAR (4-day), BESS (4-day), CSIF (4-day), ESA-CCI (daily), temporal gaps no longer than 5 days
(8 days for 4-day resolution products) were filled with medians of 15-day moving windows in the first
step. An exception is MODIS LST (daily), for which we used a shorter moving window of 9 days due
to rapid changes in surface temperature. GIMMS LAI4g and NDVI4g data were only filled with mean
seasonal cycle due to their low temporal resolution (half-month). This is because vegetation structure
could experience significant changes at half-month intervals, and gap-filling using temporal medians
within moving windows could introduce considerable uncertainties and potentially over-smooth the
time series.
Finally, all the datasets were aggregated to a monthly time step and 0.05-degree spatial resolution.
We employed the conservative resampling approach using the xESMF python package (Zhuang et al.,
2023). To generate the machine learning model training data, we extracted values from the nearest
0.05 degree pixel relative to the site locations within the gridded dataset.

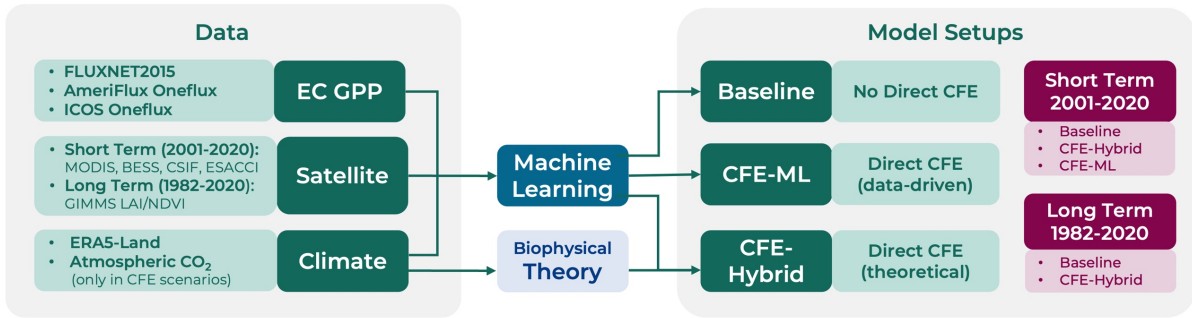


Figure 2. Schematic overview of the CEDAR-GPP model setups.

## 2.3  Machine learning upscaling

### 2.3.1  CEDAR-GPP model setups

We trained machine learning models with eddy covariance GPP measurements as targets and
climate/satellite variables as input features. We created ten model setups to produce different global
monthly GPP estimates (Figure 2; Table 2). The model setups were characterized by the temporal
range depending on input data availability, the configuration of $CO_2$ fertilization effects, and the
partitioning approach used to derive the GPP from eddy covariance measurements.
The short-term (ST) model configuration produced GPP from 2001 to 2020, and the long-term
(LT) configuration spanned 1982 to 2020. Each temporal configuration uses a different set of input
variables depending on their availability. Inputs for the short-term configuration included MODIS,

CSIF, BESS PAR, ESA-CCI soil moisture, ERA5-Land, as well as PFT and Koppen Climate zone as categorical variables with one-hot encoding. The long-term used GIMMS NDVI4g and LAI4g data, ERA5-land, PFT and Koppen climate. ESA CCI soil moisture datasets were excluded from the long-term model setups due to concerns about the product quality in the early years when the number and quality of microwave satellite data were limited (Dorigo et al., 2015). A detailed list of input features for each setup is provided in Table S1.

Regarding the direct $CO_2$ fertilization effects (CFE), we established a "Baseline" configuration that did not incorporate these effects, a "CFE-Hybrid" configuration that incorporated the effects via eco-evolutionary theory, and a "CFE-ML" configuration that inferred the direct effects from eddy covariance data using machine learning. Detailed information about these approaches is provided in Sec. 2.3.2. Furthermore, separate models were trained for GPP target variables from the night-time (NT) and daytime (DT) partitioning approaches.

Table 2 lists the characteristics of ten model setups. Due to the limited availability of eddy covariance observations before 2001, we did not apply the CFE-ML approach to the long-term setups. The CFE-ML model, when trained on data from 2001 to 2020 with atmospheric $CO_2$ ranging from 370 to 412 ppm, would not accurately predict GPP response to $CO_2$ for the period 1982 – 2000 when the CO2 levels were markedly lower (340 – 369 ppm). This is because machine learning models, especially tree-based models, could not extrapolate beyond the range of the training data.

Table 2. Specifications of the CEDAR-GPP model setups.

| Model Setup Name | Temporal range | Direct $CO_2$ Fertilization Effects | | GPP Partitioning Method |
|---|---|---|---|---|
| | | Configuration | Method | |
| ST_Baseline_NT | Short-term (ST) 2001 – 2020 | Baseline | Not incorporated | Night-time (NT) |
| ST_Baseline_DT | | | | Day-time (DT) |
| ST_CFE-Hybrid_NT | | CFE-Hybrid | Theoretical | NT |
| ST_CFE-Hybrid_DT | | | | DT |
| ST_CFE-ML_NT | | CFE-ML | Data-driven | NT |
| ST_CFE-ML_DT | | | | DT |
| LT_Baseline_NT | Long-term (LT) 1982 – 2020 | Baseline | Not incorporated | NT |
| LT_Baseline_DT | | | | DT |
| LT_CFE-Hybrid_NT | | CFE-Hybrid | Theoretical | NT |
| LT_CFE-Hybrid_DT | | | | DT |

2.3.2   $CO_2$ fertilization effect

We established three configurations regarding the direct $CO_2$ fertilization effects on photosynthesis. In the baseline configuration, we trained machine learning models with eddy

covariance GPP, input climate and satellite features, but excluding $CO_2$ concentration. As such, the
models only include indirect $CO_2$ effects from the satellite-based proxies of vegetation greenness or
structure representing changes in canopy light interception, and they do not consider the direct effect
of $CO_2$ on leaf-level photosynthetic rates (or light use efficiency, LUE). Our baseline model is
therefore directly comparable to other satellite-derived GPP products that only account for indirect
CO2 effects (Joiner and Yoshida, 2020; Jung et al., 2020).

In the CFE-ML configuration, we added monthly $CO_2$ concentration into the feature set in

addition to those incorporated in the baseline models. Models inferred the functional relationship
between GPP and $CO_2$ from the eddy covariance data. They thus encompass both $CO_2$ fertilization
pathways – direct effects on LUE and indirect effects from the satellite-based proxies of vegetation
greenness and structure.

In the CFE-Hybrid configuration, we applied biophysical theory to estimate the response of

LUE to elevated $CO_2$, i.e. the direct CFE (Appendix B). First, we estimated a reference GPP, where
LUE was not affected by any increase in atmospheric $CO_2$, by applying the CFE-ML model with a
constant atmospheric $CO_2$ concentration equal to the 2001 level while keeping all other variables
temporally dynamic. Then, the impacts of $CO_2$ on LUE were prescribed onto the reference GPP
estimates using a theoretical $CO_2$ sensitivity function of LUE according to the optimal coordination
theory (Appendix B). The theoretical $CO_2$ sensitivity function represents a $CO_2$ sensitivity that is
equivalent to that of the electron-transport-limited (light-limited) photosynthetic rate. When light is
limited, elevated $CO_2$ suppresses photorespiration leading to increased photosynthesis at a lower rate
than when photosynthesis is limited by $CO_2$ (Lloyd and Farquhar, 1996; Smith and Keenan, 2020).
Thus, the CFE-Hybrid scenario provides a conservative estimation of the direct $CO_2$ effects on LUE.
Note that the theoretical sensitivity function describes the fractional change in LUE due to direct $CO_2$
effects relative to a reference period (i.e. 2001). Therefore, we used the CFE-ML model to establish
this reference GPP by fixing the $CO_2$ effects to the 2001 level, rather than simply using the GPP from
the Baseline model in which the direct $CO_2$ effects were not represented. Long-term trends from the
reference and the Baseline models are consistent.

For both CFE-ML and CFE-Hybrid scenarios, we made another conservative assumption that

C4 plants do not benefit from elevated $CO_2$, despite potential increases in photosynthesis during
water-limited conditions due to enhanced water use efficiency (Walker et al., 2021). Data from flux
tower sites dominated by C4 plants were removed from our training set, so the machine learning
models inferred $CO_2$ fertilization only from flux tower sites dominated by C3 plants. When applying

models globally, we assumed the reference GPP values (with constant atmospheric $CO_2$ concentration equal to the 2001 level) to represent C4 plants, and GPP estimates from CFE-ML or CFE-Hybrid models were applied in proportion to the percentage of C3 plants in a grid cell.

### 2.3.3  Machine learning model training and validation

We employed the state-of-the-art XGBoost machine learning model, known for its high accuracy in regression problems across various domains, including environmental and ecological predictions (Berdugo et al., 2022; Chen and Guestrin, 2016; Kang et al., 2020). XGBoost is a scalable and parallelized implementation of the gradient boosting technique that iteratively trains an ensemble of decision trees, with each iteration targeting to minimize the residuals from the last iteration. A notable merit of XGBoost is its ability to make predictions in the presence of missing values, a common issue in remote sensing datasets. The model is also robust to multi-collinearity between the predictors in our dataset, particularly for the variables derived from MODIS data.

We used five-fold cross-validation for model evaluation. Training data was randomly split into five groups (folds), with each fold held out for testing while the rest four folds were used for model training. We imposed two restrictions on fold splitting: each flux site was entirely assigned to a fold to test model performance over unseen locations; the random sampling was stratified based on PFT to ensure coverage of the full range of PFTs in both training and testing. Additionally, co-located sites, defined as those within 0.05° of each other, were also assigned to the same fold, as they were often setup as a cluster with different treatments. This approach avoids conflated estimates of model uncertainty, as these sites are not independent. We also used a nested-cross-validation strategy, during which we performed a randomized search of hyperparameters using three-fold cross-validation within the training set. The nested-cross-validation was aimed to reduce the risk of overfitting and improve the robustness of the evaluation.

We assessed the models' ability to capture the temporal and spatial characteristics of GPP, including monthly GPP, mean seasonal cycles, monthly anomalies, and cross-site variability. Model performance was assessed separately for each model setup (Table 2) and summarized by PFT and Koppen climate zone. Mean seasonal cycles were calculated as the mean monthly GPP over the site observation period, and monthly anomalies were the residuals of monthly GPP after subtracting mean seasonal cycles. Monthly GPP averaged over years for each site was used to assess cross-site variability. Goodness-of-fit metrics include RMSE, bias, and coefficient of determination ($R^2$).

To evaluate the models' ability to capture long-term GPP trends, we aggregated the monthly

GPP to annual values following Chen et al. (2022) , which detected the $CO_2$ fertilization effect across
global eddy covariance sites. For sites with at least five years of observations, GPP anomalies were
computed by subtracting the multi-year mean GPP from the annual GPP for each site. Anomalies
were aggregated across sites to achieve a single multi-site GPP anomaly per year. We excluded a site-
year if less than 11 months of data was available and used linear interpolation to fill the remaining
temporal gaps. This resulted in 81 sites used in the GPP trend evaluation. We used the Sen slope and
Mann-Kendall test to examine the GPP trends from 2002 to 2019, excluding 2001 and 2020, due to
the limited number of available sites with more than five years of data. We further assessed the
aggregated annual trend by grouping the sites based on plant functional types and the Koppen climate
zones. Categories with less than six long-term sites available were excluded from the analysis, which
includes EBF and Tropics.

To further analyze GPP responses to $CO_2$ in the CFE-ML models, we leveraged two explainable

machine learning approaches: ALE (Accumulated Local Effects) (Apley and Zhu, 2020; Baniecki et
al., 2021) and SHAP (SHapley Additive exPlanations) (Lundberg and Lee, 2017). SHAP is a model
interpretation method derived from game theory, providing a value for each feature's contribution to
a prediction, elucidating how each feature impacts the model's output in a specific instance. Conversely,
ALE quantifies the average effect of a feature across the data, isolating its impact by aggregating local
effects and avoiding the biases associated with correlated features.
2.3.4   Product generation and uncertainty quantification

In the CEDAR-GPP product, we generated GPP estimates from ten model setups, by applying

the model to global gridded datasets (Table 2). GPP estimates were named after the corresponding
model setups. We used bootstrapping to estimate prediction uncertainties. For each model setup, we
generated 30 bootstrapped sample sets of eddy covariance data, which were then used to train an
ensemble of 30 XGBoost models. The bootstrapping was performed at the site level, and each
bootstrapped sample set contained around 140 to 150 unique sites, 17000 to 19000 site months
covering all PFTs. The relative PFT composition in the bootstrapped sample sites was consistent with
the full dataset. Hyperparameters of the XGBoost models used in the final product generation were
described in Supplementary Text S2. The 30 models trained with bootstrapped samples generated an
ensemble of 30 GPP values. We provided the ensemble GPP mean and used standard deviation to
indicate uncertainties, for each of the ten model setups.

## 2.4 Product inter-comparison

We compared the global spatial and temporal patterns of CEDAR-GPP with other major satellite-based GPP products, including three machine learning upscaled and two LUE-based datasets. We obtained two FLUXCOM products (Jung et al., 2020), the latest version of FLUXCOM-RS (FLUXCOM-RSv006) available from 2001 to 2020 based on remote sensing (MODIS collection 6) datasets only, as well as the FLUXCOM-RS+METEO ensemble available between 1979 to 2018 and based on the climatology of remote sensing observations and ERA5 forcings (hereafter FLUXCOM-ERA5). We used FluxSat (Joiner and Yoshida, 2020), available from 2001 to 2019, which is an upscaled dataset based on MODIS NBAR surface reflectance and PAR from Modern-Era Retrospective analysis for Research and Applications 2 (MERRA-2). Importantly, FluxSat does not incorporate climate forcings. We used the MODIS GPP product (MOD17) available since 2001, which was generated based on MODIS fAPAR and LUE as a function of air temperature and vapor pressure deficit but not atmospheric $CO_2$ concentration (Running et al., 2015). We also used the rEC-LUE products, available from 1982 to 2018 and based on a revised LUE model that incorporated the effect of atmospheric $CO_2$ concentration and the fraction of diffuse PAR on LUE (Zheng et al., 2020). Additionally, to evaluate GPP trends, we further included three process-based models forced by remote sensing data – BEPS (Leng et al., 2024), BESSv2 (Li et al., 2023a), and PML V2 (Zhang et al., 2019). These products estimate GPP by scaling leaf-level biochemical photosynthesis models to the canopy level, using satellite-derived vegetation structural variables such as LAI. All three products incorporate the direct $CO_2$ effects within their biochemical photosynthesis models.

All datasets were resampled to 0.1 ° spatial resolution, and a common mask for the vegetated land area was applied. We evaluated global mean annual GPP, mean seasonal cycle, interannual variability, and trend among different datasets, comparing them over a common time period determined by their data availability. Global total GPP was computed by scaling the global area-weighted average GPP flux with the global land area (122.4 million $km^2$) following Jung et al. (2020). Mean seasonal cycle was defined as above (Sec. 2.3.3). We used the standard deviation of annual GPP to indicate the magnitude of interannual variability, the Sen slope to indicate the GPP annual trend, and the Mann-Kendall test for the statistical significance of trends.

# 3. Results

## 3.1 Evaluation of model performance

### 3.1.1 Overall performance

The short-term and long-term models explain approximately 72% and 67%, respectively, of the variation in monthly GPP across global eddy covariance sites (Figure 3a). The long-term models consistently yield lower performance than the short-term models, likely due to differences in the satellite remote sensing datasets used, as the short-term models benefited from richer information from surface reflectance of individual bands, LST, CSIF, as well as soil moisture, while the long-term model only exploited NDVI and LAI. The models with different CFE configurations and target GPP variables (i.e. partitioning approaches) have similar performance in predicting monthly GPP (Figure 3b, Table S2). All models exhibit minimal bias of less than 0.1.

Model performance in terms of the different temporal and spatial characteristics of monthly GPP is variable (Figure 3c-h). The models are most successful at predicting mean seasonal cycles, with the short-term and long-term models explaining around 77% and 72% of the variability, respectively (Figure 3c-d). The short-term and long-term models capture 63% and 54% , respectively, of the spatial variabilities in multi-year mean GPP across global sites (i.e., cross-site variability) (Figure 3g-h). However, all models underestimate monthly anomalies across the sites, with $R^2$ values below 0.12 (Figure 3e-f). Patterns from the DT setups do not significantly differ from those of the NT setups (Figure S1, Table S2). Model performance also varies across sites, and models are more advantageous in explaining mean seasonal cycles than monthly anomalies in most sites (Figure S2).

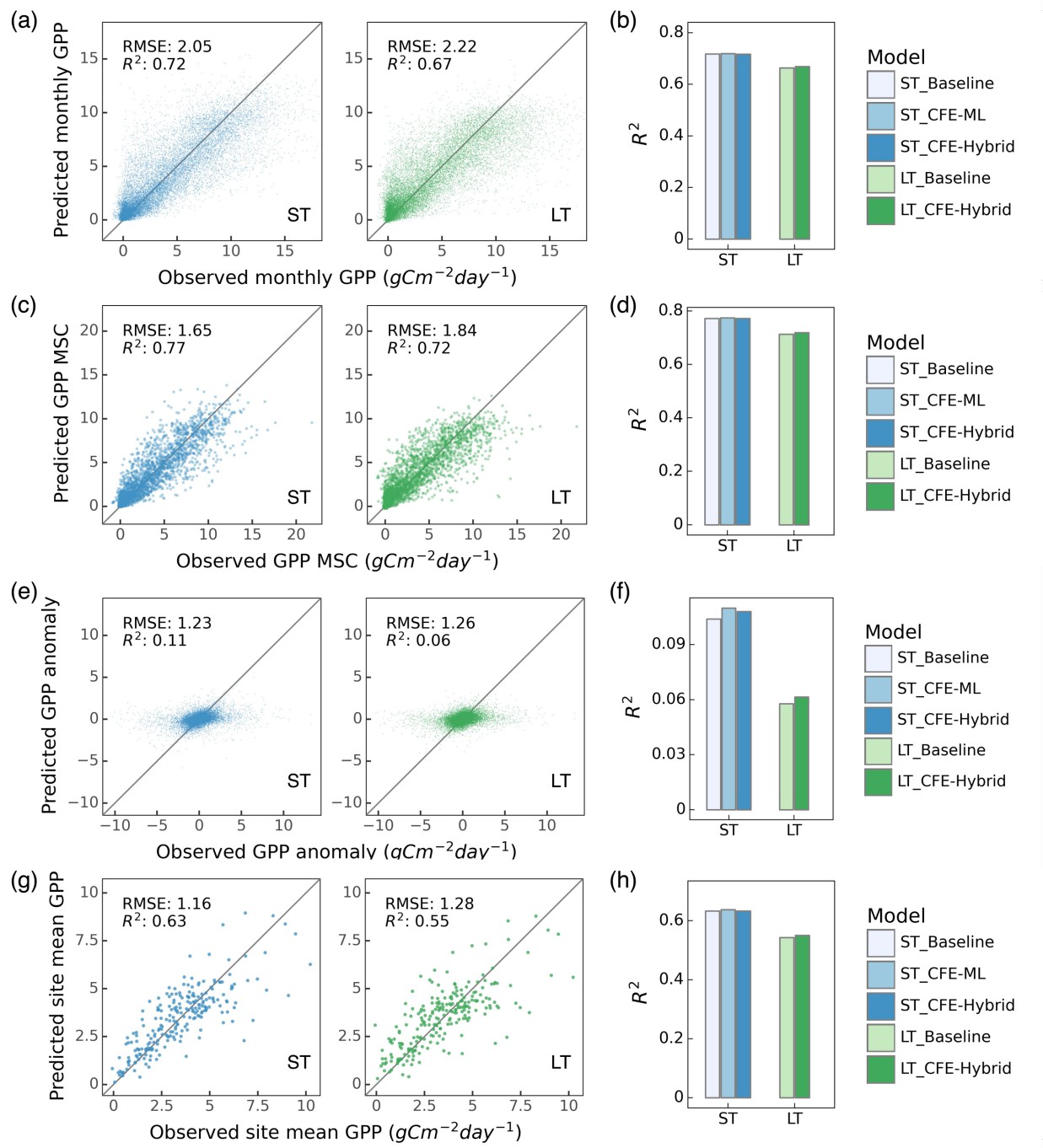

Figure 3. Machine learning model performance in predicting monthly GPP and its spatial and temporal variability. Only NT models are shown and DT results are provided in Supplementary Figure S1. Scatter plots illustrated relationships between model predictions and observations for monthly GPP (a), mean seasonal cycles (MSC) (c), monthly anomaly (e), and cross-site variability (g) for ST_CFE-Hybrid_NT (left, blue) and LT_CFE-Hybrid_NT (right, green) models. Corresponding bar plots show the $R^2$ values for five NT model setups in predicting monthly GPP (b), MSC (d), monthly anomaly (f), and cross-site variability (h).

3.1.2    Performance by biome and climate zone

The predictive ability of our models varies across different PFTs and Koppen climate zones

(Figure 4). Here we present results from the CFE-Hybrid LT and ST models based on NT partitioning
and note that patterns for the other CFE configurations and the DT GPP were similar (Figure S3, S4,
S5).

Model performance in terms of monthly GPP is the highest for deciduous broadleaf forests,

mixed forests, and evergreen needleleaf forests, with $R^2$ values above 0.76. Model accuracies are also
high for savannas and grasslands, followed by croplands and wetlands, with $R^2$ values between 0.48
and 0.76. Model accuracies are lowest in evergreen broadleaf forests and shrublands, with $R^2$ values
as low as 0.13. Across climate zones, models achieve the highest accuracy in predicting monthly GPP
in cold climates with $R^2$ around 0.73 – 0.78, followed by tropics and temperate zones ($R^2 \sim 0.47$ –
0.65). The short-term models have the lowest performance in polar regions with an $R^2$ value of around
0.37, and the long-term models have the lowest performance in arid regions with an $R^2$ value of 0.28.
Interestingly, short-term and long-term models exhibit substantial differences in arid regions and
shrublands marked by strong seasonality and interannual variabilities.

Model performance in terms of mean seasonal cycles across PFTs and climate zones follows

patterns for monthly GPP, while disparities emerge for performance in terms of GPP anomaly and
cross-site variability (Figure 4, Figure S3, S4, S5). The short-term model shows the highest predictive
power in explaining monthly anomalies in arid regions with an $R^2$ value of 0.48, where savanna and
shrublands sites are primarily located. Model performance in all other climate zones is significantly
lower. The short-term model also demonstrates good performance in capturing anomalies in
deciduous broadleaf forests. The long-term model's relative performance between PFTs and climate
zones is mostly consistent with that of the short-term model, with lower accuracy in shrublands when
compared to the short-term model.

Models demonstrate the highest accuracy in predicting cross-site variability in savannas,

grasslands, evergreen needleleaf forests, and evergreen broadleaf forests ($R^2 > 0.36$) and the lowest
accuracy in deciduous broadleaf forests, mixed forests, and croplands ($R^2 < 0.1$). The short-term
model additionally shows good performance in shrublands and wetlands ($R^2 > 0.36$), whereas the
long-term model fails to capture any variability for shrublands. In terms of climate zones, models are
most successful at explaining the variabilities within tropical and cold climate zones ($R^2 > 0.50$), the
short-term model has moderate performance in temperature and polar regions ($R^2 \sim 0.22$), and the
long-term model has low performance for both temperate and arid regions with $R^2$ values below 0.16.

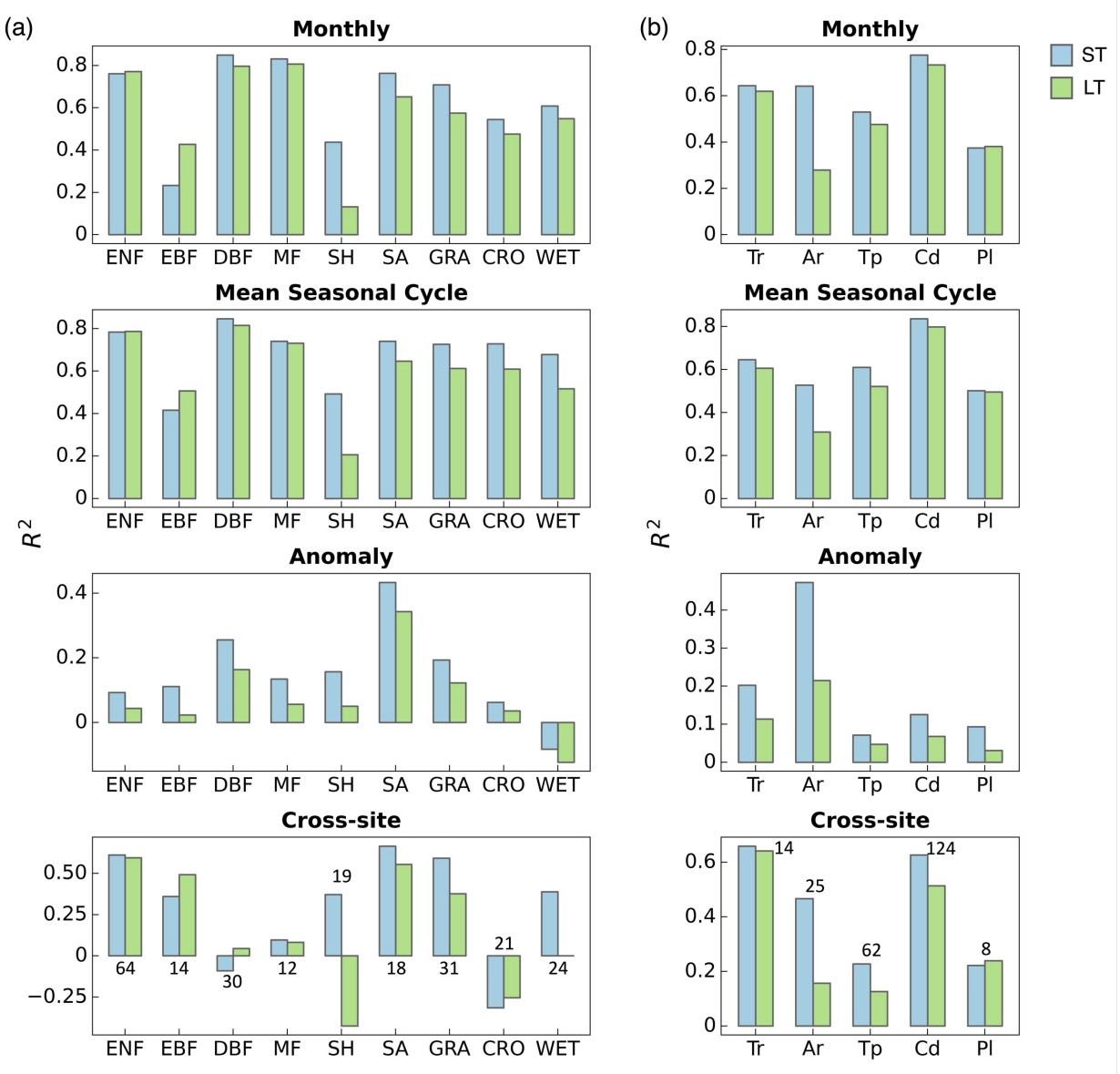

Figure 4. Performance of the ST_CFE-Hybrid_NT (blue) and LT_CFE-Hybrid_NT (green) models on GPP spatiotemporal estimation by plant functional types (a) and climate zones (b). The cross-site panels included the number of sites within each category. Color indicates short-term (ST) or long-term (LT) models. ENF: evergreen needleleaf forest, EBF: evergreen broadleaf forest, DBF: deciduous broadleaf forest, MF: mixed forest, SH: shrubland, SA: savanna, GRA: grassland, CRO: cropland, WET: wetland. Tr: tropical, Ar: arid, Tp: temperate, Cd: cold, Pl: polar. The performance of DT models is displayed in Supplementary Figure S3.

### 3.1.3 Prediction of long-term trends

Eddy covariance derived GPP presents a substantial increasing trend across flux sites between 2002 and 2019 (Figure 5a, Figure S6a). The eddy covariance GPP from the night-time partitioning

approach indicates an overall trend of 7.7 gC m$^{-2}$ year$^{-2}$. In contrast, the ST_ Baseline_NT model
predicts a more modest overall trend of 3.1 gC m$^{-2}$ year$^{-2}$ across the flux sites, primarily reflecting the
indirect $CO_2$ effect manifested through the growth of LAI. Both the ST_CFE-ML_NT and ST_CFE-
hybrid_NT models predict much higher trends of 5.4 and 4.5 gC m$^{-2}$ year$^{-2}$ respectively, representing
an improvement from the Baseline model by 74% and 45%, aligning more closely to eddy covariance
observations. Similarly, the LT_CFE-Hybrid_NT model shows an improved trend estimation than
the LT_Baseline_NT model. All trends were statistically significant ($p < 0.05$).Aggregated eddy
covariance GPP experiences increasing trends of varied magnitudes across different climate zones and
plant functional types (Figure 5b,c; Figure S6b,c). While the machine learning models generally do not
fully capture the enhancement in GPP for most categories, the CFE-ML and/or CFE-hybrid models
consistently outperform the Baseline models in both ST and LT setups. The CFE-ML setup predicts
a higher trend than CFE-hybrid in most cases, suggesting that the data-driven approach captures more
dynamics not represented in the theoretical model, which is based on conservative assumptions
regarding the $CO_2$ sensitivity of photosynthesis (see Sect. 2.3.2 and Appendix B). The choice of remote
sensing data (ST vs. LT configurations) does not lead to substantial differences in the predicted GPP
trend. Most long-term flux sites (at least 10 years of records) with a significant trend experienced an
increase in GPP, and the CFE-ML and/or CFE-hybrid models align closer to eddy covariance data
than the Baseline models (Figure S7). Additionally, we found a considerably higher trend in eddy
covariance GPP measurements derived from the day-time versus night-time partitioning approach,
potentially associated with uncertainties in GPP partitioning methods (Figure S6). Yet, machine
learning model predicted trends are not strongly affected by GPP partitioning methods.

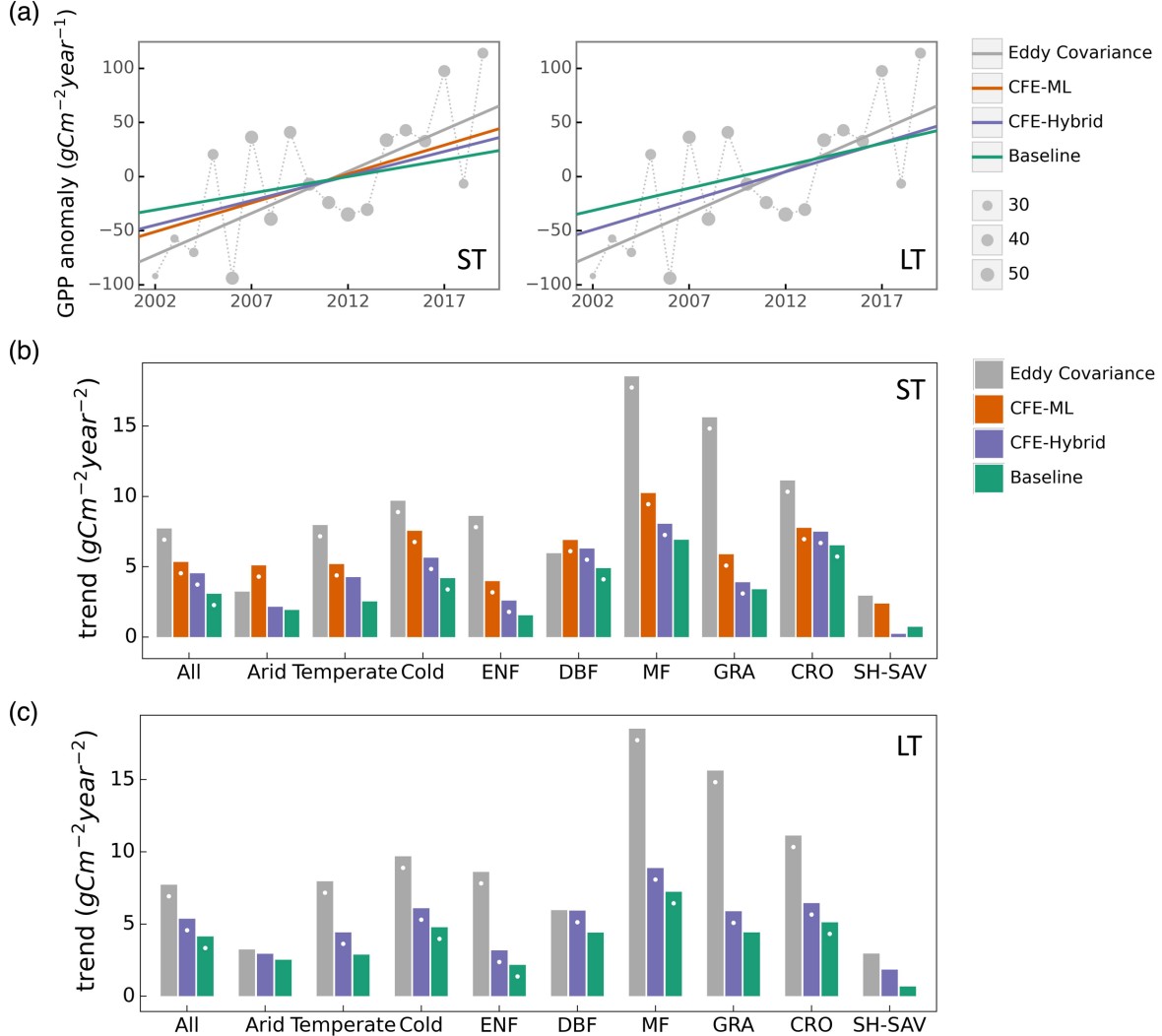

Figure 5. Comparison of observed and predicted GPP (from NT models only) trends across eddy covariance flux towers. (a) Aggregated annual GPP anomaly from 2002 to 2019 and trend lines from eddy covariance (EC) data, and three CFE model setups (short-term, night-time partitioning) for ST (left) and LT (right) models. The size of grey circle markers is proportional to the number of sites. (b) Comparison of annual GPP trends from eddy covariance measurements and the short-term (ST) CEDAR-GPP model setups by plant functional types and climate zones. (c) Comparison of annual GPP trends from eddy covariance measurements and the long-term (LT) CEDAR-GPP model setups by plant functional types and climate zones. In (b) and (c), Categories with less than 6 sites, including Tropics and EBF, were not shown. While dots on the bars indicate statistically significant trends with p-value < 0.1. Results for the DT models are shown in Supplementary Figure S3.

The differences in estimated GPP trends between the Baseline and CFE models underscore the significant long-term GPP changes driven by the direct $CO_2$ effect. Using explainable machine learning

approaches – ALE and SHAP, we further assessed the CFE-ML models for quantifying the direct $CO_2$ effect. Both approaches reveal a consistently positive influence of $CO_2$ on GPP, aligning with biophysical theories (Figure S8). Compared to the effects from light (PAR) and vegetation structures (e.g. NIRv), the impacts of $CO_2$ are considerably smaller, which explains the minimal differences in overall model accuracy between the Baseline and CFE models.

Finally, we evaluate CEDAR-GPP using independent eddy covariance data (11 sites, Table S3) that was not involved in model training and obtained from the OzFlux FluxNet dataset (Ozflux, 2024). Among these sites, only two - AU-Cpr (Tropical) and AU-Stp (Aird) - with more than five years of records exhibit a GPP trend with p-value less than 0.3. CEDAR-GPP shows strong consistency with the observed trend (Figure S9). Additionally, CEDAR-GPP achieves reasonable accuracy in predicting monthly GPP ($R^2 \sim 0.73 – 0.75$), mean seasonal cycle ($R^2 \sim 0.74 – 0.78$), and monthly anomalies ($R^2 \sim 0.26 – 0.50$) (Table S4, Figure S10), closely aligning with the cross-validation results.

## 3.2 Evaluation of GPP spatial and temporal dynamics

We compared CEDAR-GPP estimates with other upscaled or LUE-based datasets regarding the mean annual GPP (Sect. 3.2.1), GPP seasonality (Sect. 3.2.2), interannual variability (Sect. 3.2.3), and annual trends (Sect. 3.2.4). CEDAR-GPP model setups generally show similar patterns in mean annual GPP, seasonality, and interannual variability, therefore, in corresponding sections, we present the CFE-Hybrid model setups as representative examples for comparisons with other datasets, unless otherwise stated. Supplementary figures include comparisons involving CEDAR-GPP estimates from all model setups.

### 3.2.1 Mean annual GPP

Global patterns of mean annual GPP are generally consistent among CEDAR-GPP model setups, FLUXCOM, FLUXSAT, MODIS, and rEC-LUE, with few noticeable regional differences (Figure 6, Figure S11). Differences among CEDAR-GPP model setups are minimal and only evident between the NT and DT setups in the tropics (Figure 6b-c, Figure S11). CEDAR-GPP short-term datasets show highest consistency with FLUXSAT in terms of mean annual GPP magnitudes (2001 – 2018) and latitudinal variations, although FLUXSAT presents slightly higher GPP values in the tropics compared to CEDAR-GPP (Figure 6b). Mean annual GPP magnitudes for FLUXCOM-RS006 and MODIS are lower globally than CEDAR-GPP and FLUXSAT, with the most pronounced differences observed in the tropical areas. Among the long-term datasets (CEDAR-GPP LT, FLUXCOM-ERA5,

and rEC-LUE), mean annual GPP (1982 – 2018) exhibits greater disparities in the northern mid-
latitudes than in the tropics and southern hemisphere (Figure 6c). CEDAR-GPP aligns more closely
with FLUXCOM-ERA5 than with rEC-LUE, with the latter showing lower annual mean GPP globally,
particularly between 20ºN to 50º N.

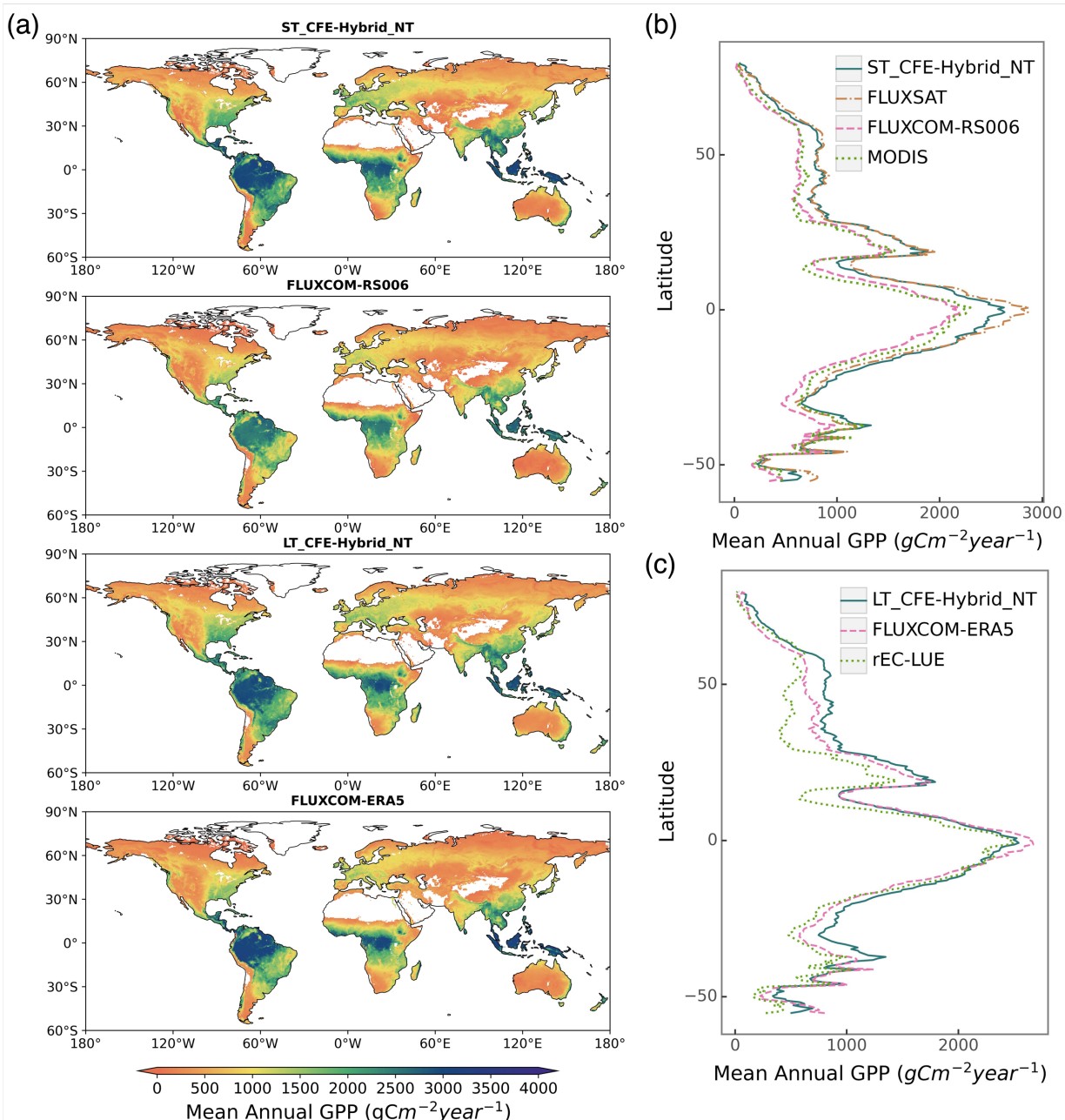


Figure 6. Global distributions of mean annual GPP from CEDAR-GPP and other
machine learning upscaled and LUE-based reference datasets. (a) Global patterns of
mean annual GPP from two short-term datasets including ST_CFE-Hybrid_NT, and
FLUXCOM-RS006, and two long-term datasets including LT_CFE-Hybrid_NT, and
FLUXCOM-ERA5. (b) Latitudinal distributions of mean annual GPP from short-term

datasets (ST_CFE-Hybrid_NT, FLUXSAT, FLUXCOM-RS006, and MODIS). (c) Latitudinal distributions of mean annual GPP from long-term datasets (LT_CFE-Hybrid_NT, FLUXCOM-ERA5, and rEC-LUE). Mean annual GPP was computed between 2001 and 2018 for short-term datasets and between 1982 and 2018 for long-term datasets.

### 3.2.2  Seasonal variability

CEDAR-GPP agrees with other GPP datasets on seasonal variabilities (average between 2001 and 2018) at the global scale, characterized by a peak in GPP in July and a nadir between December and January (Figure 7, Figure S12). At the global scale, CEDAR-GPP is most closely aligned with FLUXSAT in GPP seasonal magnitude and amplitude, while both FLUXCOM and MODIS display a relatively less pronounced magnitude.

In boreal and temperate regions of the Northern Hemisphere, all datasets agree on seasonal GPP variation, with only minor variances in the magnitude of peak GPP. In Southern Hemisphere temperate regions, datasets demonstrate similar seasonality, though with greater variability in peak amplitudes compared to the Northern Hemisphere. The largest disparities are found in the South American tropical areas, where seasonal variation is less prominent. Here, FLUXSAT shows a distinct bi-modal pattern with peaks in March-April and September-October. CEDAR-GPP and FLUXCOM-ERA5 aligns with the second peak, but exhibit a less pronounced first peak. Interestingly, the DT setups of CEDAR-GPP show slightly higher peaks in March-April in this region (Figure S13). MODIS, in contrast, indicates an inverse seasonal pattern with a small peak from June to August. Across all regions, CEDAR-GPP's seasonality aligns more closely with FLUXSAT and FLUXCOM-ERA5 than with other datasets. Differences among the ten CEDAR-GPP model setups are minimal, except for small variations in GPP magnitude in some tropical areas between NT and DT setups (Figure S13).

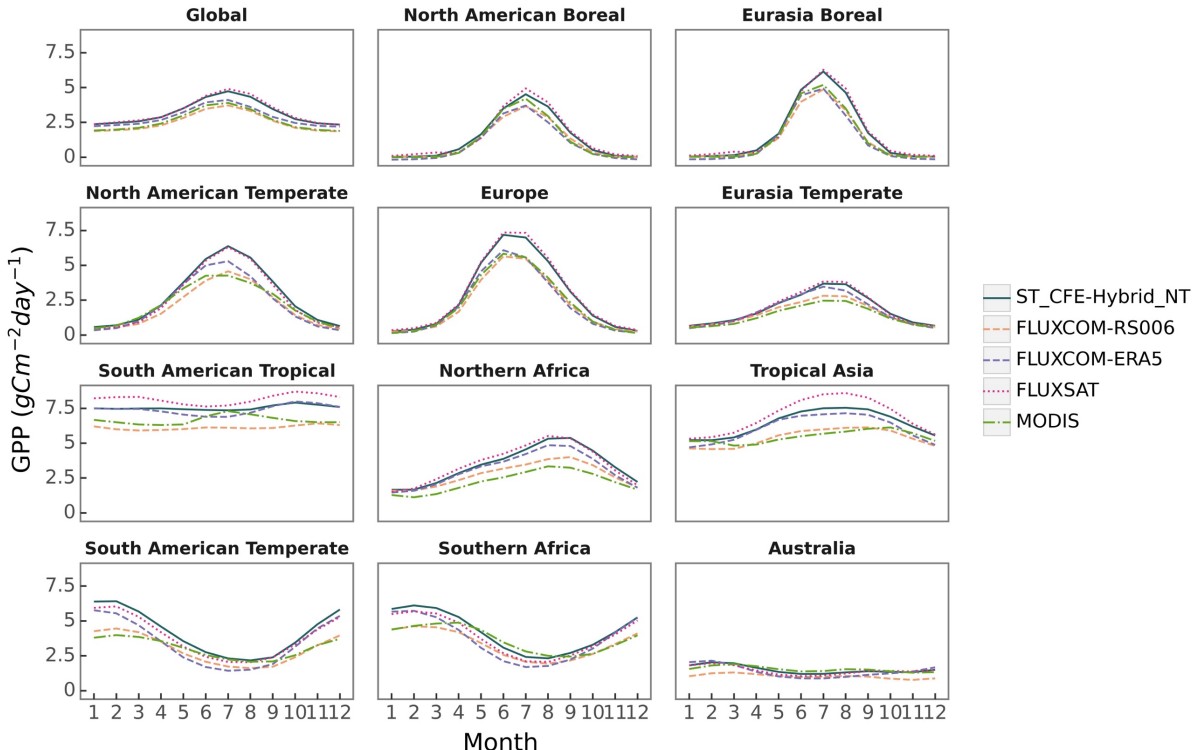

Figure 7. Comparison of global and regional GPP mean seasonal cycle between different datasets on a global scale. Monthly means were averaged from 2001 to 2018 for all datasets. Geographic boundaries of the 11 TransCom land regions were obtained from the CarbonTracker (CT2022) dataset and shown in Figure S18.

### 3.2.3 Interannual variability

We found distinct spatial patterns in GPP interannual variability between upscaled and LUE-based datasets and a high level of agreement within each category, with the exception of FLUXCOM-ERA5, which show minimal interannual variability globally (Figure 8, Figure S14). All datasets agree on the presence of GPP interannual variability hotspots in eastern and southern South America, central North America, southern Africa, and western Australia. These hotspots primarily correspond to arid and semi-arid areas characterized by grasslands, shrubs, and croplands (Figure 9). CEDAR-GPP is highly consistent with FLUXSAT, and both datasets also display relatively high interannual variability in the dry subhumid areas of Europe, predominantly covered by croplands. FLUXCOM-RS006 mirrors the relative spatial patterns of CEDAR-GPP and FLUXSAT, albeit at lower magnitudes. The LUE-based datasets (MODIS and rEC-LUE) predict a much higher interannual variability than the upscaled datasets in the tropical areas, particularly in evergreen broadleaf forests and woody savannas (Figure 8, Figure 9). These datasets also depict slightly higher interannual variability for other types of forests, including evergreen needleleaf forests and deciduous broadleaf

forests, compared to the upscaled datasets. The lack of interannual variability in FLUXCOM-ERA5
is attributable to the use of mean seasonal cycles of remotely sensed vegetation greenness indicators
rather than their dynamic time series. Ten CEDAR-GPP model setups present consistent patterns in
interannual variability, and differences were minimal (Figure S14).

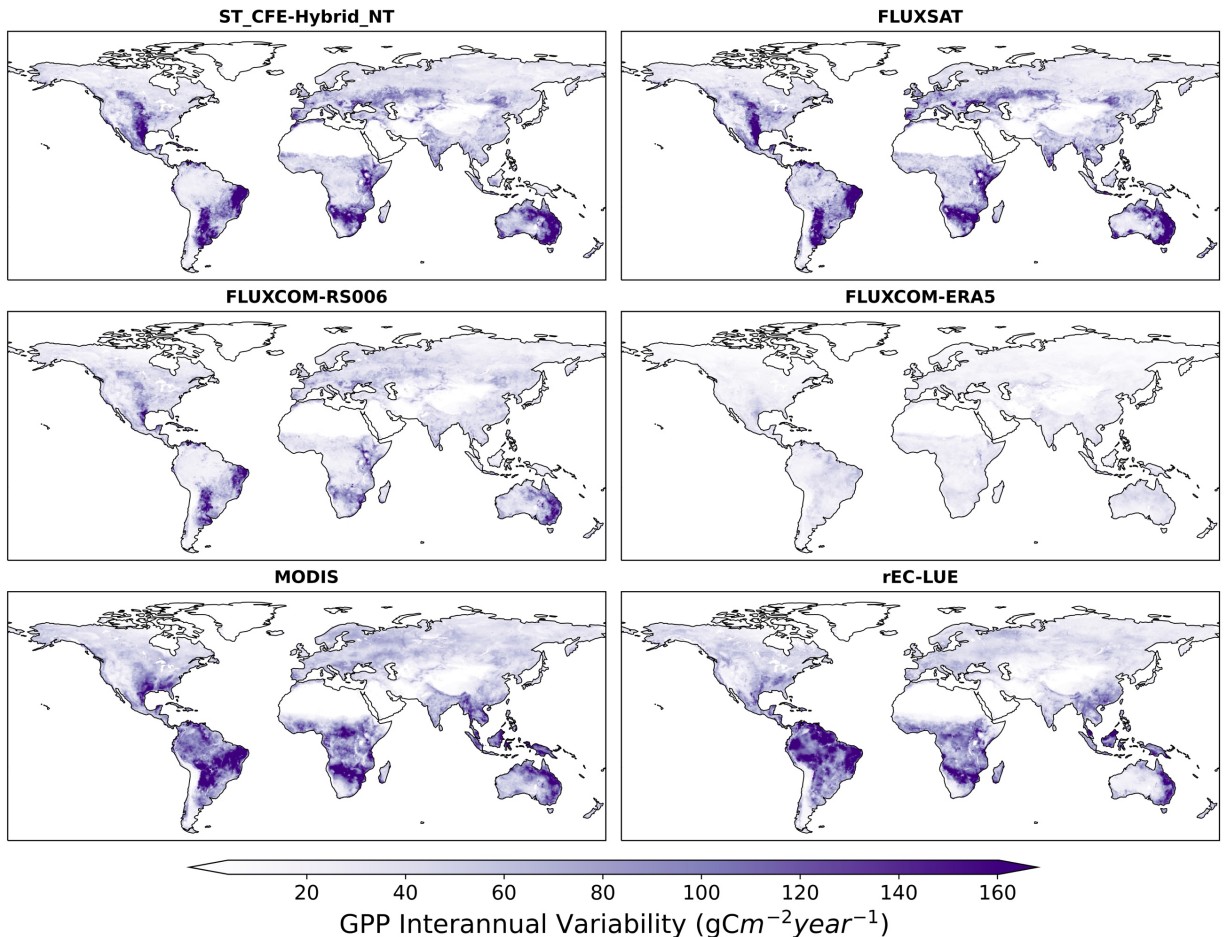


Figure 8. Spatial patterns of GPP interannual variability extracted over 2001 to 2018
for CEDAR-GPP (ST_CFE-Hybrid_NT), FLUXSAT, FLUXCOM-RS006, MODIS,
FLUXCOM-ERA5, and rEC-LUE.

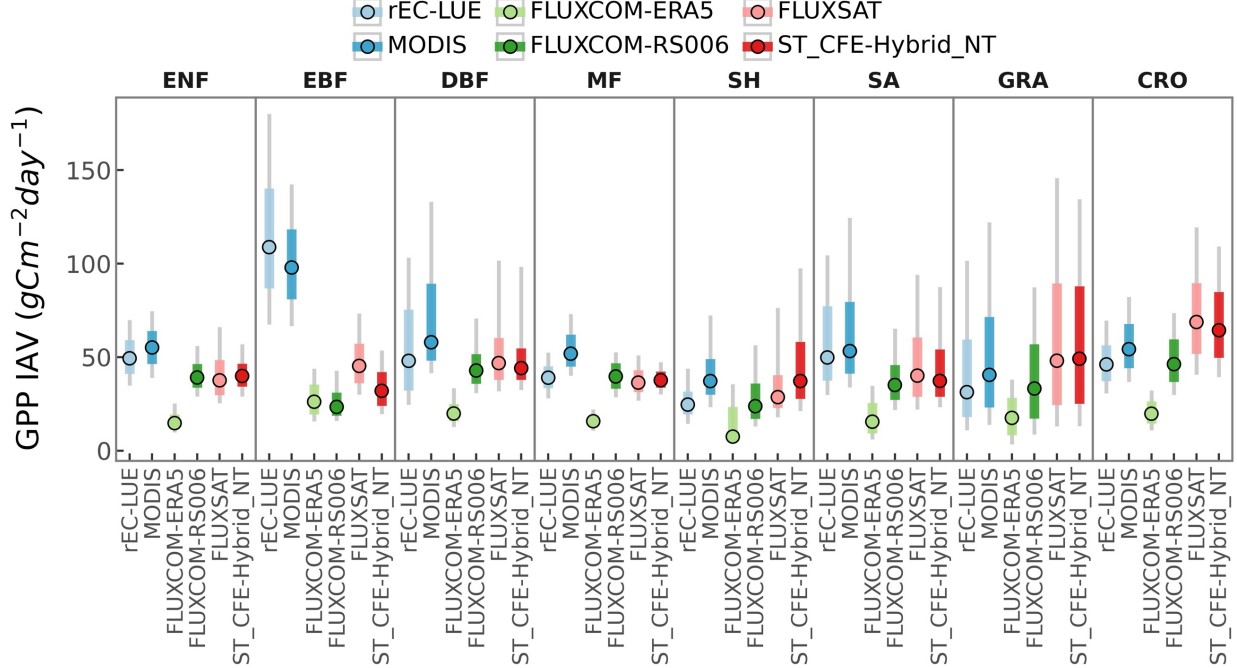

Figure 9. Comparison of GPP interannual variability (IAV) across global datasets by PFT. Colored dots represent the median IAV, thicker gray bars indicate the 25% to 75% percentiles of IAV distributions, and thinner grey bards show the 10% to 90% percentiles.

### 3.2.4 Trends

Differences in annual GPP trends among CEDAR-GPP model setups and other upscaled and LUE-based datasets mainly reflect the variability in the representation of $CO_2$ fertilization effects (Figure 10, 11, Figure S15). From 2001 to 2018, the CEDAR-GPP Baseline model setups show spatial variations in GPP trends consistent with the other upscaled datasets without direct $CO_2$ fertilization effects, including FLUXSAT and FLUXCOM-RSv006. In these datasets, substantial increases are seen in southeastern China and India, western Europe, and part of North and South America. These increases are largely associated with rising LAI due to land use changes and indirect $CO_2$ fertilization effects, as identified by previous studies (Chen et al., 2019; Zhu et al., 2016). Although MODIS, which also does not include a direct $CO_2$ fertilization effect, generally agrees with these increasing trends, it shows a declining GPP in the tropical Amazon and a stronger positive trend in central South America. After incorporating the direct $CO_2$ fertilization effects, both the CFE-Hybrid and CFE-ML setups predict positive trends in tropical forests, an observation absent in all other upscaled datasets. Furthermore, the CFE-Hybrid and CFE-ML models also reveal increasing GPP in temperate and boreal forests of North America and Eurasia. These patterns are also observed in BESS v2 and BEPS,

while PML V2 presents minimal GPP changes in tropics and substantial reduction in Africa. Notably, all datasets agree on a pronounced GPP decrease in eastern Brazil and minimal changes in Australia.

From 2001 to 2018, a positive trend in global annual GPP is uniformly detected by all datasets, albeit with varying magnitudes (Figure 12a, Figure 13a, Figure S16). The ST_Baseline_NT model predicts a GPP growth rate of 0.35 (±0.02) Pg C year$^{-2}$, aligning with FLUXCOM-RS , but lower than FLUXSAT (0.51 Pg C year$^{-2}$) and MODIS (0.39 Pg C year$^{-2}$). The CFE-hybrid models estimate a notably faster GPP growth at 0.58 (±0.03) Pg C year$^{-2}$, similar to BESS V2 and BEPS, both around 0.55 Pg C year$^{-2}$. The CFE-ML models predict the highest trend, up to 0.76 (±0.15) Pg C year$^{-2}$ from the ST_CFE-ML_NT model and 0.59 (±0.13) Pg C year$^{-2}$ from the ST_CFE-ML_DT model. PML V2 displays a neutral trend of 0.08 Pg C year$^{-1}$, and rEC-LU demonstrates an overall decline (0.20 Pg C year$^{-1}$).

The LT_Baseline_NT model identifies increasing GPP trends in large areas of Europe, East and South Asia, as well as the Northern Amazon from 1982 to 2018 (Figure 11). The pattern from the LT_CFE-Hybrid_NT model aligns closely with the LT_Baseline_NT model but exhibit a stronger positive trend in global tropical areas as well as Eurasian boreal forests. Spatial patterns of GPP trends from BESS V2 are consistent with LT_CFE-Hybrid_NT, though with considerably higher magnitudes. FLUXCOM-ERA5 shows overall negative trends in the tropics. rEC-LUE agrees with CEDAR-GPP in positive GPP trends in the extratropical areas, but predicts a pronounced negative trend in the tropics. At the global scale, all the CEDAR-GPP long-term models predict a positive global GPP trend (Figure 12b, 13b). The LT_Baseline_NT and LT_Baseline_DT models show a trend of 0.13 (±0.02) and 0.15 (±0.02) Pg C year$^{-2}$ respectively,  while the LT_CFE-Hybrid_NT and LT_CFE-Hybrid_DT models double these rates with 0.33 (±0.02) and 0.31 (±0.03) Pg C year$^{-2}$ respectively. BESS V2 predicts the highest trend at 0.61 Pg C year$^{-2}$. rEC-LUE shows a two-phased pattern with a strong increase in GPP from 1982 to 2000 (0.54 Pg C year$^{-2}$), followed by a decreasing trend after 2001 (-0.20 Pg C year$^{-2}$) (Figure S17). This results in an overall positive change at a rate comparable to that of the Baseline model. FLUXCOM-ERA5 exhibited a small negative trend.

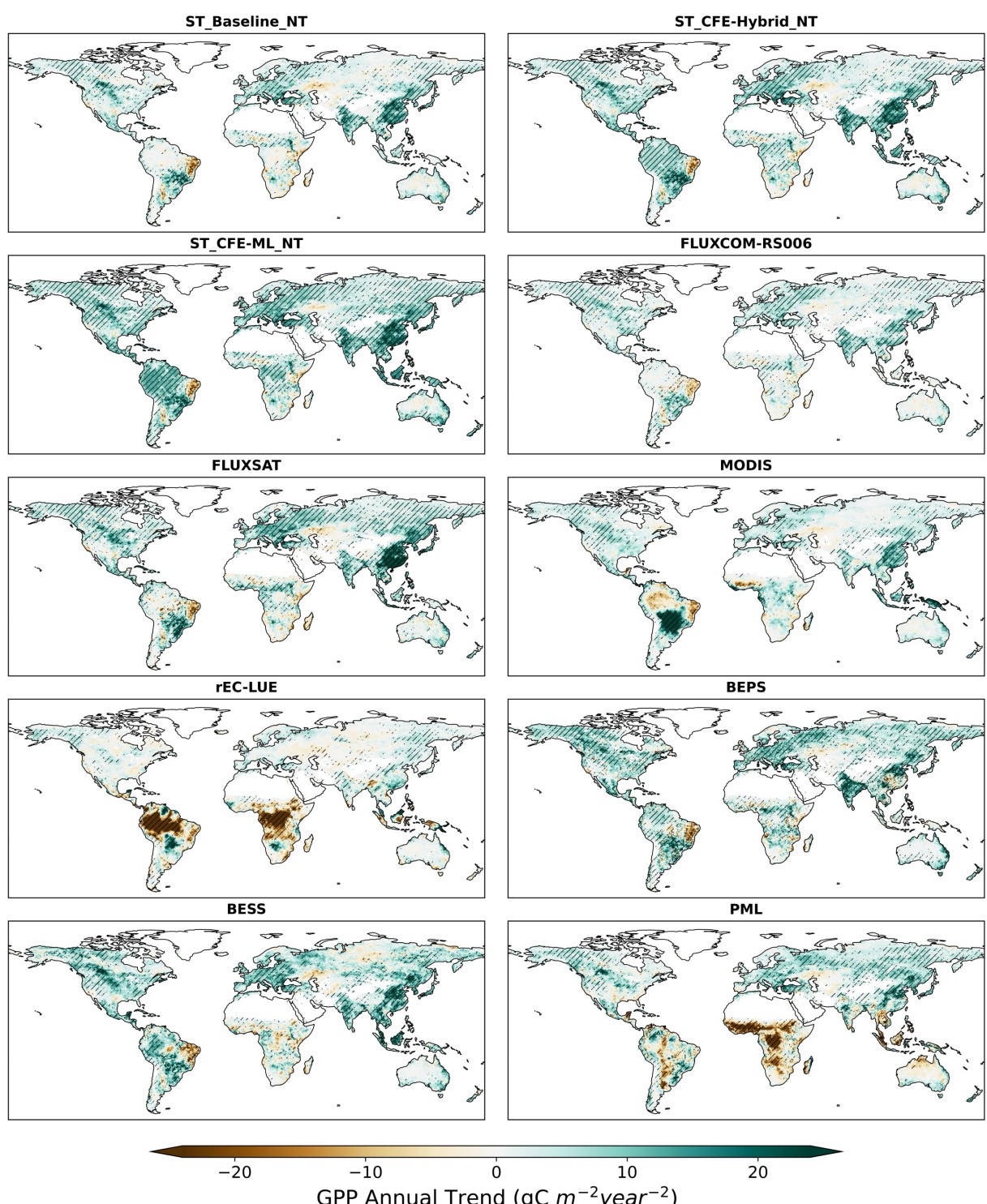

Figure 10. Annual GPP trend over 2001 – 2018 for short-term CEDAR-GPP, FLUXCOM-RS006, FLUXSAT, MODIS, BESS, BEPS, and PML datasets. Hatched areas indicate the GPP trend that is statistically significant at p < 0.05 level under the Mann-Kendall test.

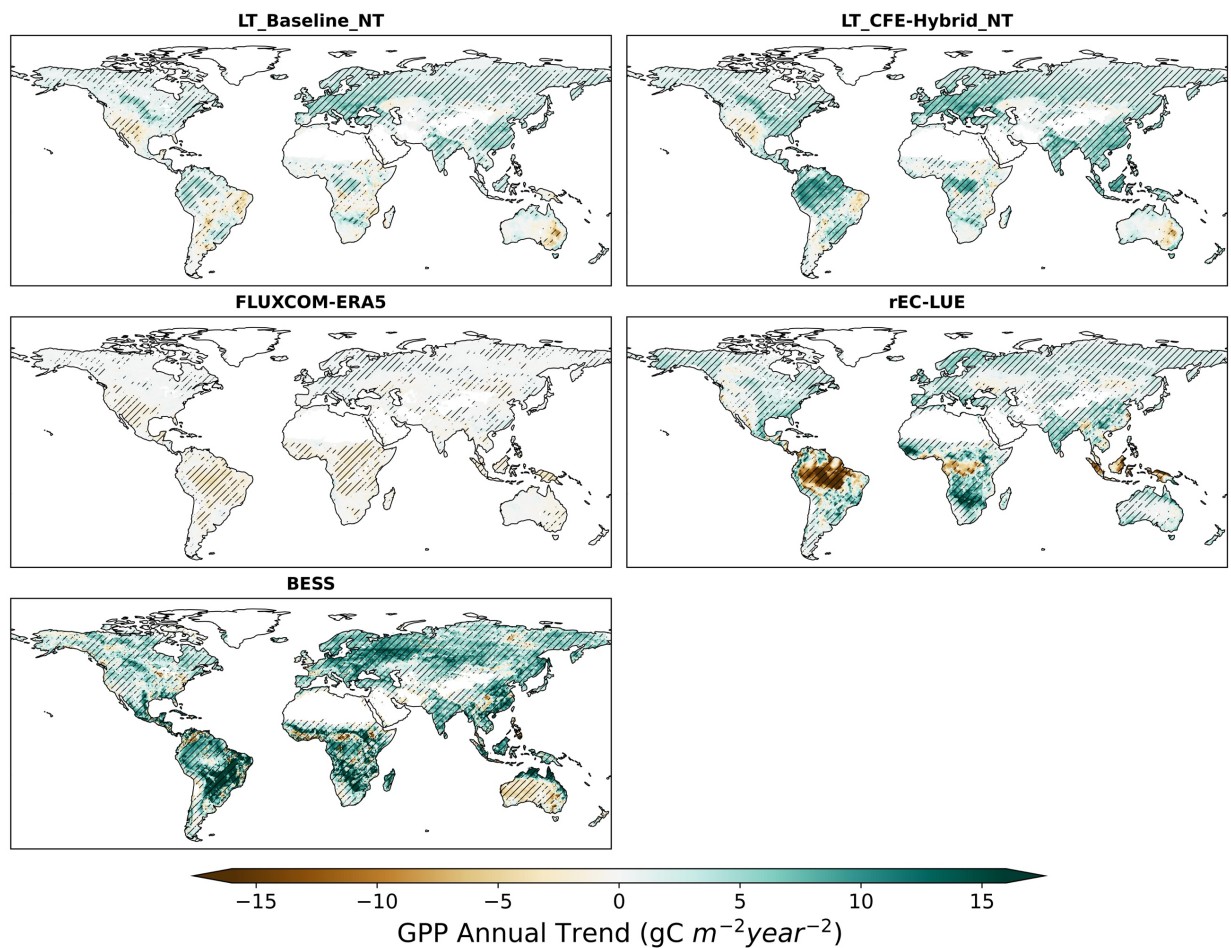


Figure 11. Annual GPP trend over 1982 – 2018 for long-term CEDAR-GPP, rEC-
LUE and BESS datasets. Hatched areas indicate the GPP trend that is statistically
significant at p < 0.05 level under the Mann-Kendall test.



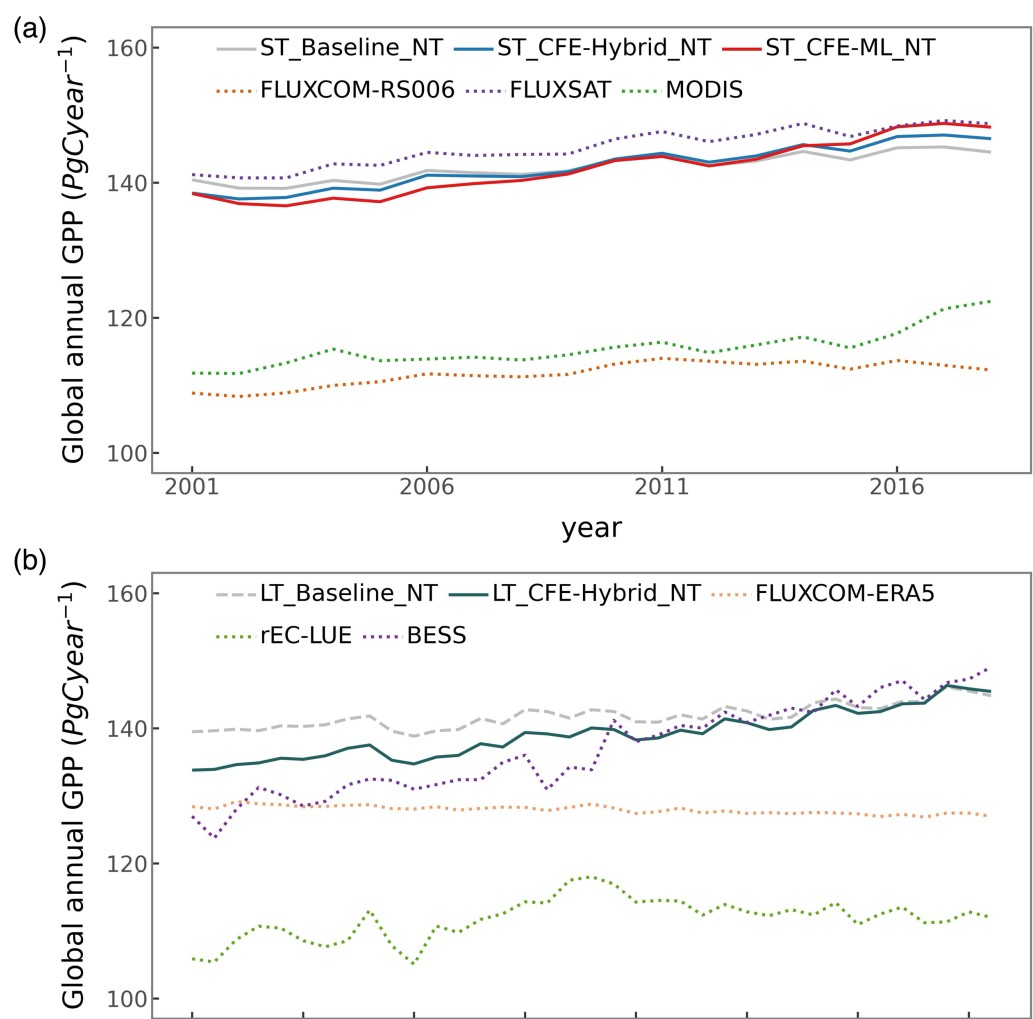


Figure 12. Global annual GPP variations (a) from 2001 to 2018 and (b) from 1982 to
660 2018.

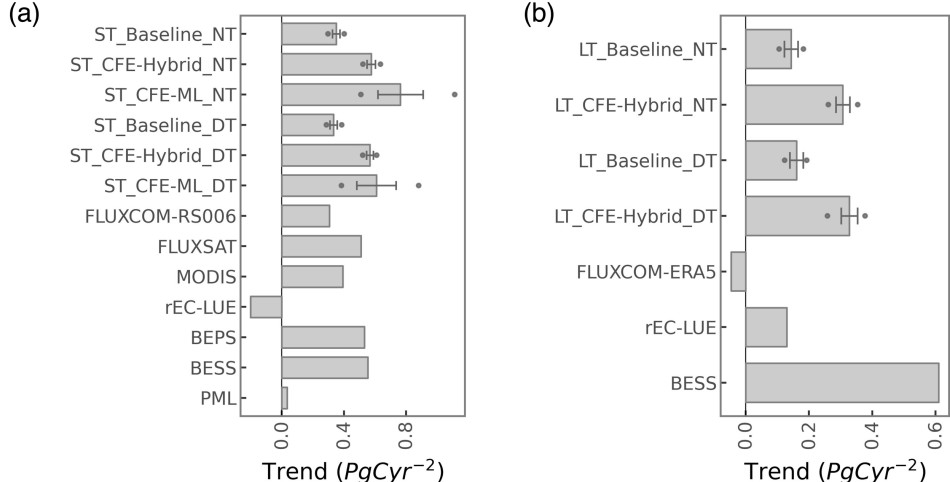


Figure 13. Global annual GPP trends for (a) 2001 to 2018 and (b) 1982 to 2018 time periods. Error bars represent the 25% to 75% percentile from the model ensembles of CEDAR-GPP. Dots indicate the minimum and maximum from the model ensembles of CEDAR-GPP.

## 3.3 GPP estimation uncertainties

We analyzed the spread between the 30 model ensemble members in CEDAR-GPP as an indicator of uncertainties in GPP estimations. The spatial pattern of uncertainty in estimating annual mean GPP largely resembles that of the mean map (Figure 14, Figure 6a). The largest model spread is found in highly productive tropical forests, and this uncertainty decrease in temperate and cold areas (Figure 14a). Tropical ecosystems, with a mean annual GPP between 1000 to 3500 Pg C year[-1], only exhibit a 2% and 6% variation within the model ensemble (Figure 14b). Ecosystems in the temperate and cold climates have a smaller annual GPP and proportionally small uncertainties of up to 6%. However, ecosystems in Arid and Polar climates, despite their similarly low GPP, show higher model uncertainty, reaching 10% to 40% of the ensemble mean.

The estimation uncertainty of GPP trends is generally below 15% to 20% in the CEDAR-GPP datasets under the ST_Baseline and ST_CFE-Hybrid setups (Figure 14c). However, in the ST_CFE-ML setup, the estimation increases substantially, with model spread reaching up to 40% in tropical areas. Figure 15 (Figure S19) further illustrates the trend uncertainties with the ensemble mean error range based on one standard deviation. Both the CFE-ML models show large discrepancies between

the upper and lower uncertainty ranges particularly within the tropics. Additionally, the long-term
models also show a higher uncertainty compared to the short-term models.

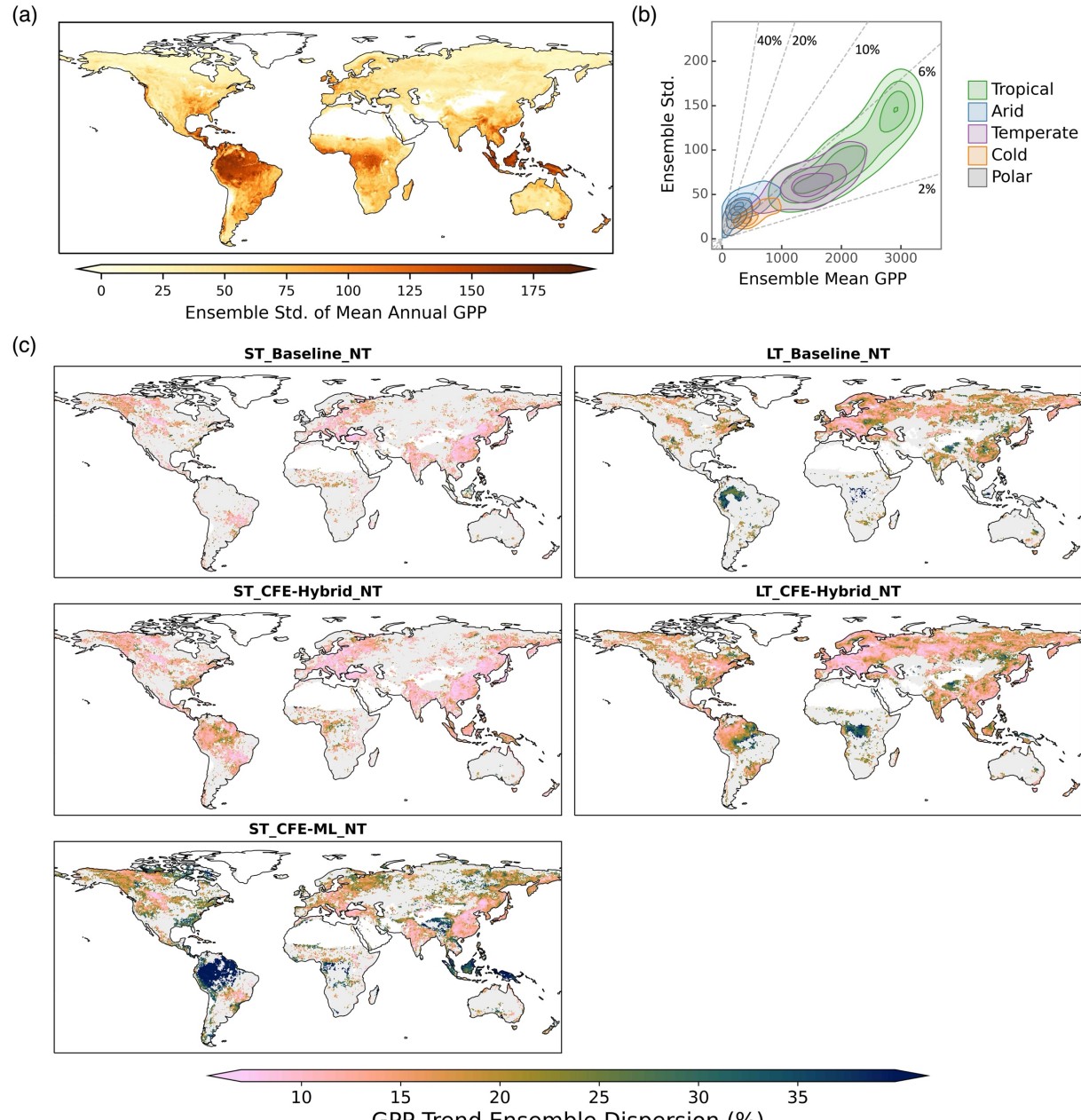


Figure 14. CEDAR-GPP estimation uncertainty derived from ensemble spread
(standard deviation of 30 model predictions). (a) Spatial patterns of the absolute
standard deviation from ensemble members in estimating the mean annual GPP from
2001 to 2018, using data from the ST_CFE-Hybrid_NT setup. (b) Relationships
between ensemble standard deviation and ensemble mean in mean annual GPP.
Colored contours denote clusters of Koppen climate zones. Dashed lines indicate the
ratio between the ensemble standard deviation and the ensemble mean with values
shown in percentage. (c) Spatial patterns of model uncertainty in GPP long-term trend

estimation. Only areas where 90% of the ensemble members showed a statistically
significant trend (p<0.05) are shown in the maps. The trend for the short-term datasets
(left column) was computed between 2001 to 2018. The trend for the long-term
datasets (right column) was computed between 1982 to 2018.

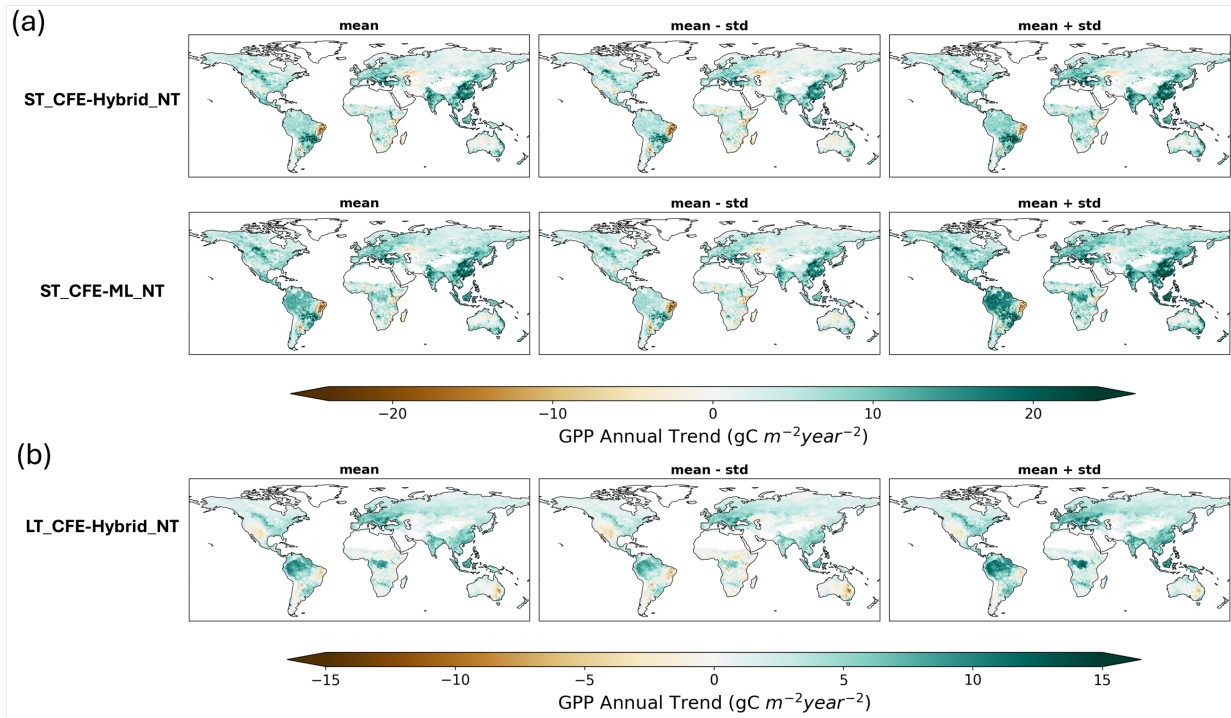

**Figure 15**. Maps of GPP trends and uncertainty range for CEDAR-GPP CFE datasets
(NT only). The first column presents ensemble mean trends, the second column shows
trends from the mean minus one standard deviation (upper, and third column indicates
the trend from the mean plus one standard deviation. (a) Trends from the short-term (ST)
datasets evaluated from 2001 to 2020. (b) Trends from the long-term (LT) dataset
evaluated from 1982 to 2020. DT datasets were shown in Figure S19.


# 4. Discussion


## 4.1 Reducing uncertainties in GPP upscaling


Here we examine the three predominant sources of uncertainties in machine learning upscaling

of GPP: eddy covariance measurements, input datasets, and the machine learning model. We discuss
strategies used in CEDAR-GPP to reduce the impacts of these uncertainties and highlight potential
future research directions.

### 4.1.1 Eddy covariance data

Uncertainties associated with eddy covariance measurement and data processing can propagate through the upscaling process. CEDAR-GPP was produced using monthly aggregated eddy covariance data, where the impact of random errors in half-hourly measurements was minimized due to the temporal aggregation (Jung et al., 2020). Our stringent quality screening further reduced data processing uncertainties such as those associated with gap-filling. Yet, the discrepancy in GPP patterns between the CEDAR-GPP NT and DT setups is indicative of systematic biases linked to the partitioning approaches used to derive GPP from the NEE measurements (Keenan et al., 2019; Pastorello et al., 2020). Interestingly, the mean annual GPP from the DT setup is slightly higher than that from the NT setup (Figure 6), and the DT setup also predicts a higher GPP trend in the long-term dataset (Figure 13). While these discrepancies are relatively small compared to the predominant spatiotemporal patterns, the separate DT and NT setups in CEDAR-GPP offer an interesting quantification of the GPP partitioning uncertainties over space and time, providing insights for future methodology improvements.

The unbalanced spatial representativeness of the eddy covariance data constitutes a more significant source of uncertainty, as highlighted by previous studies (Jung et al., 2020; Tramontana et al., 2015). Effective generalization of machine learning models requires a substantial volume of training data that adequately represents and balances varied conditions. In CEDAR-GPP, this issue was mitigated with a large set of eddy covariance data (~18000 site-months) integrating FLUXNET2015 and two regional networks. However, data availability remains limited in critical carbon exchange hotspots such as tropics, subtropics, drylands, and boreal regions, as well as in mountainous areas (Figure 1). Contrary to widespread perception that sparse training data leads to high upscaling uncertainties, our findings from the bootstrapped model spread indicates modest uncertainties in tropical areas relative to their high GPP magnitude (Figure 14). This observation aligns with findings from the FLUXCOM product, revealing low extrapolation uncertainty in humid tropical regions (Jung et al., 2020). Nevertheless, to fully understand the upscaling uncertainty, it is essential to evaluate the generalization or extrapolation errors within the predictor space and consider the potential limitations of model structures (van der Horst et al., 2019; Villarreal and Vargas, 2021). Additionally, data limitations in mountainous areas and the absence of topology information in the predictor space in our models suggest potential uncertainties related to topographical effects on GPP (Hao et al., 2022; Xie et al., 2023).

Furthermore, our analysis suggests that the estimated global GPP magnitudes are related to
the specific eddy covariance GPP data used in upscaling. Notably, global GPP magnitudes derived
from CEDAR-GPP closely align with those from FLUXSAT, while the estimates from FLUXCOM
were considerably lower (Figure 6, Figure 12). FLUXSAT used eddy covariance data from
FLUXNET2015, which largely overlapped with that included in CEDAR-GPP (Joiner and Yoshida,
2020). FLUXCOM utilized data from FLUXNET La Thuile set and CarboAfrica network, which
consists of a distinct set of sites (Tramontana et al., 2016). The influence from the predictor datasets
is minimal since all three datasets relied on MODIS-derived products. For a more in-depth evaluation
of the impacts of flux site representativeness on upscaling, future research directions could include
conducting synthetic experiments with simulations of ensembles of terrestrial biosphere models.
4.1.2   Input predictors and controlling factors

Upscaled GPP inherent uncertainties from the input predictors, including satellite and climate
datasets. First, satellite remote sensing data contains noises resulting from sun-earth geometry,
atmospheric conditions, soil background, and geolocation inaccuracies. The models or algorithms
used for retrieving LAI, fAPAR, LST, and soil moisture, also contain random errors and systematic
biases specific to certain regions, biome types, or climatic conditions (Fang et al., 2019; Ma et al., 2019;
Yan et al., 2016b). Moreover, satellite observations frequently contain missing values due to clouds,
aerosols, snow, and algorithm failure, leading to both systematic and random uncertainties. In
producing CEDAR-GPP, we mitigated these uncertainties through comprehensive preprocessing
procedures. Our temporal gap-filling strategy exploits both the temporal dependency of vegetation
status and long-term climatology, to reduce biases from missing values. Temporal and spatial
aggregation further reduces the remaining data gaps and random noises. Nevertheless, considerable
uncertainties likely remain in satellite datasets impacting the upscaled estimations.

A potentially more impactful source of uncertainty is the mismatch between the footprint of
the eddy covariance measurements and the coarse resolution of satellite observations. While flux
towers typically have a footprint of around ~1 km$^2$ (Chu et al., 2021), satellite observations employed
in CEDAR-GPP and most other upscaled datasets are at 5 km or lower resolution. Systematic and
random errors could be introduced due to this mismatch, particularly in heterogenous biomes and
areas with a mixture of vegetation and non-vegetated land covers. One mitigation strategy is to
generate upscaled datasets at a higher spatial resolution (e.g. 500m). Alternatively, models could be
trained at a high resolution and applied to the coarse resolution to reduce computation and storage
requirements (Dannenberg et al., 2023; Gaber et al., 2024). However, this approach does not address
inherent scaling errors in coarse-resolution satellite images (Dong et al., 2023; Yan et al., 2016a).
Besides the quality of predictors, successful machine learning upscaling also requires a
comprehensive set of features representing all controlling factors. For example, the lack of GPP
interannual variabilities in FLUXCOM-ERA5 manifests the importance of incorporating dynamic
vegetation signals from remote sensing in the upscaling framework. CEDAR-GPP used satellite
observations from optical, thermal, and microwave systems as well as climate variables thoroughly
representing GPP dynamics. Particularly, the inclusion of LST and soil moisture data provides
important information about resource limitations and stress factors, which are crucial for certain
biomes and/or under specific conditions (Green et al., 2022; Stocker et al., 2018, 2019). Dannenberg
et al. (2023) showed that incorporating LST from MODIS and soil moisture from the SMAP satellite
datasets substantially improved the machine learning estimation accuracy of GPP in North American
drylands. Nevertheless, accurately capturing interannual anomalies remains challenging for certain
biomes, such as evergreen needleleaf forest, cropland, and wetland (Figure 4), as acknowledged by
previous studies (Tramontana et al., 2016; Jung et al., 2020). High prediction uncertainties (Figure 14,
15) in drylands also suggest the machine learning models did not sufficiently represent the mechanisms
of water stress and drought responses. Potential improvement may be achieved by incorporating
datasets related to agricultural management practices (crop type, cultivar, irrigation, fertilization) (Xie
et al., 2021), plant hydraulic and physiological properties (Liu et al., 2021), dynamic C4 plant
distributions (Luo et al., 2024), root and soil characteristics (Stocker et al., 2023), as well as topography
(Xie et al., 2023).
4.1.3   Machine learning models and uncertainty quantification
The choice of machine learning models and their parameterization has been found to have a
relatively minor impact on GPP upscaling uncertainties (Tramontana et al., 2015). CEDAR used the
state-of-the-art boosting algorithm, XGBoost, which provided high performance given the current
data availability. Further reduction of model uncertainty will likely rely on additional information, such
as increasing the number of eddy covariance sites or incorporating more high-quality predictors.
Additionally, temporal dependency of carbon fluxes responses to atmospheric controls may also be
exploited with specialized deep neural networks such as recurrent neural networks or transformers
(Besnard et al., 2019; Ma and Liang, 2022).

A key challenge, however, is the quantification of uncertainties in machine learning upscaling
(Reichstein et al., 2019). The limited availability of eddy covariance data hinders a comprehensive
assessment of the extrapolation errors; consequently, metrics of predictive performance from cross-
validation are inherently biased. CEDAR derived estimation uncertainty for each GPP prediction
using bootstrapping model ensemble, which naturally mimics the sampling bias associated with flux
tower locations. Notably, the choice of input climate reanalysis datasets could also induce systematic
differences in GPP spatial and temporal patterns (Tramontana et al., 2015). As a result, the
FLUXCOM product generates model ensembles based on different reanalysis datasets to capture
these uncertainties. Additionally, different satellite datasets of vegetation structural proxies, such as
LAI, also exhibit significant discrepancies (Jiang et al., 2017). Thus, an ensemble approach combining
site-level bootstrapping with multiple sources of input predictors could potentially provide a more
comprehensive quantification of uncertainties. Furthermore, tree-based models do not generalize well
to unseen conditions, and the uncertainty estimates derived from bootstrapping of XGBoost models
may underrepresent actual biases stemming from limitations in training data representation. Future
work may explore Bayesian neural networks, which provide uncertainty along with predictions and, at
the same time, present high predictive power comparable to ensemble tree-based algorithms (Ma et
al., 2021).

## 820  4.2  Long-term GPP changes and $CO_2$ fertilization effect

CEDAR-GPP was constructed using a comprehensive set of climate variables and multi-source
satellite observations, thus encapsulating long-term GPP dynamics from both direct and indirect
effects of climate controls. Particularly, CEDAR-GPP included the direct $CO_2$ fertilization effect,
which has been shown to dominate the increasing trend of global photosynthesis (Chen et al., 2022).
Incorporating these effects substantially improved long-term trends of GPP from site to global scales
(Figure 5, 10, 11, 12, 13). CEDAR's CFE-Hybrid setup offers a conservative estimation of the direct
$CO_2$ effects by simulating the $CO_2$ sensitivity of light-limited LUE for C3 plants (Walker et al., 2021).
However, the model does not account for the impacts of nutrient availability, which could potentially
constrain $CO_2$ fertilization (Peñuelas et al., 2017; Reich et al., 2014; Terrer et al., 2019). Robust
modeling of LUE responses to rising $CO_2$ under various environmental conditions remains
challenging (Wang et al., 2017). Future work is needed to better understand how these factors affect
the quantification of GPP and its long-term temporal variations.

The CFE-ML model adopted a data-driven approach to infer $CO_2$ effects directly from eddy
covariance data. This strategy allows the model to potentially capture multiple physiological pathways
of the $CO_2$ impact evidenced in the eddy covariance measurements, including the increases of the
biochemical rates and enhancements in the water use efficiency (Keenan et al., 2013). The model
detects a strong positive effect of $CO_2$ on eddy covariance measured GPP, consistent with previous
studies based on process-based and statistical models (Chen et al., 2022; Fernández-Martínez et al.,
2017; Ueyama et al., 2020). Moreover, spatial patterns of GPP trends derived from the CFE-ML model
reflected a strong temperature dependency, aligning with the anticipated temperature sensitivity of
photosynthetic biochemical processes (Keenan et al., 2023). Yet, the considerable ensemble spread in
the $CO_2$ trends from the CFE-ML model and discrepancies between the CFE setups (Figure 14)
underscores a high level of uncertainty in the machine learning quantified $CO_2$ effects.
Several limitations should be noted regarding GPP trend estimation and validation. First, the
CFE-ML model may not fully capture the intricate mechanisms of plant physiological responses to
$CO_2$. For example, eddy covariance towers, especially long-term sites, are typically located in
homogeneous and undisturbed ecosystems, not representative of the full diversity of ecosystems
globally. Thus, interactions between $CO_2$ and natural or human-induced disturbance, as well as many
other stresses, are likely underrepresented in the models. Ultimately, the model's capacity to robustly
quantify $CO_2$ fertilization is constrained by the scope and diversity of the eddy covariance data.
Additionally, the use of spatially invariant $CO_2$ data may not fully represent the actual $CO_2$ variations
that plants experience across different environments.
Secondly, $CO_2$ effects inferred by the CFE-ML models may be confounded by other factors
that correlate with $CO_2$ over time. Industrialization-induced nitrogen deposition could synergistically
boost GPP alongside $CO_2$ (O'Sullivan et al., 2019). Technological and management improvements in
agriculture that contribute to a global enhancement of crop photosynthesis (Zeng et al., 2014), might
also be indirectly reflected in the model estimates. Moreover, interactions with the other input features
that exhibit long-term trends, such as those induced by non-biological factors (e.g. sensor orbital
drifts), also affect the $CO_2$ effects inference. Additionally, other factors that could lead to long-term
GPP trends (e.g. forest aging, disturbances) might also be underrepresented in our models.
Finally, direct validation of GPP trends is limited, particularly in tropical regions, constrained
by the availability of long-term records. Detecting and evaluating trends is challenging and typically
requires long monitoring records (e.g. over 10 to 15 years), since long-term changes, such as those
induced by $CO_2$, are very small relative to large interannual variations. Evaluating aggregated GPP
trends across multiple sites presents an alternative approach; however, there were still insufficient sites
in tropical and evergreen broadleaf forest areas to robustly validate our estimates for those ecosystems
(Figure 5). Partly due to data limitations, uncertainties in GPP estimated from bootstrapped samples
are very high in tropical areas (Figure 14). Thus, trend estimates in these areas should be interpreted
in the context of associated uncertainties and limitations.

Our results also suggested that variations in the estimated GPP long-term trends from different
products are largely related to the representation of $CO_2$ fertilization. Products that do not consider
the direct $CO_2$ effect, including our Baseline models, FLUXSAT, FLUXCOM, and MODIS, show
minimal long-term changes in tropical GPP, while the CEDAR CFE-ML and CFE-Hybrid models
demonstrate significant GPP increases aligning with predictions from the terrestrial biosphere models
(Anav et al., 2015). FLUXCOM-ERA5, not accounting for dynamics changes in vegetation structures
and $CO_2$, does not capture either the direct or indirect $CO_2$ fertilization resulting in a slight negative
GPP trend attributable to shifted climate patterns. Notably, rEC-LUE exhibit contrasting trends
before and after circa 2000, primarily attributed to changes in vapor pressure deficit, PAR, and LAI,
while the direct $CO_2$ fertilization effect remains consistent (Zheng et al., 2020). CEDAR CFE-ML and
CFE-Hybrid models align well with two process-based models forced with remote sensing data which
consider direct $CO_2$ effects (BESS and BEPS). Nevertheless, considerable differences between
CEDAR-GPP and other remote sensing products that include direct $CO_2$ effects (rEC-LUE and PML
V2) warrant more in-depth investigations into long-term GPP responses to changes in atmospheric
$CO_2$ and climate patterns.

Lastly, quantifications of GPP trends and their causes remain highly uncertain from site to
global scales. Trend detection is often complicated by data noises and interannual variabilities, thus
requiring long-term records which are limited in certain areas, biomes, and environmental conditions,
such as tropics, polar regions, wetlands, as well as ecosystems with regular or anthropogenic
disturbances (Baldocchi et al., 2018; Zhan et al., 2022). Moreover, isolating the effect of $CO_2$ is
challenging, as it is confounded by other factors, such as forest regrowth, land cover change, and
disturbances, which also significantly impacts long-term GPP variations. To this end, continued
efforts in expanding ecosystem flux measurements and standardizing data processing present new
opportunities to assess ecosystem productivity responses to changing climate conditions (Delwiche et
al., 2024; Pastorello et al., 2020). Future research could also leverage novel machine learning techniques,
such as knowledge-guided machine learning (Liu et al., 2024) and hybrid modeling that combines
process-based and machine learning approaches (Kraft et al., 2022; Reichstein et al., 2019).

# 5. Data availability and usage note

The CEDAR-GPP product, comprising ten GPP datasets, can be accessed at https://zenodo.org/doi/10.5281/zenodo.8212706 (Kang et al., 2024). These datasets were generated at a spatial resolution of 0.05° and monthly time steps. Each dataset includes an ensemble mean GPP ("GPP_mean") and an ensemble standard deviation ("GPP_std"). Data is formatted in netCDF with the following naming convention: "CEDAR-GPP_<version>_<model setup>_<YYYYMM>.nc".

The CEDAR GPP product offers GPP estimates derived from ten different models. Models are characterized by 1) temporal coverage, 2) configuration of $CO_2$ fertilization, and 3) GPP partitioning approach (Table 2). We provide a structured approach to selecting the most appropriate dataset for research or applications.

1) Study period considerations: the Short-Term (ST) setup is ideal for studies focusing on periods after 2000. These models are constructed using a broader range of explanatory predictors, offering higher precision and smaller random errors. The Long-Term (LT) datasets shall be used for research assessing GPP dynamics over a longer time period (before 2001). It is important to note that trends from the ST and LT datasets are not directly comparable, as they were derived from different satellite remote sensing data.

2) $CO_2$ Fertilization Effect (CFE) configurations: the CFE-Hybrid and CFE-ML setups are preferable when assessing temporal GPP dynamics, especially long-term trends. The CFE-Hybrid setup includes a hypothetical trend from the direct $CO_2$ effect, while CFE-ML is purely data-driven and does not make any specific assumption about the sensitivity of photosynthesis to $CO_2$. Averaging the CFE-Hybrid and CFE-ML estimates is acceptable, with the difference between them reflecting the uncertainty surrounding the direct $CO_2$ effect. Note that the Baseline setup should not be used to study long-term GPP dynamics, especially those induced by elevated $CO_2$. The Baseline setup may be useful to compare with other remote sensing-derived GPP datasets that do not consider the direct $CO_2$ effect. Differences between these setups regarding mean GPP spatial patterns, seasonal and interannual variations are considered to be minor.

3) GPP partitioning methods: We recommend using the mean value derived from both the "NT" (Nighttime) and "DT" (Daytime). The difference between these two provides insight into the uncertainties arising from the partitioning approaches used in GPP estimation from eddy covariance measurements.

Finally, like other upscaled or remote sensing-based GPP datasets, CEDAR-GPP should not be regarded as "observations" but rather as model estimates informed by remote sensing and ground-based data. The extent of assumptions or structural constraints varies across such datasets. CEDAR-GPP, particularly in its CFE-Baseline and CFE-ML configurations, is entirely data-driven and incorporates no explicit assumptions regarding the biological and environmental processes underlying photosynthesis, apart from the generic assumptions inherent in machine learning models. Consequently, the usage and interpretation of this dataset should be carefully framed within the context of the input eddy covariance and environmental data as well as their limitations.

# 6. Code availability

The code for upscaling and generating global GPP datasets can be accessed at https://doi.org/10.5281/zenodo.8400968.

# 7. Conclusions

We present the CEDAR-GPP product generated by upscaling global eddy covariance measurements with machine learning and a broad range of satellite and climate variables. CEDAR-GPP comprises four long-term datasets from 1982 to 2020 and six short-term datasets from 2001 to 2020. These datasets encompass three configurations regarding the incorporation of direct $CO_2$ fertilization effects and two partitioning approaches to derive GPP from eddy covariance data. The machine learning models of CEDAR-GPP demonstrated high capability in predicting monthly GPP, its seasonal cycles, and spatial variability within the global eddy covariance sites, with cross-validated $R^2$ between 0.56 to 0.79. Short-term model setups consistently outperformed long-term models due to considerably more and higher-quality information from multi-source satellite observations.

CEDAR-GPP advances satellite-based GPP estimations, as the first upscaled dataset that considered the direct biochemical effects of elevated atmospheric $CO_2$ on photosynthesis, which is responsible for an increasing land carbon sink over the past decades. We show that incorporating this effect in our CFE-ML and CFE-Hybrid models substantially improved the estimation of GPP trends at eddy covariance sites. Global patterns of long-term GPP trends in the CFE-ML setups show a strong temperature dependency consistent with biophysical theories. However, trend estimation and validation remain particularly challenging in data-scarce regions, such as the tropics, emphasizing the

need for enhanced data availability and methodological advancements. Beyond trends, global spatial
and temporal GPP patterns from CEDAR generally align with other satellite-based GPP datasets.

In conclusion, CEDAR-GPP, informed by global eddy covariance measurements and a broad

range of multi-source remote sensing observations and climatic variables, offers a comprehensive
representation of global GPP spatial and temporal dynamics over the past four decades. The different
$CO_2$ fertilization configurations integrated in CEDAR-GPP offer new opportunities for
understanding global ecosystem photosynthesis's response to increases in atmospheric $CO_2$ along
different pathways over space and time. CEDAR-GPP is expected to serve as a valuable tool for
benchmarking process-based modeling and constraining the global carbon cycle.

 **Appendix A: List of eddy covariance sites**

| Site ID | IGBP | Data Range | Citation |
|---------|------|------------|----------|
| AR-SLu | MF | 2010 - 2011 | (Garcia et al., 2016) |
| AR-Vir | ENF | 2010 - 2012 | (Posse et al., 2016) |
| AT-Neu | GRA | 2002 - 2012 | (Wohlfahrt et al., 2016) |
| AU-Ade | SAV | 2010 - 2014 | (Beringer and Hutley, 2016a) |
| AU-ASM | WSA | 2007 - 2009 | (Cleverly et al., 2016) |
| AU-Cpr | SAV | 2010 - 2014 | (Meyer et al., 2016) |
| AU-Cum | EBF | 2012 - 2014 | (Pendall et al., 2016) |
| AU-DaP | GRA | 2007 - 2013 | (Beringer and Hutley, 2016b) |
| AU-DaS | SAV | 2008 - 2014 | (Beringer and Hutley, 2016g) |
| AU-Dry | SAV | 2008 - 2014 | (Beringer and Hutley, 2016c) |
| AU-Emr | GRA | 2011 - 2013 | (Schroder et al., 2016) |
| AU-Fog | WET | 2006 - 2008 | (Beringer and Hutley, 2016d) |
| AU-Gin | WSA | 2011 - 2014 | (Macfarlane et al., 2016) |
| AU-How | WSA | 2001 - 2014 | (Beringer and Hutley, 2016e) |
| AU-RDF | WSA | 2011 - 2013 | (Beringer and Hutley, 2016f) |
| AU-Rig | GRA | 2011 - 2014 | (Beringer et al., 2016a) |
| AU-Tum | EBF | 2001 - 2014 | (Woodgate et al., 2016) |
| AU-Wac | EBF | 2005 - 2008 | (Beringer et al., 2016b) |
| AU-Whr | EBF | 2011 - 2014 | (Beringer et al., 2016c) |
| AU-Wom | EBF | 2010 - 2014 | (Arndt et al., 2016) |
| AU-Ync | GRA | 2012 - 2014 | (Beringer and Walker, 2016) |
| BE-Bra | MF | 2001 - 2020 | (Warm Winter 2020 Team, 2022) |
| BE-Dor | GRA | 2011 - 2020 | (Warm Winter 2020 Team, 2022) |
| BE-Lon | CRO | 2004 - 2020 | (Warm Winter 2020 Team, 2022) |
| BE-Maa | CSH | 2016 - 2020 | (Warm Winter 2020 Team, 2022) |
| BE-Vie | MF | 2001 - 2020 | (Warm Winter 2020 Team, 2022) |
| BR-Sa1 | EBF | 2002 - 2011 | (Saleska, 2016) |
| BR-Sa3 | EBF | 2001 - 2004 | (Goulden, 2016a) |
| CA-Ca1 | ENF | 2001 - 2002 | (Black, 2023a) |
| CA-Ca2 | ENF | 2001 - 2010 | (Black, 2023b) |
| CA-Ca3 | ENF | 2001 - 2010 | (Black, 2018) |
| CA-Cbo | DBF | 2001 - 2003 | (Staebler, 2022) |
| CA-Gro | MF | 2003 - 2014 | (McCaughey, 2022) |
| CA-Man | ENF | 2001 - 2008 | (Amiro, 2016a) |
| CA-NS1 | ENF | 2002 - 2005 | (Goulden, 2022a) |
| CA-NS2 | ENF | 2001 - 2005 | (Goulden, 2022b) |
| CA-NS3 | ENF | 2001 - 2005 | (Goulden, 2022c) |
| CA-NS4 | ENF | 2002 - 2005 | (Goulden, 2016b) |
| CA-NS5 | ENF | 2001 - 2005 | (Goulden, 2022d) |
| CA-NS6 | OSH | 2001 - 2005 | (Goulden, 2022e) |
| CA-NS7 | OSH | 2002 - 2005 | (Goulden, 2016c) |
| CA-Oas | DBF | 2001 - 2010 | (Black, 2016a) |
| CA-Obs | ENF | 2001 - 2010 | (Black, 2016b) |
| CA-Qc2 | MF | 2008 - 2010 | (Margolis, 2018) |
| CA-Qfo | ENF | 2003 - 2010 | (Margolis, 2023) |
| CA-SF1 | ENF | 2003 - 2006 | (Amiro, 2016b) |
| CA-SF2 | ENF | 2003 - 2005 | (Amiro, 2023) |
| CA-SF3 | OSH | 2003 - 2006 | (Amiro, 2016c) |

| | | | |
|---|---|---|---|
| CA-SJ2 | ENF | 2003 - 2007 | (Barr and Black, 2018) |
| CA-TP1 | ENF | 2003 - 2014 | (Arain, 2016b) |
| CA-TP2 | ENF | 2003 - 2007 | (Arain, 2016c) |
| CA-TP3 | ENF | 2003 - 2014 | (Arain, 2016d) |
| CA-TP4 | ENF | 2003 - 2017 | (Arain, 2016a) |
| CA-TPD | DBF | 2012 - 2014 | (Arain, 2016e) |
| CA-WP1 | WET | 2003 - 2009 | (Flanagan, 2018a) |
| CA-WP2 | WET | 2004 - 2006 | (Flanagan, 2018b) |
| CA-WP3 | WET | 2004 - 2006 | (Flanagan, 2018c) |
| CG-Tch | SAV | 2006 - 2009 | (Nouvellon, 2016) |
| CH-Aws | GRA | 2006 - 2020 | (Warm Winter 2020 Team, 2022) |
| CH-Cha | GRA | 2005 - 2020 | (Warm Winter 2020 Team, 2022) |
| CH-Dav | ENF | 2001 - 2020 | (Warm Winter 2020 Team, 2022) |
| CH-Fru | GRA | 2005 - 2020 | (Warm Winter 2020 Team, 2022) |
| CH-Lae | MF | 2004 - 2020 | (Warm Winter 2020 Team, 2022) |
| CH-Oe1 | GRA | 2002 - 2008 | (Ammann, 2016) |
| CH-Oe2 | CRO | 2004 - 2020 | (Warm Winter 2020 Team, 2022) |
| CN-Cha | MF | 2003 - 2005 | (Zhang and Han, 2016) |
| CN-Cng | GRA | 2007 - 2010 | (Dong, 2016) |
| CN-Din | EBF | 2003 - 2005 | (Zhou and Yan, 2016) |
| CN-Du2 | GRA | 2007 - 2008 | (Chen, 2016c) |
| CN-Ha2 | WET | 2003 - 2005 | (Li, 2016) |
| CN-HaM | GRA | 2002 - 2004 | (Tang et al., 2016) |
| CN-Qia | ENF | 2003 - 2005 | (Wang and Fu, 2016) |
| CN-Sw2 | GRA | 2011 - 2012 | (Shao, 2016) |
| CZ-BK1 | ENF | 2004 - 2020 | (Warm Winter 2020 Team, 2022) |
| CZ-BK2 | GRA | 2006 - 2012 | (Sigut et al., 2016) |
| CZ-KrP | CRO | 2014 - 2020 | (Warm Winter 2020 Team, 2022) |
| CZ-Lnz | DBF | 2015 - 2020 | (Warm Winter 2020 Team, 2022) |
| CZ-RAJ | ENF | 2012 - 2020 | (Warm Winter 2020 Team, 2022) |
| CZ-Stn | DBF | 2010 - 2020 | (Warm Winter 2020 Team, 2022) |
| CZ-wet | WET | 2006 - 2020 | (Warm Winter 2020 Team, 2022) |
| DE-Akm | WET | 2009 - 2020 | (Warm Winter 2020 Team, 2022) |
| DE-Geb | CRO | 2001 - 2020 | (Warm Winter 2020 Team, 2022) |
| DE-Gri | GRA | 2004 - 2020 | (Warm Winter 2020 Team, 2022) |
| DE-Hai | DBF | 2001 - 2020 | (Warm Winter 2020 Team, 2022) |
| DE-HoH | DBF | 2015 - 2020 | (Warm Winter 2020 Team, 2022) |
| DE-Hte | WET | 2009 - 2018 | (Drought 2018 Team, 2020) |
| DE-Hzd | DBF | 2010 - 2020 | (Warm Winter 2020 Team, 2022) |
| DE-Kli | CRO | 2004 - 2020 | (Warm Winter 2020 Team, 2022) |
| DE-Lkb | ENF | 2009 - 2013 | (Lindauer et al., 2016) |
| DE-Lnf | DBF | 2002 - 2012 | (Knohl et al., 2016) |
| DE-Obe | ENF | 2008 - 2020 | (Warm Winter 2020 Team, 2022) |
| DE-RuR | GRA | 2011 - 2020 | (Warm Winter 2020 Team, 2022) |
| DE-RuS | CRO | 2011 - 2020 | (Warm Winter 2020 Team, 2022) |
| DE-RuW | ENF | 2012 - 2020 | (Warm Winter 2020 Team, 2022) |
| DE-Seh | CRO | 2007 - 2010 | (Schneider and Schmidt, 2016) |
| DE-SfN | WET | 2012 - 2014 | (Klatt et al., 2016) |
| DE-Spw | WET | 2010 - 2014 | (Bernhofer et al., 2016) |
| DE-Tha | ENF | 2001 - 2020 | (Warm Winter 2020 Team, 2022) |
| DK-Eng | GRA | 2005 - 2007 | (Pilegaard and Ibrom, 2016) |
| DK-Sor | DBF | 2001 - 2020 | (Warm Winter 2020 Team, 2022) |

| | | | |
|---|---|---|---|
| ES-Abr | WSA | 2015 - 2020 | (Warm Winter 2020 Team, 2022) |
| ES-Agu | OSH | 2006 - 2019 | (Warm Winter 2020 Team, 2022) |
| ES-Amo | OSH | 2007 - 2012 | (Poveda et al., 2016) |
| ES-LgS | OSH | 2005 - 2020 | (Reverter et al., 2016) |
| ES-LJu | WSA | 2014 - 2020 | (Warm Winter 2020 Team, 2022) |
| ES-LM1 | WSA | 2014 - 2020 | (Warm Winter 2020 Team, 2022) |
| ES-LM2 | OSH | 2007 - 2009 | (Warm Winter 2020 Team, 2022) |
| FI-Hyy | ENF | 2001 - 2020 | (Warm Winter 2020 Team, 2022) |
| FI-Jok | CRO | 2001 - 2003 | (Lohila et al., 2016) |
| FI-Ken | ENF | 2018 - 2020 | (Warm Winter 2020 Team, 2022) |
| FI-Let | ENF | 2009 - 2020 | (Warm Winter 2020 Team, 2022) |
| FI-Lom | WET | 2007 - 2009 | (Aurela et al., 2016a) |
| FI-Qvd | CRO | 2018 - 2020 | (Warm Winter 2020 Team, 2022) |
| FI-Sii | GRA | 2016 - 2020 | (Warm Winter 2020 Team, 2022) |
| FI-Sod | ENF | 2001 - 2014 | (Aurela et al., 2016b) |
| FI-Var | ENF | 2016 - 2020 | (Warm Winter 2020 Team, 2022) |
| FR-Aur | CRO | 2005 - 2020 | (Warm Winter 2020 Team, 2022) |
| FR-Bil | ENF | 2014 - 2020 | (Warm Winter 2020 Team, 2022) |
| FR-FBn | MF | 2008 - 2020 | (Warm Winter 2020 Team, 2022) |
| FR-Fon | DBF | 2005 - 2020 | (Warm Winter 2020 Team, 2022) |
| FR-Gri | CRO | 2004 - 2020 | (Warm Winter 2020 Team, 2022) |
| FR-Hes | DBF | 2014 - 2020 | (Warm Winter 2020 Team, 2022) |
| FR-Lam | ENF | 2001 - 2008 | (Warm Winter 2020 Team, 2022) |
| FR-LBr | WET | 2017 - 2020 | (Berbigier et al., 2016) |
| FR-LGt | CRO | 2005 - 2020 | (Warm Winter 2020 Team, 2022) |
| FR-Pue | EBF | 2001 - 2014 | (Ourcival et al., 2016) |
| FR-Tou | GRA | 2018 - 2020 | (Warm Winter 2020 Team, 2022) |
| GF-Guy | EBF | 2015 - 2015 | (Warm Winter 2020 Team, 2022) |
| GH-Ank | EBF | 2011 - 2014 | (Valentini et al., 2016a) |
| GL-NuF | WET | 2008 - 2014 | (Hansen, 2016) |
| GL-ZaF | WET | 2009 - 2011 | (Lund et al., 2016a) |
| GL-ZaH | GRA | 2001 - 2014 | (Lund et al., 2016b) |
| IL-Yat | ENF | 2001 - 2020 | (Warm Winter 2020 Team, 2022) |
| IT-CA1 | DBF | 2011 - 2014 | (Sabbatini et al., 2016a) |
| IT-CA2 | CRO | 2011 - 2014 | (Sabbatini et al., 2016b) |
| IT-CA3 | DBF | 2011 - 2014 | (Sabbatini et al., 2016c) |
| IT-Col | DBF | 2001 - 2014 | (Matteucci, 2016) |
| IT-Cp2 | EBF | 2012 - 2020 | (Warm Winter 2020 Team, 2022) |
| IT-Cpz | EBF | 2001 - 2008 | (Valentini et al., 2016b) |
| IT-La2 | ENF | 2001 - 2002 | (Cescatti et al., 2016) |
| IT-Lav | ENF | 2003 - 2020 | (Warm Winter 2020 Team, 2022) |
| IT-Lsn | OSH | 2016 - 2020 | (Warm Winter 2020 Team, 2022) |
| IT-MBo | GRA | 2003 - 2020 | (Warm Winter 2020 Team, 2022) |
| IT-Noe | CSH | 2004 - 2014 | (Spano et al., 2016) |
| IT-PT1 | DBF | 2002 - 2004 | (Manca and Goded, 2016) |
| IT-Ren | ENF | 2001 - 2020 | (Warm Winter 2020 Team, 2022) |
| IT-Ro1 | DBF | 2001 - 2008 | (Valentini et al., 2016c) |
| IT-Ro2 | DBF | 2002 - 2012 | (Papale et al., 2016) |
| IT-SR2 | ENF | 2013 - 2020 | (Warm Winter 2020 Team, 2022) |
| IT-SRo | ENF | 2001 - 2012 | (Gruening et al., 2016) |
| IT-Tor | GRA | 2008 - 2020 | (Warm Winter 2020 Team, 2022) |
| JP-MBF | DBF | 2004 - 2005 | (Kotani, 2016a) |

| | | | |
|---|---|---|---|
| JP-SMF | MF | 2002 - 2006 | (Kotani, 2016b) |
| MY-PSO | EBF | 2003 - 2009 | (Kosugi and Takanashi, 2016) |
| NL-Hor | GRA | 2004 - 2011 | (Dolman et al., 2016a) |
| NL-Loo | ENF | 2001 - 2018 | (Drought 2018 Team, 2020) |
| PA-SPn | DBF | 2007 - 2009 | (Wolf et al., 2016) |
| RU-Che | WET | 2002 - 2005 | (Merbold et al., 2016) |
| RU-Cok | OSH | 2003 - 2013 | (Dolman et al., 2016b) |
| RU-Fy2 | ENF | 2015 - 2020 | (Warm Winter 2020 Team, 2022) |
| RU-Fyo | ENF | 2001 - 2020 | (Warm Winter 2020 Team, 2022) |
| RU-Ha1 | GRA | 2002 - 2004 | (Belelli et al., 2016) |
| SD-Dem | SAV | 2007 - 2009 | (Ardö et al., 2016) |
| SE-Deg | WET | 2001 - 2020 | (Warm Winter 2020 Team, 2022) |
| SE-Htm | ENF | 2015 - 2020 | (Warm Winter 2020 Team, 2022) |
| SE-Lnn | CRO | 2014 - 2018 | (Drought 2018 Team, 2020) |
| SE-Nor | ENF | 2014 - 2020 | (Warm Winter 2020 Team, 2022) |
| SE-Ros | ENF | 2014 - 2020 | (Warm Winter 2020 Team, 2022) |
| SE-Svb | ENF | 2014 - 2020 | (Warm Winter 2020 Team, 2022) |
| SJ-Adv | WET | 2013 - 2014 | (Christensen, 2016) |
| SN-Dhr | SAV | 2010 - 2013 | (Tagesson et al., 2016) |
| US-ARM | CRO | 2004 - 2018 | (Biraud et al., 2022) |
| US-Atq | WET | 2003 - 2008 | (Zona and Oechel, 2016a) |
| US-Bar | DBF | 2005 - 2017 | (Richardson and Hollinger, 2023) |
| US-Blo | ENF | 2001 - 2007 | (Goldstein, 2016) |
| US-Cop | CRO | 2011 - 2013 | (Bowling, 2016) |
| US-CRT | GRA | 2001 - 2007 | (Chen and Chu, 2023) |
| US-Dk1 | GRA | 2004 - 2008 | (Oishi et al., 2016a) |
| US-Dk2 | DBF | 2004 - 2008 | (Oishi et al., 2016b) |
| US-Dk3 | ENF | 2004 - 2008 | (Oishi et al., 2016c) |
| US-Fmf | WSA | 2005 - 2008 | (Dore and Kolb, 2023a) |
| US-FR2 | ENF | 2005 - 2010 | (Litvak, 2016) |
| US-Fuf | ENF | 2005 - 2010 | (Dore and Kolb, 2023b) |
| US-GBT | ENF | 2001 - 2003 | (Massman, 2016a) |
| US-GLE | ENF | 2005 - 2014 | (Massman, 2016b) |
| US-Goo | GRA | 2002 - 2006 | (Meyers, 2016) |
| US-Ha1 | DBF | 2001 - 2012 | (Munger, 2016) |
| US-Ho1 | ENF | 2012 - 2018 | (Hollinger, 2016) |
| US-Ivo | WET | 2004 - 2007 | (Zona and Oechel, 2016b) |
| US-KFS | GRA | 2009 - 2017 | (Brunsell, 2022) |
| US-KS2 | CSH | 2003 - 2006 | (Drake and Hinkle, 2016) |
| US-Los | WET | 2001 - 2014 | (Desai, 2016a) |
| US-Me2 | DBF | 2001 - 2017 | (Law, 2022) |
| US-Me3 | ENF | 2003 - 2017 | (Law, 2016a) |
| US-Me5 | ENF | 2004 - 2009 | (Law, 2016b) |
| US-Me6 | ENF | 2001 - 2002 | (Law, 2016c) |
| US-MMS | ENF | 2010 - 2014 | (Novick and Phillips, 2022) |
| US-Mpj | OSH | 2008 - 2017 | (Litvak, 2021) |
| US-Myb | WET | 2011 - 2014 | (Sturtevant et al., 2016) |
| US-Ne1 | ENF | 2001 - 2014 | (Suyker, 2016a) |
| US-Ne2 | CRO | 2001 - 2013 | (Suyker, 2016b) |
| US-Ne3 | CRO | 2001 - 2013 | (Suyker, 2016c) |
| US-NR1 | CRO | 2001 - 2013 | (Blanken et al., 2016) |
| US-Oho | DBF | 2004 - 2013 | (Chen et al., 2023) |

| | | | |
|---|---|---|---|
| US-PFa | MF | 2001 - 2014 | (Desai, 2016b) |
| US-Prr | ENF | 2010 - 2016 | (Iwahana et al., 2016) |
| US-Rls | CSH | 2014 - 2017 | (Flerchinger, 2023) |
| US-Rms | CSH | 2014 - 2017 | (Flerchinger, 2022a) |
| US-Ro1 | CRO | 2004 - 2016 | (Baker et al., 2022) |
| US-Rws | OSH | 2014 - 2017 | (Flerchinger, 2022b) |
| US-Seg | MF | 2008 - 2014 | (Litvak, 2023a) |
| US-Ses | WSA | 2004 - 2014 | (Litvak, 2023b) |
| US-SRC | GRA | 2007 - 2017 | (Kurc, 2016) |
| US-SRM | OSH | 2007 - 2017 | (Scott, 2016a) |
| US-Sta | OSH | 2005 - 2009 | (Ewers and Pendall, 2016) |
| US-Syv | MF | 2001 - 2014 | (Desai, 2016c) |
| US-Ton | WSA | 2001 - 2014 | (Baldocchi and Ma, 2016) |
| US-Tw1 | WET | 2011 - 2017 | (Valach et al., 2021) |
| US-Tw4 | WET | 2014 - 2017 | (Eichelmann et al., 2023) |
| US-Twt | CRO | 2009 - 2014 | (Baldocchi, 2016) |
| US-Uaf | DBF | 2007 - 2017 | (Ueyama et al., 2018) |
| US-UMB | DBF | 2008 - 2017 | (Gough et al., 2023) |
| US-UMd | ENF | 2003 - 2017 | (Gough et al., 2022) |
| US-Var | GRA | 2001 - 2014 | (Baldocchi et al., 2016) |
| US-Vcm | ENF | 2008 - 2017 | (Litvak, 2023c) |
| US-Vcp | ENF | 2007 - 2017 | (Litvak, 2023d) |
| US-WCr | DBF | 2001 - 2014 | (Desai, 2016d) |
| US-Whs | WET | 2011 - 2013 | (Scott, 2016b) |
| US-Wi3 | OSH | 2007 - 2014 | (Chen, 2016a) |
| US-Wi4 | DBF | 2002 - 2004 | (Chen, 2016b) |
| US-Wjs | ENF | 2002 - 2005 | (Litvak, 2022) |
| US-WPT | SAV | 2007 - 2017 | (Chen and Chu, 2016) |
| ZM-Mon | DBF | 2007 - 2009 | (Kutsch et al., 2016) |


**Appendix B: CO₂ sensitivity function of Light Use Efficiency**

In the CFE-Hybrid model, the direct $CO_2$ fertilization effect was prescribed onto machine learning estimated GPP at a reference $CO_2$ level using a theoretical $CO_2$ sensitivity function of LUE. The sensitivity function, which describes the fractional change in LUE due to $CO_2$ relative to the reference period, is described below.

The Light Use Efficiency (LUE) model (Monteith, 1972) of GPP states that,

$$GPP = APAR \times LUE = PAR \times fAPAR \times LUE \tag{A1}$$

where $PAR$ is the photosynthetic active radiation, $fAPAR$ is the fraction of $PAR$ that plant canopy has absorbed, and $APAR$ is the absorbed $PAR$. Eco-evolutionary theory, specifically the optimal coordination hypothesis, predicts that the electron-transport-limited (light-limited) ($A_j$) and Rubisco-limited ($A_c$) rates of photosynthesis converge on the time scale of physiological acclimation, which is in the order of a few weeks (Harrison et al., 2021; Haxeltine and Prentice, 1996; Wang et al., 2017). Thus, at a monthly time scale, we assume that

$$A = A_c = A_j \tag{A2}$$

where $A$ is the gross photosynthetic rate, here equivalent to GPP.

In the following, we derive our sensitivity function based on $A_j$, which has a smaller response to $CO_2$ than $A_c$ , thus providing conservative estimates of the direct $CO_2$ fertilization effect (Walker et al., 2021). According to the Fauquhar, von Caemmerer and Berry (FvCB) model (Farquhar et al., 1980),

$$A_j = \varphi_0 I \frac{c_i - \Gamma^*}{c_i + 2\Gamma^*} \tag{A3}$$

where $\varphi_0$ is the intrinsic quantum efficiency of photosynthesis, $I$ is the absorbed PAR ($I = APAR$), $c_i$ is the leaf-internal partial pressure of $CO_2$, and $\Gamma^*$ is the photorespiratory compensation point that depends on temperature:

$$\Gamma^* = r_{25} e^{\frac{\Delta H(T - 298.15)}{298.15 RT}} \tag{A4}$$

where $r_{25} = 4.22 \, Pa$ is the photorespiratory point at 25 °C, $\Delta H$ is the activation energy ($37.83 \cdot 10^3$ J mol⁻¹), $T$ is the air temperature in Kelvin, and $R$ is the molar gas constant (8.314 J mol⁻¹ K⁻¹). We denote atmospheric $CO_2$ concentration as $c_a$, and $\chi$ is the ratio of leaf internal and external $CO_2$, so

$$c_i = \chi c_a \tag{A5}$$

Combing (A1), (A3), (A5), and assuming (A2), LUE can be written as,

$$LUE = \varphi_0 \frac{c_i - \Gamma^*}{c_i + 2\Gamma^*} = \varphi_0 \frac{\chi c_a - \Gamma^*}{\chi c_a + 2\Gamma^*} \qquad (A6)$$

We can therefore show that under constant absorbed light ($I$ or $APAR$), the sensitivity of GPP to $CO_2$
is proportional to that of LUE,

$$\frac{\partial GPP}{\partial c_a} = \frac{\partial \varphi_0 I \frac{\chi c_a - \Gamma^*}{\chi c_a + 2\Gamma^*}}{\partial c_a} = I \frac{\partial LUE}{\partial c_a} \qquad (A7)$$

Thus from (A7), we can express the actual GPP at the time $t$ and a $CO_2$ level $c_a^t$ as the product of a
reference GPP with a $CO_2$ level $c_a^0$ and the ratio between actual and reference LUE (A8-9). We denote
the actual GPP as time t as $GPP^t_{c_a = c_a^t}$, and the reference GPP at time t as $GPP^t_{c_a = c_a^0}$.

$$\frac{GPP^t_{c_a = c_a^t}}{GPP^t_{c_a = c_a^0}} = \frac{LUE^t_{c_a = c_a^t}}{LUE^t_{c_a = c_a^0}} = \frac{\frac{\chi c_a^t - \Gamma^*}{\chi c_a^t + 2\Gamma^*}}{\frac{\chi c_a^0 - \Gamma^*}{\chi c_a^0 + 2\Gamma^*}} = \frac{\phi^t_{CO2}}{\phi^{t0}_{CO2}} \qquad (A8)$$

$$GPP^t_{c_a = c_a^t} = GPP^t_{c_a = c_a^0} \times \frac{\phi^t_{CO2}}{\phi^{t0}_{CO2}} \qquad (A9)$$

The reference GPP represents the GPP value at time $t$ if the $CO_2$ were at the level of a reference
level, while all other factors, such as $PAR$, $fAPAR$, temperature, and other environmental controls
remain unchanged. Here the $CO_2$ impacts on LUE depend on atmospheric $CO_2$ ($c_a$), $\chi$ , and air
temperature. We fixed $\chi$ to the global long-term average value 0.7 typical to C3 plants (Prentice et al.,
2014; Wang et al., 2017). We further tested a dynamic model that quantified $\chi$ as a function of air
temperature and vapor pressure deficit following an eco-evolutionary theory across global flux sites
(Keenan et al., 2023). The estimated $\chi$ had a mean and median of 0.7 and a standard deviation of 0.04
(Figure S20a). Differences in the direct $CO_2$ effect between the dynamic and fixed $\chi$ approaches were
minimal, with an $R^2$ of 0.99 and a slope of 0.99 from a least squares linear regression line (Figure S20b).
GPP trends across flux towers were also highly consistent between the two approaches, with a
difference less than 0.1 gC m$^{-2}$ year$^{-2}$ (Figure S20b, c). Since these results indicated that $\chi$ is relatively
stable, we used the fixed $\chi$ approach to produce the CEDAR-GPP dataset.
In the CFE-Hybrid model, we estimated the reference GPP by fixing the $CO_2$ at the level of the
year 2001 while keeping all other variables dynamic in the CFE-ML model. Then the actual GPP can
be estimated following (A9). Fixing $CO_2$ values to the 2001 level, the start year of eddy covariance
data used in model training, essentially removed the effects of $CO_2$ inferred by the CFE-ML model.

**Supplement**

The supplement related to this article is available online.

**Author contributions**

T. K. and Y. K. conceptualized the study. Y. K. performed the formal analysis and generated the final product. Y. K., T. K., M. B., and M. G. contributed to the development and investigation of the research. Y. K., M. G., and X. L. contributed to data curation and processing. Y. K. prepared the manuscript with contributions from all co-authors. T. K. supervised the project.

**Competing interests**

The authors declare that they have no conflict of interest.

**Acknowledgments**

We are grateful to Dr. Youngryel Ryu for providing the BESS_Rad dataset and Dr. Martin Jung for sharing the FLUXCOM-RS006 dataset. We also thank Dr. Muyi Li, Dr. Zaichun Zhu, and Dr. Sen Cao for sharing early versions of the PKU GIMMS NDVI4g and LAI4g datasets with us. We extend our gratitude to four reviewers for their constructive feedback for improving this paper.

**Financial support**

This research was supported by the U.S. Department of Energy Office of Science Early Career Research Program award #DE-SC0021023 and a NASA Award 80NSSC21K1705. YK acknowledges additional support from a NASA award 80NSSC24K1562. TFK acknowledges further support from the LEMONTREE (Land Ecosystem Models based On New Theory, obseRvations and ExperimEnts) project, funded through the generosity of Eric and Wendy Schmidt by recommendation of the Schmidt Futures programme, support from the RUBISCO SFA, which is sponsored by the Regional and Global Model Analysis (RGMA) Program in the Climate and Environmental Sciences Division (CESD) of the Office of Biological and Environmental Research (BER) in the U.S. Department of Energy Office of Science, and NASA awards 80NSSC20K1801 and 80NSSC25K7327. MB acknowledges additional support from the U.S. Department of Agriculture NIFA award #2023-67012-40086.

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
