# Peer review of "CEDAR-GPP: spatiotemporally upscaled estimates of gross primary productivity incorporating CO2 fertilization"

_Earth System Science Data, 2023_

## Author Comment (AC1)

**Response Letter to Reviewer #1**

**CEDAR-GPP: spatiotemporally upscaled estimates of gross primary productivity incorporating $CO_2$ fertilization**

Dear reviewer,

Thank you very much for your thorough review of our manuscript. We greatly appreciate your constructive feedback and comments, which have significantly improved our paper. Below, we outline our responses (in blue) to your comments (in black) and detail the corresponding revisions made to the manuscript (in red). We included line numbers from the "track-change" version of the revised manuscript for your reference.

Thank you once again for the time and efforts you dedicated to reviewing our manuscript. We hope that our revisions have adequately addressed your comments and concerns. We eagerly look forward to any further feedback and suggestions you may have!

Sincerely,

Yanghui Kang
On behalf of all co-authors

Kang et al. developed a new machine learning based global GPP dataset (CEDAR-GPP), which incorporated CO2 fertilization effect (CFE). Since the CFE significantly affected the trend of global GPP, the new dataset can answer the question of how the CFE benefits GPP over the last 40 years. This dataset overcomes the previous dataset without consider the direct CFE to GPP (i.e. FLUXCOM, FLUXSAT), so it could have the potential for evaluating the GPP increasing trend caused by the CFE. However, I found serval main issues in the current manuscript and dataset.

Main point#1 Since there are ten different setups for CEDAR-GPP, so which one is best among them? Or can the authors provide a guidance for the readers use this product to address the specific question? This is a particularly important point for the dataset availability. As the results mostly showing the NT data, why not move all the DT results to the SI?

**Response**: Thanks for the suggestion! We agree that guidance on dataset selection would be greatly helpful. We have incorporated detailed guidelines on dataset selection in the "Data availability and usage" section. We also added the details in the user guide accompanying the dataset in Zenodo. Furthermore, we moved the DT results in Section 3.1 – 3.3.to SI for clarity and added a few more displays in SI to illustrate the performance of DT models.

**Revisions:**
Line 845 – 868 (Section 5): *The CEDAR GPP product offers GPP estimates derived from ten different models. Models are characterized by 1) temporal coverage, 2) configuration of CO2 fertilization, and 3) GPP partitioning approach (Table 2). We provide a structured approach to selecting the most appropriate dataset for research or applications.*

*1) Study period considerations: the Short-Term (ST) setup is ideal for studies focusing on periods after 2000. These models are constructed using a broader range of explanatory predictors, offering higher precision and smaller random errors. The Long-Term (LT) datasets shall be used for research assessing GPP dynamics over a longer time period (before 2001). It is important to note that trends from the ST and LT datasets are not directly comparable, as they were derived from different satellite remote sensing data.*

*2) CO2 Fertilization Effect (CFE) configurations: the CFE-Hybrid and CFE-ML setups are preferable when assessing temporal GPP dynamics, especially long-term trends. The CFE-Hybrid setup includes a hypothetical trend for the direct CO2 effect, while CFE-ML is purely data-driven and does not make any specific assumption about the sensitivity of photosynthesis to CO2. Averaging the CFE-Hybrid and CFE-ML estimates is acceptable, with the difference between them reflecting the uncertainty surrounding the direct CO2 effect. Note that the Baseline setup should not be used to study long-term GPP dynamics, especially those induced by elevated CO2. The Baseline setup may be useful to compare with other remote sensing-derived GPP datasets that do not consider the direct CO2 effect. Differences between these setups regarding mean GPP spatial patterns, seasonal and interannual variations are minor.*

*3) GPP partitioning methods: We recommend using the mean value derived from both the "NT" (Nighttime) and "DT" (Daytime). The difference between these two provides insight into the uncertainties arising from the partitioning approaches used in GPP estimation from eddy covariance measurements.*

*Figure 3 was revised to include NT results only.*
*Figure 5 was revised to include NT results only and with an additional display of the validation of estimated trend across different biomes and climate zones.*
*Figure 6: Remove the DT model results from the display of latitudinal distributions.*
*Figure S1: newly added to demonstrate DT model performance in predicting monthly GPP and its spatiotemporal dynamics.*
*Figure S2: newly added to display DT model performance by biomes and climate zones.*
*Figure S3: newly added to illustrate DT model performance in predicting long-term trends of GPP across eddy covariance sites.*
*Figure S5: Added a panel showing the latitudinal profiles of ten CEDAR-GPP datasets*

Main point#2 The authors claimed that they consider the direct CFE to photosynthesis. If my understanding is right, the direct CFE benefits the GPP over the past 4 decades and the CFE-ML based GPP (consider the direct CFE) will have a higher increasing trend compared to the Baseline based GPP. The baseline GPP did not contain the direct CFE, but it has the trend of VI, which can represent for the in-direct CFE trend. The reason I have this question is, as the machine learning model is black box, the trend is based on statistic relationships, so the output for the direct CFE or indirect CFE may just depend on the model training. This means that the Baseline GPP could also capture the indirect CFE to GPP and the indirect CFE can reflect the GPP trend in the real world. Without the site validation, the GPP trend from CEDAR-GPP is hard to distinguish whether they are right or not.

**Response**: Thank you for your question. Our study distinguishes between the direct CFE, which influences photosynthetic rate or light use efficiency, and the indirect CFE, which is related to leaf area expansion and enhanced light interception. Since these are separate effects, we note that both need to be considered to fully represent GPP's long-term changes in response to increased CO2.

The effectiveness of a machine learning model in capturing functional relationships depends on the inclusion of relevant controlling factors during training. To this end, our CFE-ML

model incorporates atmospheric CO2 levels and remote sensing greenness indicators (VIs, LAI) as inputs, allowing it to capture both the direct and indirect CFE. In contrast, the Baseline model, lacking CO2 as an input, reflects only the indirect CFE trends through changes in LAI and VIs. We understand concerns regarding the "black box" nature of the ML models and would like to address that our objective is to accurately estimate GPP trends by incorporating both CFE pathways. Quantitatively disentangling these effects is critical; However, it would require separate investigations.

To verify the CFE-ML and CFE-Hybrid models, we validated long-term GPP trend estimates against eddy covariance site data, as detailed in Section 3.1.3 and Figure 5. Results show that the Baseline models substantially underestimated GPP long-term trends, while the CFE-ML and CFE-Hybrid models showed significant improvements aligning more closely with observed data. Since the only difference between the CFE-ML and Baseline models is the inclusion of CO2 input in CFE-ML models, these results suggest that the underestimation of the Baseline models is due to the overlooking of the direct effect of CO2 on GPP.

In response to the feedback, we have revised our introduction to better explain the direct and indirect CFE and highlighted our site-level validation results in the abstract. Additionally, we corrected the legend in Figure 5, which falsely indicated eddy covariance data (EC) as "Model."

**Revisions:**
Line 27 – 29 (Abstract): "*Incorporation of the direct CO2 effects substantially enhanced the predicted long-term trend in GPP across global flux towers by up to 51%, aligning much closer to a strong positive trend from eddy covariance data.*"
Line 82 – 91 (Section 1): "*Increasing atmospheric CO2 directly stimulates the biochemical rate or the light use efficiency (LUE) of leaf-level photosynthesis, known as the direct CO2 fertilization effect (CFE). Enhanced photosynthesis could lead to greater net carbon assimilation, contributing to an increase in total leaf area. This expansion, contributing to a higher light interception, further enhances canopy-level photosynthesis (i.e. GPP), which is referred to as the indirect CFE. The direct CFE has been found to dominate GPP responses to CO2 compared to the indirect effect, from both theoretical and observational analyses (Haverd et al., 2020; Chen et al., 2022).*"

Main point#3 The validation for the GPP product is inadequate, especially the GPP trend. It should have a specific validation for ten setups at Monthly, Seasonal, Annual trend for different vegetation types or climate zones. Therefore, the readers choose the specific region with high accuracy data they needed. I am not doubting that the global GPP has continuously increased during the past 40 years. However, different regions may have different greening/browning patterns, so what is their contribution to the global GPP trend? The spatial continuously product can answer this, so the author should first provide the GPP trend at sites and proved they are right, and then upscale this result to the global. Therefore, the site validation is very important!

**Response**: Thank you for the suggestion! We concur that model performance varies by biomes and climate zones. In the revised manuscript, we incorporated the assessment of predicted GPP trends in eddy covariance sites by plant functional types and climate zones in Section 3.1.3 and Figure 5. The results consistently suggest that the CFE-ML and CFE-Hybrid models agreed better with eddy covariance GPP than the Baseline model. We note that the validation results of monthly anomalies and seasonal cycles by biomes and climate zones are presented in Section 3.1.2 and Figure 4. We focus on evaluating long-term trends using annual GPP data instead of monthly/seasonal values to mitigate the impact of seasonality.

**Revisions:**

Line 467 – 485 (Section 3.1.3): *Aggregated eddy covariance GPP experienced increasing trends of varied magnitudes across different climate zones and plant functional types (Figure 5b,c; Figure S3b,c). While the machine learning models generally did not fully capture the enhancement in GPP for most categories, the CFE-ML and/or CFE-hybrid models consistently outperformed the Baseline models in both ST and LT setups. The CFE-ML setup predicted a higher trend than CFE-hybrid in most cases, suggesting that the data-driven approach captured more dynamics not represented in the theoretical model, which was based on conservative assumptions regarding the $CO_2$ sensitivity of photosynthesis (see Sect. 2.3.2 and Appendix A). The choice of remote sensing data (ST vs. LT configurations) did not lead to substantial differences in the predicted GPP trend. Most long-term flux sites (at least 10 years of records) with a significant trend experienced an increase in GPP, and the CFE-ML and/or CFE-hybrid models aligned closer to eddy covariance data than the Baseline models (Figure S4). Additionally, we found a considerably higher trend in eddy covariance GPP measurements derived from the day-time versus night-time partitioning approach, potentially associated with uncertainties in GPP partitioning methods (Figure S4). Yet, machine learning model predicted trends were not strongly affected by GPP partitioning methods (Figure S3, S4).*

Figure 5: Comparison of observed and predicted GPP (from NT models only) trends across eddy covariance flux towers by plant functional types and koppen climate zones.

Main point #4 The annual trend should be evaluated on the long-term reliable observation data. Although the authors used the site GPP with more than 5 years for validation by the method provided by Chen et al. 2022 PNAS. However, the ICOS and the FLUXNET2015 has the longer-term observation data, so the author can evaluate the long-term trend (>10 yr) from these sites. Sometimes, the earlier year GPP data will have more data missing during the year, which underestimated the annual GPP at sites, so the annual trend is not reliable when only using 5 years of GPP observations.

**Response**: Thanks for the comment. We aimed to incorporate a substantial number of sites for the robustness of our analysis which evaluates GPP anomalies aggregated over multiple sites. Therefore, we used a threshold of 5 years in site selection following Chen et al. (2022). Regarding missing data, we excluded one site year with more than one month of missing data. We have detailed the approach in Section 2.3.3 of the revised manuscript.

Furthermore, we performed additional analysis to evaluate GPP trends in sites with at least 10 years of valid data. We found that most sites showing a significant (p-value < 0.3) GPP trend exhibited a positive change. Notably, our CFE-ML and CFE-Hybrid models consistently outperformed the Baseline models in capturing trends in these sites. The results were incorporated in Section 3.1.3 and Figure S4.

**Revisions:**

Line 341 – 342 (Section 2.3.3): *We excluded a site-year if less than 11 months of data was available and used linear interpolation to fill the remaining temporal gaps.*

Line 475 – 478 (Section 3.1.3): *Most long-term flux sites (at least 10 years of records) with a significant trend experienced an increase in GPP, and the CFE-ML and/or CFE-hybrid models again aligned closer to eddy covariance data than the Baseline models (Figure S4).*

Figure S4 was newly added to compare estimated and observed trends in long-term eddy covariance sites.

Main point #5 Cross validation among the GPP products should have more discussions. Although the authors did some comparison between the CEDAR-GPP with existed products, they just showed the results, but did not analyze the results and provide the insight on which product can be improved or what is the inherit reason for the uncertainties. So the discussion section can list some viewpoints for the further GPP product improvement.

**Response**: Thank you for the suggestion! In the manuscript, we have discussed potential factors that led to differences between GPP products when relevant. For example, we discussed the potential impacts of eddy covariance data selection on the predicted GPP magnitudes in upscaled products (Section 4.1.1 last paragraph). Additionally, we analyzed the discrepancy in the global GPP trends across products and their connection to the representation of CO2 fertilization effect in Section 3.2.4. In the revised manuscript, we have expanded our discussions to provide more in-depth explanations for these differences and insights for future improvement, particularly regarding interannual variabilities (Section 3.2.3) and long-term trends (Section 4.1). Our discussion mainly focuses on machine learning upscaled products (Section 4.1, 4.2).

**Revisions:**
Line 570 – 574 (Section 3.2.3): *The lack of interannual variability in FLUXCOM-ERA5 is attributable to the use of mean seasonal cycles of remotely sensed vegetation greenness indicators rather than their dynamic time series*.
Line 741 – 743 (Section 4.1.2): *For example, the lack of GPP interannual variabilities in FLUXCOM-ERA5 manifests the importance of incorporating dynamic vegetation signals from remote sensing in the upscaling framework.*
Line 824 – 836 (Section 4.2): *Our results suggested that variations in the estimated GPP long-term trends from different products were largely related to the representation of CO2 fertilization. Products that did not consider the direct CO2 effect, including our Baseline models, FLUXSAT, FLUXCOM, and MODIS, showed minimal long-term changes in tropical GPP, while the CEDAR CFE-ML and CFE-Hybrid models demonstrated significant GPP increases aligning with predictions from the terrestrial biosphere models (Anav et al., 2015). FLUXCOM-ERA5, not accounting for dynamics changes in vegetation structures and CO2, did not capture either the direct or indirect CO2 fertilization resulting in a slight negative GPP trend attributable to shifted climate patterns. Notably, rEC-LUE exhibited contrasting trends before and after circa 2000, primarily attributed to changes in VPD, PAR, and LAI, while the direct CO2 fertilization effect remained consistent (Zheng et al., 2020). Nevertheless, considerable differences between CEDAR-GPP and rEC-LUE, as well as between our CFE-ML and CFE-Hybrid products, warrant more in-depth investigations into long-term GPP responses to changes in atmospheric CO2 and climate patterns.*

Main point #6 The water stress effect should be considered. Line 718 to L720, "Yet the model assumed a fixed ratio of leaf-internal to ambient CO2, and thus did not include any responses to vapor pressure deficit.". So is the CEDAR-GPP consider VPD effect for GPP? Although CFE is one of the most significant effects to the GPP trend. However, the VPD trend cannot be omitted. Li et al. 2023 reported that the VPD significantly affects the GPP at different vegetation types, so the CEDAR-GPP consider this or some setups reproduce such condition? If the CEDAR-GPP cannot reproduce the VPD effect, so under what condition can I use this product? The soil moisture is also coupled with VPD, so is the CEDAR-GPP consider the soil moisture stress, and can it reproduce the stress from soil moisture? These are needed to be mentioned.

Li, S., Wang, G., Zhu, C., Lu, J., Ullah, W., Hagan, D. F. T., ... & Peng, J. (2023). Vegetation growth due to CO2 fertilization is threatened by increasing vapor pressure deficit. Journal of Hydrology, 619, 129292.

**Response**: Thank you for the comment! We acknowledge the importance of VPD and soil moisture in influencing GPP dynamics. The mentioned sentence was intended to clarify that in our theoretical model within the CFE-Hybrid setup, the internal to ambient CO2 ratio ($\chi$) was held constant to a long-term average value typical for C3 plants, when describing CO2 sensitivity in photosynthetic rate. This assumption does not imply that the estimated GPP from CEDAR ignores VPD effects. Indeed, as we incorporated VPD from ERA5-Land and soil moisture from ESA-CCI as predictors in machine learning models (Section 2.2), CEDAR-GPP predictions are expected to represent both effects. Thus, trends in GPP driven by increasing VPD, as outlined in Li et al. (2023), should be captured by our model. We have revised the text for clarity and expanded our discussions on the uncertainties associated with $\chi$.

**Revisions:**
Line 792 – 799 (Section 4.2): *For simplicity, we assumed a fixed ratio of leaf internal to ambient CO2 ($\chi$) representing an average long-term value typical for C3 plants in the theoretical CO2 sensitivity function. However, $\chi$ varies by environmental conditions, including temperature and vapor pressure deficit, and robustly modeling these dependencies remains challenging (Wang et al., 2017). Future work could incorporate more comprehensive representations of the $\chi$ and evaluate how the associated uncertainties affect the quantification of GPP and its temporal variations.*

Specific comments:

L14 sugars and starches->carbohydrate
**Response**: We have modified it accordingly.

L26 the annual trend validation with statistics should be highlighted since it is the most important outcome for CFE induced GPP trend.
**Response:** Good point! We have revised the abstract accordingly.
**Revisions:** Line 27-30: *Incorporation of the direct $CO_2$ effects substantially improved the predicted long-term trend in GPP across global flux towers by up to 51%, aligning much closer to a strong GPP enhancement based on eddy covariance data.*

L38 references needed
**Response**: We have provided a reference to the statement.

L90 'yet, this important mechanism is still missing in GPP products upscale from in situ eddy covariance flux measurements.' This is not accurate, since Zheng et al. also parameterize the model by in situ eddy covariance flux measurements. So this sentence should be revised.
**Response**: Thanks! We have revised the sentence accordingly.
**Revisions:** Line 101 – 103: "*yet, this important mechanism is still missing in GPP products upscaled from in situ eddy covariance flux measurements based on machine learning models.*"

L125 So totally 233 sites are used but not listed, the authors should list them at the SI.

**Response**: Thanks for the comment! We have added a table listing all sites and their data citation in SI.
**Revisions:** Table S1

L137 The C3 and C4 map is constant among the investigated years or not? Will the C3/C4 vegetation fraction change?

**Response**: We used a static C4 proportion map, but the fraction could potentially vary over time. In the revision, we made the clarification and highlighted potential future improvements from incorporating dynamics C3/C4 distributions.
**Revisions:** Line 218 – 219: "*The C4 percentage dataset was constant over time.*"
Line 754 - 757: "*potential improvement may be achieved by incorporating datasets related to agricultural management practices (crop type, cultivar, irrigation, fertilization) (Xie et al., 2021), plant hydraulic and physiological properties (Liu et al., 2021), dynamic C4 plant distributions (Luo et al., 2024), root and soil characteristics (Stocker et al., 2023), as well as topography (Xie et al., 2023),*"

Table 1 'Surface reflectance b1 – b7, Vegetation indices (NIRv, NDVI, kNDVI, EVI, GCI, NDWI), percent snow'. What is GCI?

**Response**: Thanks for the note! The correct term should be 'CIgreen', representing Green Chlorophyll Index, as stated in the main text (Line 177 – 180). We have made the correction.

L 158 vegetation indexes or vegetation indices?

**Response**: We have changed "indexes" to "indices" in the manuscript.

L172 'PKU GIMMS LAI4g consisted of AVHRR-based LAI from 1982 to 2003 (generated using machine learning models trained with Landsat-based LAI data and NDVI4g) and MODIS BNU LAI from 2004 onwards (Yuan et al., 2011).' To my knowledge, PKU GIMMS LAI4g didn't use the Yuan et al 2011 as input data.

**Response**: Thanks for the comment. Upon reviewing the PKU GIMMS LAI4g paper (Cao et al., 2023), we confirmed that the AVHRR GIMMS LAI4g data was consolidated against the reprocessed MODIS LAI generated based on approaches from Yuan et al. (2011). The final product comprised AVHRR-based GIMMS LAI4g (1982-2003) and reprocessed MODIS LAI (2004-2020) (Sections 3.2 and 2.3 in Cao et al., 2023). We have rephrased our description to align with the details more accurately from the GIMMS LAI4g paper.
**Revisions:** Line 197 - 199: *PKU GIMMS LAI4g consisted of consolidated AVHRR-based LAI from 1982 to 2003 (generated using machine learning models trained with Landsat-based LAI data and NDVI4g) and reprocessed MODIS LAI (Yuan et al., 2011) from 2004 onwards.*

L210 'GIMMS LAI4g and NDVI4g data were only filled with mean seasonal cycle due to their low temporal resolution (bimonthly).' This is unclear, as the CEDAR-GPP with the minimum temporal resolution is at monthly, why the bimonthly data just filled with mean seasonal cycle?

**Response**: Thanks for the note. The original description was unclear. The temporal resolution of GIMMS LAI4g and NDVI4g is biweekly. We performed gap-filling at the native temporal

resolutions of each dataset before resampling them to a monthly time scale to produce CEDAR-GPP. Since vegetation structure could experience significant changes at bi-weekly (half-month) intervals, gap-filling using temporal medians within moving windows could introduce significant uncertainties and potentially over-smooth the time series. We have revised the text to clarify this aspect.

**Revisions:** Line 236 – 240: *GIMMS LAI4g and NDVI4g data were only filled with mean seasonal cycle due to their low temporal resolution (half-month). This is because vegetation structure could experience significant changes at half-month intervals, and gap-filling using temporal medians within moving windows could introduce considerable uncertainties and potentially over-smooth the time series.*

Figure 2 The CO2 should be separate at the left. Besides, I cannot understand why the CFE-ML is omit in the long-term product.

**Response**: Thank you for the suggestion. We consider CO2 concentration an atmospheric condition and thus have included it in the "Climate" category. We linked "Climate" directly to "Biophysical Theory" in the diagram since the theoretical function depends on both air temperature and CO2. Therefore, we did not present CO2 as a separate group for cohesion and simplicity of the schematic. We hope this addressed your concerns, but please feel free to let us know if our interpretation was inaccurate.

Regarding the exclusion of the CFE-ML in the long-term product., we would like to clarify that the model was trained on data spanning 2001 to 2020, during which CO2 levels ranged from 370 to 412 ppm. Applying it to the 1980-1990s, when CO2 levels were much lower (340 to 365 ppm), would likely result in inaccurate GPP responses to CO2. This is because machine learning models typically do not extrapolate outside the range of their training data. Unlike CO2, other climate and environmental conditions did not have dramatic shifts in their magnitudes, despite interannual variations. Therefore, we include only Baseline and CFE-Hybrid setups in the long-term products. In the revised manuscript, we have added a more detailed explanation of this choice.

**Revisions**: Line 268 – 272: *Due to the limited availability of eddy covariance observations before 2001, we did not apply the CFE-ML approach to the long-term setups. The CFE-ML model, when trained on data from 2001 to 2020 with atmospheric CO2 ranging from 370 to 412 ppm, would not accurately predict GPP response to CO2 for the period 1982 – 2000 when the CO2 levels were markedly lower (roughly 340 – 369 ppm). This is because machine learning models, especially tree-based models, could not extrapolate beyond the range of the training data.*

L236 should be section 2.3.2

**Response**: Thanks for the note. We have made the correction.

L238- L241 If the CFE-ML model is well trained by the LAI4g and other input data during 2001-2020, it could also be extent 1982-2000. So why missing the flux data before 2000 hinder the CFE-ML processing? The CFE-ML before 2000 should also be compared to other products.

**Response**: Thanks for the question. We have clarified the rationale for excluding the CFE-ML setup from our long-term products in our response earlier (the comment starting with "Figure 2"). We have revised the text to clarify the underlying reasons for excluding CFE-ML models in the long-term product.

**Revisions:** Line 268 – 272: *Due to the limited availability of eddy covariance observations before 2001, we did not apply the CFE-ML approach to the long-term setups. The CFE-ML model, when trained on data from 2001 to 2020 with atmospheric CO2 ranging from 370 to 412 ppm, would not accurately predict GPP response to CO2 for the period 1982 – 2000 when the CO2 levels were markedly lower (roughly 340 – 369 ppm). This is because machine learning models, especially tree-based models, could not extrapolate beyond the range of the training data.*

L249 ' do not consider the direct effect of CO2 on light use efficiency', this is inaccurate, the VI sometimes can capture the LUE change from direct CFE.

**Response**: We would like to clarify that vegetation indices, such as NDVI and EVI, primarily respond to vegetation structural variations, reflecting the amount of healthy green biomass. They were designed to exploit the strong reflectivity in NIR associated with light scattering within leaf internal mesophyll structures rather than leaf biochemical properties (Gamon et al., 1996). Therefore, these VIs mainly capture changes in vegetation greenness or LAI, indicating canopy light absorption (fAPAR). They do not directly reflect changes in light use efficiency (LUE) associated with the CO2 impacts on the photosynthetic biochemical processes. Thus, our Baseline model, along with other upscaled datasets including FLUXCOM and FLUXSAT, does not consider the direct CFE on LUE.

It's also noteworthy that the narrowband Photochemical Reflectance Index (PRI), sensitive to xanthophyll pigment changes, reflects leaf to canopy level LUE, particularly under physiological stresses (Garbulsky et al., 2014; Penuelas et al., 2011). PRI can be derived from satellite data, such as ocean bands from MODIS. However, its relationships with LUE are confounded by sun-sensor geometry, vegetation structure, and soil background (Middleton et al., 2016). Therefore, future research is necessary to assess the potential of PRI data derived from satellites in tracking long-term LUE changes.

We have revised the corresponding sentence and improved the description of the direct and indirect CFE in the introduction for clarity. We hope that explanations in our response and revisions throughout the manuscript clarify this important aspect of our work compared to existing upscaled datasets.

Reference:
Gamon, J. A., Field, C. B., Goulden, M. L., Griffin, K. L., Hartley, A. E., Joel, G., Peñuelas, J., and Valentini, R.: Relationships Between NDVI, Canopy Structure, and Photosynthesis in Three Californian Vegetation Types, Ecological Applications, 5, 28–41, https://doi.org/10.2307/1942049, 1995.
Garbulsky, M. F., Filella, I., Verger, A., and Peñuelas, J.: Photosynthetic light use efficiency from satellite sensors: From global to Mediterranean vegetation, Environmental and Experimental Botany, 103, 3–11, https://doi.org/10.1016/j.envexpbot.2013.10.009, 2014.
Middleton, E. M., Huemmrich, K. F., Landis, D. R., Black, T. A., Barr, A. G., and McCaughey, J. H.: Photosynthetic efficiency of northern forest ecosystems using a MODIS-derived Photochemical Reflectance Index (PRI), Remote Sensing of Environment, 187, 345–366, https://doi.org/10.1016/j.rse.2016.10.021, 2016.
Peñuelas, J., Garbulsky, M. F., and Filella, I.: Photochemical reflectance index (PRI) and remote sensing of plant CO2 uptake, New Phytologist, 191, 596–599, https://doi.org/10.1111/j.1469-8137.2011.03791.x, 2011.

**Revisions**: Line 82 - 91 (Introduction): *Increasing atmospheric CO2 directly stimulates the biochemical rate or the light use efficiency (LUE) of leaf-level photosynthesis, known as the*

*direct CO2 fertilization effect (CFE). Enhanced photosynthesis could lead to greater net carbon assimilation, contributing to an increase in total leaf area. This expansion, contributing to a higher light interception, further enhances canopy-level photosynthesis (i.e. GPP), which is referred to as the indirect CFE. The direct CFE has been found to dominate GPP responses to CO2 compared to the indirect effect, from both theoretical and observational analyses (Haverd et al., 2020; Chen et al., 2022).*

Line 279 – 283 *(Section 2.3.2): As such, the models only include indirect CO2 effects from the satellite-based proxies of vegetation greenness or structure representing changes in canopy light interception, and they do not consider the direct effect of CO2 on leaf-level photosynthetic rates (or light use efficiency, LUE). Our baseline model is therefore directly comparable to other satellite-derived GPP products that only account for indirect CO2 effects (Jung et al., 2020; Joiner and Yoshida, 2020).*

L257-271 an open question here, I read the EEO theory from Chen et al. 2022 PNAS. The authors may refer to the SI from that paper, you may see the EEO theory cannot reproduce the GPP trend at tropical rain forests (GF-Guy). So whether the CFE-hybrid can capture the actual trend of GPP, more validation is needed.

**Response**: Thanks for the question. We acknowledge that it remains challenging to accurately model long-term GPP trends, and all existing models present uncertainties of various extent, and some models, including the EEO model by Chen et al., (2022), are underperformed in specific biomes such as evergreen broadleaf forests. In the revised manuscript, we have included direct validation of the CFE-Hybrid and CFE-ML models in predicting long-term trends across global eddy covariance sites, categorized by different plant functional types and climate zones. These results could provide comprehensive information on the uncertainties of our products in representing GPP dynamics.
**Revisions:** Section 3.2.4, Figure 5, Figure S3, S4

L325 To my knowledge, the FLUXCOM has removed the annual trend which has been pointed out by the data use guideline, so why used the FLUXCOM data?

**Response**: Thank you for the question. The FLUXCOM product, as described in Jung et al. (2020), "does not account for CO2 fertilization effects", hence presents unrealistic trends in carbon fluxes. (Note that it was not specifically detrended.) The CEDAR-GPP product aimed to address this issue by introducing the direct CO2 effect into upscaled GPP estimates. Comparing our product with FLUXCOM thus presents an informative assessment of the effectiveness of our approach in producing long-term GPP trends. Moreover, benchmarking CEDAR-GPP with FLUXCOM and other products also provides useful evaluation regarding GPP spatiotemporal patterns, including mean annual values, mean seasonal cycles, and interannual variability.

Figure3 it should have systematic validation for ten setups but not only validate the CFE-Hybrid_NT. Besides, the GPP anomaly seems to be underestimated compared to the site observation?

**Response**: Thank you for the comment! We would like to clarify that Figure 3b, d, f, h shows R2 for all five models based on the NT setup, and Figure S1 displays the results for the other five models (DT). Table S3 presented the performance metrics (RMSE, Bias, and R2) for the estimation of monthly GPP and its mean seasonal cycles and interannual variabilities by all ten model setups. Additionally, thanks for your observation regarding the underperformance of GPP anomaly prediction, which is a long-standing challenge documented by previous investigations.

In the revised manuscript, we have improved the clarity of the text when describing the results of GPP anomalies.

**Revisions:** Line 399 – 400 (Section 3.1.1): *However, all models underestimated monthly anomalies across the sites, with R2 values below 0.12 (Figure 3e-f).*

Line 750 – 757 (Section 4.1.2): *Nevertheless, accurately capturing interannual anomalies remains challenging for certain biomes, such as evergreen needleleaf forest, cropland, and wetland (Figure 4), as acknowledged by previous studies (Tramontana et al., 2016; Jung et al., 2020). This suggests that vital information on GPP is missing or inadequately represented in existing datasets. To this end, potential improvement may be achieved by incorporating datasets related to agricultural management practices (crop type, cultivar, irrigation, fertilization) (Xie et al., 2021), plant hydraulic and physiological properties (Liu et al., 2021), dynamic C4 plant distributions (Luo et al., 2024), root and soil characteristics (Stocker et al., 2023), as well as topography (Xie et al., 2023).*

Figure5 the trend should be analyzed among PFT and climate zone similar to figure4. More importantly, the sites showed a higher GPP trend than the models. Do you mean the CEDAR-GPP underestimate the GPP trend? Why the trend in CFE-ML-NT is much higher than it in CFE-ML-DT?

**Response**: Thank you for the feedback. We have incorporated additional evaluation results by PFT and climate zones in Figure 5 and noted the underestimation of the GPP trend by the CEDAR models in section 3.1.3.

Moreover, the observed differences between the NT and DT models for the CFE-ML setup reflect that the GPP trends from the NT partitioning can be more effectively captured by machine learning, based on the set of predictor variables, than those from DT based on GPP. This discrepancy may highlight the underlying uncertainties or systematic biases in the NT and DT approaches (see Section 4.1.1, Line 668 – 671). These results also suggest the considerable uncertainties present in the machine learning quantification of CO2 fertilization and underscore the need for further studies using advanced approaches, such as explainable machine learning and causal inference.

**Revisions:** Figure 5, Section 3.1.3

Line 468 – 471 (Section 3.1.3) *While the machine learning models generally did not fully capture the enhancement in GPP for most categories, the CFE-ML and/or CFE-hybrid models consistently outperformed the Baseline models in both ST and LT setups.*

Line 816 – 823 (Section 4.2): *Nonetheless, the considerable ensemble spread in the CO2 trends from the CFE-ML model and discrepancies between the CFE setups (Figure 11, Figure 13) underscored a high level of uncertainty in the machine learning quantified CO2 effects. Moreover, disentangling the direct CO2 effects on LUE, water use efficiency, and its indirect effects on fAPAR remains challenging with machine learning models due to the correlations and interactions between CO2 and other climatic or environmental factors. Future work may exploit explainable machine learning and causal inference to unravel the complex mechanisms and distinct pathways of CO2 effects on vegetation carbon uptake.*

Figure7 The standard deviation of GPP at different months should be added to the figure.

**Response**: Thank you for the suggestion. Since the global and regional variations of GPP mean seasonal cycles are substantial, we are concerned that adding the standard deviation to Figure 7 may reduce its clarity. Instead, we have provided this information in an additional figure (Figure S7) in the Supplementary Information.

**Revisions:** Figure S7

Figure8 The FLUXCOM data cannot be applied to evaluate the IAV as the product guideline has pointed out. The baseline and CFE-ML product should be listed here.

**Response**: Thanks for the comment! We recognize the issue of underestimated IAV in FLUXCOM product noted by Jung et al., (2020). However, we used an updated FLUXCOM GPP dataset based on MODIS v006 dataset (as indicated in Section 2.4), which showed substantial improvement in IAV representation (as shown in Figure 8).

Regarding the CEDAR-GPP model setups, we presented the estimated GPP IAV from all CEDAR model setups in Figure S7, where all models showed consistent IAV patterns. Therefore, in Figure 8, which aimed to compare IAV patterns between CEDAR-GPP and other products, we selected the ST_CFE-Hybrid_NT as a representative example to minimize redundancy. We have added a clarification regarding this aspect in the text.

**Revisions:** Line 572 – 574 (Section 3.2.3): *Ten CEDAR-GPP model setups presented consistent patterns in interannual variability, and differences were minimal (Figure S7).*

Figure11 Can I infer the GPP trend from 2001 to 2018 is double compared to it from 1982 to 2000? Is there any existed GPP product (i.e. machine learning or TRENDY data) can support this result?

**Response**: Thanks for pointing this out! We would like to clarify that trends from the ST (Figure 11b) and LT setups (Figure 11d) are not directly comparable because the models were driven by different sets of remote sensing input datasets. For trends analysis between 2001 and 2020, we recommend using the short-term (ST) datasets as they were based on a more comprehensive set of remote sensing data and have demonstrated a better performance in capturing trends from the eddy covariance data (Figure 5). To analyze trends from 1982 to 2020 and assess shifts in trends between 1982-2000 and 2001-2020, the long-term (LT) dataset should be used for consistency in underlying satellite data and machine learning models. We have highlighted these distinctions and considerations in the newly added user guideline.

The LT product (Figure 11c) did indicate a small difference in GPP trends between the 1982 - 2000 and the 2001 – 2020 periods (Figure S10). This suggests a potential shift in trends from the underlying GIMMS datasets. Also, note that the significance of this difference can be affected by the specific start and end years chosen for analysis due to significant interannual variabilities in GPP. Thus, future analysis would benefit from careful statistical evaluation and intercomparison with other products to robustly assess the changes in GPP trend over time.

**Revisions:** Line 850 – 854 (Section 5): *1) Study period considerations: the Short-Term (ST) setup is ideal for studies focusing on periods after 2000. These models are constructed using a broader range of explanatory predictors, offering higher precision and smaller random errors. The Long-Term (LT) datasets shall be used for research assessing GPP dynamics over a longer time period (before 2001). It is important to note that trends from the ST and LT datasets are not directly comparable, as they were derived from different satellite remote sensing data.*

Function A2 $A = A_C = A_j$? I cannot agree with this! There are just one condition Ac = Aj is the photosynthesis transfer from lighted-limited to nutrient limited.

**Response**: We would like to clarify that our derivation of the CO2 sensitivity is based on the eco-evolution theory, which predicts that light-limited (A_j) and Rubisco-limited (A_c) converges through Vcmax acclimation to optimize resource allocation over the scale of a few weeks (Harrison et al., 2021; Wang et al., 2017). Thus, the assumption of the equality of A_c and A_j is

valid at the monthly time scale in our model. Additionally, given that Aj has a lower response to CO2 than A_c (Walker et al., 2021), our approach provides conservative estimates of the direct CFE. We described this approach and the underlying assumptions in the main text (Section 2.3.2). In the revised manuscript, we have further elaborated our approach in Appendix A.

**Revisions:** Line 902 – 906 (Appendix A): *Eco-evolutionary theory predicts that the electron-transport-limited (light-limited) (A_j) and Rubisco-limited (A_c) rates of photosynthesis converge on the time scale of physiological acclimation, which is in the order of a few weeks (Harrison et al., 2021; Wang et al., 2017). Thus, at a monthly time scale, we assume that …A = A_c = A_j (A3).*

Line 908 – 910: *In the following, we derive our sensitivity function based on A_j, which has a smaller response to CO2 than A_c , thus providing conservative estimates of the direct CO2 fertilization effect (Walker et al., 2021).*

---

## Author Comment (AC2)

**Response Letter to Reviewer #3**
**CEDAR-GPP: spatiotemporally upscaled estimates of gross primary productivity incorporating $CO_2$ fertilization**

Dear reviewer,

Thank you very much for your thorough review of our manuscript. We greatly appreciate your constructive feedback and comments, which have significantly improved our paper. Below, we outline our responses (in blue) to your comments (in black) and detail the corresponding revisions made to the manuscript (in red). We included line numbers from the "track-change" version of the revised manuscript for your reference.

Thank you once again for the time and efforts you dedicated to reviewing our manuscript. We hope that our revisions have adequately addressed your comments and concerns. We eagerly look forward to any further feedback and suggestions that you may have!

Sincerely,

Yanghui Kang
On behalf of all co-authors

In this manuscript, Kang and colleagues present a novel GPP product that incorporates the CO2 fertilization effect. After several thorough readings over the past two weeks, it has become evident to me that this manuscript is both timely and well-prepared. I have some minor suggestions to enhance its clarity and impact:

1. In the Introduction section, it would be beneficial to provide a more comprehensive explanation of why the authors chose scaling as their approach for global GPP development. Given that there are various other GPP development methods, such as LUE models and SIF retrievals, it's essential to highlight the unique advantages of scaling in comparison to these alternatives.

**Response**: This is a great point. We have enriched the introduction with a more detailed description of the upscaling approach and its advantages.
**Revisions:** Line 56 – 64: *This "upscaling" approach provides data-driven and observation-based quantifications without prescribed functional relations between GPP and its climatic or environmental drivers. It offers unique empirical constraints of ecosystem carbon dynamics, complementing those derived from process-based and semi-process-based approaches such as terrestrial biosphere models or the Light Use Efficiency (LUE) models (Beer et al., 2010; Jung et al., 2017; Schwalm et al., 2017; Gampe et al., 2021). In recent years, the growth of global and regional flux networks, coupled with increasing efforts in data standardization, has offered new opportunities for the advancement of upscaling frameworks, enabling comprehensive quantifications of terrestrial photosynthesis (Joiner and Yoshida, 2020; Pastorello et al., 2020).*

2. The concept of "direct CO2 fertilization effect" may not be familiar to all readers. It would be helpful if the authors could provide a clearer explanation of this term and include relevant references in the introduction to ensure a better understanding among readers.

**Response**: Thank you for the suggestion! We have revised the introduction to provide a clearer description of the direct and indirect CO2 fertilization effect.
**Revisions:** Line 82 – 19: *Increasing atmospheric CO2 directly stimulates the biochemical rate or the light use efficiency (LUE) of leaf-level photosynthesis, known as the direct CO2 fertilization effect (CFE). Enhanced photosynthesis could lead to greater net carbon assimilation, contributing to an increase in total leaf area. This expansion, contributing to a higher light interception, further enhances canopy-level photosynthesis (i.e. GPP), which is referred to as the indirect CFE. The direct CFE has been found to dominate GPP responses to CO2 compared to the indirect effect, from both theoretical and observational analyses (Haverd et al., 2020; Chen et al., 2022).*

3. Figure 1 effectively illustrates the spatial distribution of flux tower data used in this study. it would be more beneficial to indicate how these flux tower sites represent different biomes. I would suggest adding a Whittaker biome figure within Figure 1 to visually depict the biome types associated with the flux tower locations used in the study.

**Response**: Thank you for the great suggestion! We have added a Whittaker biome plot to illustrate site distributions in the climate space in Figure 1d. Additionally, we have incorporated a plot showing the site distribution by the actual plant functional types (Figure 1c). We would also like to note that the number of sites in each biome and climate zone was also provided in Figure 4 to assist the interpretation of the results.
**Revisions:** Line 138 – 140 (Section 2.1): *Despite their uneven geographical distribution, these sites effectively cover a diverse range of climatic conditions and are representative of global biomes (Figure 1c, 1d).*

4. Given the potential wide interest in the various GPP products developed in this work, it would be very useful to include a section that provides guidance on how future researchers and users can effectively utilize these datasets in the future. This could include tips on data access, processing, and interpretation to facilitate broader adoption.

**Response**: Thank you for the suggestion! In the revised manuscript, we have provided a comprehensive guideline on data usage and selection in the Section "Data availability and usage". We have also incorporated this information into the user guide in the Zenodo data repository.
**Revisions:** Line 845 – 868 (Section 5): *The CEDAR GPP product offers GPP estimates derived from ten different models. Models are characterized by 1) temporal coverage, 2) configuration of CO2 fertilization, and 3) GPP partitioning approach (Table 2). We provide a structured approach to selecting the most appropriate dataset for research or applications.*
        *1) Study period considerations: the Short-Term (ST) setup is ideal for studies focusing on periods after 2000. These models are constructed using a broader range of explanatory predictors, offering higher precision and smaller random errors. The Long-Term (LT) datasets shall be used for research assessing GPP dynamics over a longer time period (before 2001). It is important to note that, due to basis in different satellite remote sensing data, trends from the ST and LT datasets are not directly comparable.*

*2) CO2 Fertilization Effect (CFE) configurations: the CFE-Hybrid and CFE-ML setups are preferable when assessing temporal GPP dynamics, especially long-term trends. The CFE-Hybrid setup includes a hypothetical trend for the direct CO2 effect, while CFE-ML is purely data-driven and does not make any specific assumption about the sensitivity of photosynthesis to CO2. Averaging the CFE-Hybrid and CFE-ML estimates is acceptable, with the difference between them reflecting the uncertainty surrounding the direct CO2 effect. Note that the Baseline setup should not be used to study long-term GPP dynamics, especially those induced by elevated CO2. The Baseline setup may be useful to compare with other remote sensing-derived GPP datasets that do not consider the direct CO2 effect. Differences between these setups regarding mean GPP spatial patterns, seasonal and interannual variations are minor.*

*3) GPP partitioning methods: We recommend using the mean value derived from both the "NT" (Nighttime) and "DT" (Daytime). The difference between these two provides insight into the uncertainties arising from the partitioning approaches used in GPP estimation from eddy covariance measurements.*

5. To streamline the manuscript, consider moving some supplementary materials, such as Table 1 and Table 3, to the supplementary section. This would help maintain a concise and focused main manuscript while still providing access to important supporting information.

**Response**: Thank you again for the feedback! To streamline the manuscript, we have moved Table 3 and results related to the DT models (i.e. setups based on day-time partitioned GPP) in Figure 3, Figure 5 to the supplementary information. We have opted to keep Table 1 in the main text as it provides essential information on the input datasets, which are critical to our upscaling approaches.

---

## Author Comment (AC3)

**Response Letter to Reviewer #2**
**CEDAR-GPP: spatiotemporally upscaled estimates of gross primary productivity incorporating $CO_2$ fertilization**

Dear Dr. Wang,

Thank you very much for your thorough review of and positive comments on our manuscript. We greatly appreciate your constructive feedback, which has significantly improved our paper. Below, we outline our responses (in blue) to your comments (in black) and detail the corresponding revisions made to the manuscript (in red). We included line numbers from the "track-change" version of the revised manuscript for your reference.

Thank you once again for the time and efforts you dedicated to reviewing our manuscript. We hope that our revisions have adequately addressed your comments and concerns. We eagerly look forward to any further feedback and suggestions that you may have!

Sincerely,

Yanghui Kang
On behalf of all co-authors

General comments:

This manuscript presents a new global GPP dataset by using the machine learning method and many various datasets. They mainly considered the direct CO2 fertilization effect on GPP through both the machine learning and the theoretical approaches. They then modelled the short and long-term global GPP by using different periods of earth observations and climate variables. They also validated their new GPP data by comparing to flux sites GPP data and other products.

This study fits well for ESSD and I appreciate the efforts that the authors made. Several minor concerns from my side are listed below, which I think may be helpful for revising the manuscript.

Specific comments:

Line 20: Maybe not the "first", since you also said that the revised EC-LUE GPP have considered this.

**Response**: Thank you for pointing this out. We have revised this sentence to elaborate our product was the first dataset upscaled from eddy covariance measurements using machine learning models that incorporated the direct CO2 fertilization effect.
**Revisions:** Line 20 – 21 (Abstract): *Here, we introduce CEDAR-GPP, the first global machine-learning-upscaled GPP product that incorporates the direct CO2 fertilization effect on photosynthesis.*

Line 114: Some flux sites have the data before 2001? Why did include them into the model? And, this means that the information before 2001 from flux sites are not included into your GPP products, right?

Response: Thank you for the comment! It is correct that eddy covariance data before 2001 were not included in our models. While we recognize that sites with data back to the 1990s are valuable for evaluating long-term changes, such sites are very few (e.g. only four sites have data before 1996) and could induce model biases towards a limited range of environmental conditions in these early years. This is particularly critical for modeling the relationship between GPP and CO2. Therefore, we were concerned that applying such models globally could result in unreliable predictions. Moreover, the short-term models were based on MODIS data that was only available after 2000, and thus did not require training data beyond the time range. This approach also ensures the consistency between our short-term (ST) and long-term (LT) models.
Revisions: Line 140 – 143 (Section 2.1): *Note that we did not include eddy covariance data before 2001, since it was limited to only a few sites. This scarcity might introduce biases in the machine learning models, particularly in the relationship between GPP and CO2, leading to unreliable extrapolations across space and time*.

Line 214: Both the MODIS and CISF data are used in the model. Will them from the same observations, i.e., the CSIF is also be trained from MODIS data right. Will the multi-collinearity issue exist in this?

Response: This is a great point. Indeed, CSIF, along with other variables derived from MODIS, are highly correlated. We expect the XGBoost model to effectively manage such multi-collinearity in the feature space, as it does not rely on assumptions about the statistical distributions of the predictor variables. While multi-collinearity may affect model interpretability, repetitive information shared among the different MODIS-derived variables can actually enhance the machine learning model's predictive ability and reduce noise.
Revisions: Line 319 – 321 (Section 2.3.3): *Without relying on prior assumptions about the functional forms or statistical distributions, the model is also robust to multi-collinearity between the predictors in our dataset, particularly for the variables derived from MODIS data*.

Line 252-256: When adding the CO2 into the machine-learning model, the output of your model is GPP, but LUE. That means, the model could not distinguish the direct CO2 fertilization on LUE and the indirect CO2 impact on FAPAR? A simple way to check this is to check the CO2 impacts on GPP, LUE and FAPAR at some FACE sites, and to check that if your model is correct.

Response: Thank you for the feedback and suggestion! We agree that disentangling the direct CO2 effect on LUE and the indirect effects on FAPAR is a major challenge within machine learning frameworks, which primarily focus on optimizing the overall predictive accuracy of GPP.
    Indeed, FACE experiments offer unique opportunities to quantify the direct CO2 effect on LUE and indirect effects on LAI, as thoroughly analyzed by previous studies analysis (e.g. Ainsworth et al., 2005). However, applying our machine learning models directly to such experimental conditions is challenging. These experiments involved elevated CO2 levels (e.g., 542 ppm in Duke, 547 ppm in Oak Ridge) significantly above current ambient concentration (420 ppm as of 2020). Given that the machine learning models do not extrapolate over unseen conditions, they cannot reliably produce predictions under the FACE experiment scenarios.

Our objective was to provide comprehensive quantifications of the overall GPP trends, encompassing both the direct and indirect CFE, to address omissions of the direct effects in previous upscaled datasets. By incorporating both CO2 and remote sensing proxies indicating canopy structure (LAI, fAPAR) as predictors, our machine learning models (i.e. CFE-ML) aimed to capture the overall CO2 effects. The enhanced performance of the CFE-ML models compared to the baseline models, in capturing the significant increase of GPP from eddy covariance data, suggests that the CFE-ML model effectively captured the direct CFE absent in the Baseline models, providing an improved representation of the overall trend.

In light of your valuable feedback, we have expanded our discussion (Section 4.2) to emphasize the importance and challenges of isolating the direct and indirect CFE with the machine learning frameworks and highlight future opportunities through advanced machine learning and causal inference techniques. Moreover, we have extended our model validation against eddy covariance data across different plant functional types and climate zones (Section 3.1.3, Figure 5, Figure S3), and included further analysis at sites undergoing long-term GPP changes (Figure S4). These additional analyses aimed to demonstrate our models' capacity to capture the overall GPP trends including both the direct and indirect CFE.

**Revisions:** Line 818 – 823 (Section 4.2): *Moreover, disentangling the direct CO2 effects on LUE, water use efficiency, and its indirect effects on fAPAR remains challenging with machine learning models due to the correlations and interactions between CO2 and other climatic or environmental factors. Future work may exploit explainable machine learning and causal inference to unravel the complex mechanisms and distinct pathways of CO2 effects on vegetation carbon uptake.*

Line 467 – 485 (Section 3.1.3): *Aggregated eddy covariance GPP experienced increasing trends of varied magnitudes across different climate zones and plant functional types (Figure 5b,c; Figure S3b,c). While the machine learning models generally did not fully capture the enhancement in GPP for most categories, the CFE-ML and/or CFE-hybrid models consistently outperformed the Baseline models in both ST and LT setups. The CFE-ML setup predicted a higher trend than CFE-hybrid in most cases, suggesting that the data-driven approach captured more dynamics not represented in the theoretical model, which was based on conservative assumptions regarding the CO2 sensitivity of photosynthesis (see Sect. 2.3.3 and Appendix A). The choice of remote sensing data (ST vs. LT configurations) did not lead to substantial differences in the predicted GPP trend. Most long-term flux sites (at least 10 years of records) with a significant trend experienced an increase in GPP, and the CFE-ML and/or CFE-hybrid models again aligned closer to eddy covariance data than the Baseline models (Figure S4). Additionally, we found a considerably higher trend in eddy covariance GPP measurements derived from the day-time versus night-time partitioning approach, potentially associated with uncertainties in GPP partitioning methods (Figure S4). Yet, machine learning model predicted trends were not strongly affected by GPP partitioning methods (Figure S3, S4).*

Line 261-262: A constant $\chi$ value for global and long-term analysis seems will introduce large uncertainties. Since we know that $\chi$ will change with different climate conditions, and also will change largely with the CO2 concentration. Although we know that the modelling of global $\chi$ is very hard, but some discussions and limitations on this aspect is at least needed.

**Response**: Thank you for the feedback! We totally concur that applying a constant $\chi$ value in global and long-term analysis can introduce uncertainties. We have expanded our discussion to underscore the dynamics of $\chi$ in response to meteorological conditions and associated uncertainties, and provided insights for potential improvements in future work.

**Revisions:** Line 792 – 799 (Section 4.2): *For simplicity, we assumed a fixed ratio of leaf internal to ambient CO2 ($\chi$) representing an average long-term value typical for C3 plants in the*

*theoretical CO2 sensitivity function. However, χ varies by environmental conditions, including temperature and vapor pressure deficit, and robustly modeling these dependencies remains challenging (Wang et al., 2017). Future work could incorporate more comprehensive representations of the χ and evaluate how the associated uncertainties affect the quantification of GPP and its temporal variations.*

---

## Author Response (AR2)

**Response Letter**

**CEDAR-GPP: spatiotemporally upscaled estimates of gross primary productivity incorporating CO2 fertilization**

**Reviewer #1**

Dear reviewer,

Thank you very much for your review and feedback of our revised manuscript! We appreciate the opportunity to further clarify these issues and address any remaining concerns. Please find our detailed point-to-point response below.

Sincerely, Yanghui Kang On behalf of all authors

[**Reviewer Comment 1**] I felt disappointed that the authors didn't run their model again since the GPP is wrong as I mentioned at the last round of comments.

**Response**: We regret that our previous revisions did not fully resolve your concerns regarding the GPP model. In our last revision, we aimed to address concerns related to GPP modeling, on the incorporation of direct CO2 fertilization effect (CFE) and the capacity of VIs to represent the indirect CFE. We clarified potential misunderstandings about these aspects and improved the clarity of the related text in our manuscript and responses. We also performed additional validation to demonstrate the robustness of our models in quantifying GPP and particularly its long-term trends. We are sorry to hear that you felt disappointed that we did not run our model again to address the concerns raised, but we did not feel that was necessary.

Additionally, we would also like to emphasize that we demonstrated the robustness of our GPP models through rigorous validation and evaluation (Section 3.1, Figure 3-5, Figure S1-4, Table S3, also see responses below). Our models' performance in cross-validation with the eddy covariance data aligns well with previous datasets. Moreover, intercomparisons with other RS-based datasets, including FLUXCOM, FLUXSAT, and MODIS, confirm strong consistency in global patterns of annual mean GPP, interannual variability, and mean seasonal cycles (Section 3.2, Figure 6-9, Figure S5-8).

Our estimates of long-term trends agree with eddy covariance data across global sites. Notably, the Baseline model, which does not consider the direct CFE, underestimates GPP trends at flux towers. In contrast, the CFE-ML and CFE-Hybrid models show significant improvements, underscoring the need to consider both direct and indirect CFE. In our last revision, we performed additional validation of GPP trend by climate zones and plant functional types (Figure 5b, 5c). We also provided comparisons of estimated and observed trends at long-term sites (Figure S4).

We recognize the limitations in validation within tropical areas, which is due to data scarcity. In our response to your next comment (see Page 5-9 of this document), we show that CEDAR-GPP exhibits higher consistency with TRENDY models after incorporating the direct CFE.

Below we provide further clarifications and additional analyses to address previous comments concerning GPP modeling. We hope that these assuage your concerns but would welcome further suggestions for improvement.

[**Previous comment: Main point #6**]: The water stress effect should be considered. Line 718 to L720, "Yet the model assumed a fixed ratio of leaf-internal to ambient CO2, and thus did not include any responses to vapor pressure deficit.". So is the CEDAR-GPP consider VPD effect for GPP? Although CFE is one of the most significant effects to the GPP trend. However, the VPD trend cannot be omitted. Li et al. 2023 reported that the VPD significantly affects the GPP at different vegetation types, so the CEDAR-GPP consider this or some setups reproduce such condition? If the CEDAR-GPP cannot reproduce the VPD effect, so under what condition can I use this product? The soil moisture is also coupled with VPD, so is the CEDAR-GPP consider the soil moisture stress, and can it reproduce the stress from soil moisture? These are needed to be mentioned.

**Response**: We would like to highlight that water stress factors, such as atmospheric dryness (VPD), precipitation, and soil water availability (soil moisture) were indeed considered in our GPP predictions. We include these factors as explanatory variables in our machine learning models. The sentence highlighted by the reviewer here relates to our approach of simulating the CO2 sensitivity of LUE (i.e. the direct CFE) in the CFE-Hybrid approach. This is not directly relevant to water stresses.

In the CFE-Hybrid model, we prescribed the direct CFE onto GPP estimated by machine learning models at a reference  $CO_2$  level, which already accounts for water stress. The direct CFE was quantified by a  $CO_2$  sensitivity function of LUE based on the optimal coordination theory. Our  $CO_2$  sensitivity function of LUE uses a fixed leaf-internal to ambient  $CO_2$ concentration ratio (ci/ca) based on the global long-term average, without considering potential responses to air temperature or VPD.

In this round of revision, we provided additional analyses to demonstrate that the fixed ci/ca approach provided consistent results compared to a more dynamic approach, justifying our choice of using the simplest model with equivalent performance. We applied a dynamic ci/ca model based on temperature and VPD to estimate the direct CFE at monthly intervals across the global flux towers. Results show that the direct CFE and GPP trends from CFE-Hybrid model are highly consistent between the fixed and dynamic ci/ca approaches. The dynamic model predicted ci/ca has a mean of 0.7, consistent with the global long-term mean value used in our fixed ci/ca approach. The standard deviation is only 0.04 suggesting that ci/ca is relatively stable. We have incorporated this analysis and relevant figures in the newly revised manuscript. Here we provide the relevant figure. Additionally, we also revised the description of the CO2 sensitivity function of LUE for clarity.

Figure S11. Comparison of CO2 sensitivity of LUE with dynamic vs. fixed values of  $\chi$ , i.e. the leaf internal to atmospheric CO2 concentration ratio (ci/ca). The dynamic model simulates  $\chi$  as a function of air temperature and VPD, whereas the other approach has a fixed  $\chi$  at the global long-term average ( $\chi$ =0.7). (a) Statistical distribution of ci/ca (monthly values) across global eddy covariance tower. (b) Comparison of the direct CO2 fertilization effect (CFE) between the two models. The direct CFE is quantified as the ratio between LUE under ambient CO2 levels and LUE at a reference CO2 level (the value of year 2001). This corresponds to the ( $\phi_{CO_2}^t/\phi_{CO_2}^{t_0}$ ) term in eq. A8. (c) Aggregated GPP trends across global flux towers over 2002 to 2019 from eddy covariance data and model estimates. The CFE-Hybrid-fixed model assumes a constant ci/ca and the CFE-Hybrid-dynamic model computes ci/ca as a function of air temperature and VPD based on an eco-evolutionary optimality theory.

**Revisions:**

Line 943 – 951: We fixed  $\chi$  to the global long-term average value 0.7 typical to C3 plants (Prentice et al., 2014; Wang et al., 2017). We further tested a dynamic model that quantified  $\chi$  as a function of air temperature and vapor pressure deficit following an eco-evolutionary theory across global flux sites (Keenan et al., 2023). The estimated  $\chi$  had a mean and median of 0.7 and a standard deviation of 0.04 (Figure S11a). Differences in the direct CO2 effect between the dynamic and fixed  $\chi$  approaches were minimal, with an R2 of 0.99 and a slope of 0.99 from a least squares linear regression line (Figure S11b). GPP trends across flux towers were also highly consistent between the two approaches (Figure S11b, c). Since these results indicated that  $\chi$  is relatively stable, we used the fixed  $\chi$  approach to produce the CEDAR-GPP dataset.

Line 900 – 905: Appendix A:  $CO_2$  sensitivity function of Light Use Efficiency. In the CFE-Hybrid model, the direct  $CO_2$  fertilization effect was prescribed onto machine learning estimated GPP at a reference  $CO_2$  level using a theoretical  $CO_2$  sensitivity function of LUE. The sensitivity function, which describes the fractional change in LUE due to  $CO_2$  relative to the reference period, is described below.

Line 785 - 790: Robust modeling of LUE responses to rising  $CO_2$  under various environmental conditions remains challenging (Wang et al., 2017). Future work is needed to better understand how these factors affect the quantification of GPP and its long-term temporal variations.

[Previous comment]: L249 ' do not consider the direct effect of CO2 on light use efficiency', this is inaccurate, the VI sometimes can capture the LUE change from direct CFE.

**Response**: As clarified in our last response, VIs used in our analysis, such NDVI, EVI, and NIRv, primarily capture changes in vegetation structures (or LAI), thus the ability to intercept

light. These VIs are designed to highlight the strong NIR reflectivity caused by leaf internal mesophyll structures in vegetation. These VIs are not sensitive to direct CO2 impacts on LUE, which are associated with photochemical processes. Therefore, our baseline model, without considering CO2, does not capture the direct CFE.

[Previous comment]: Function A2 A = AC = Aj? I cannot agree with this! There are just one condition Ac = Aj is the photosynthesis transfer from lighted-limited to nutrient limited.

**Response**: This comment is associated with our CO2 sensitivity function of LUE as part of the CFE-Hybrid model. As we clarified previously, this function is based on the eco-evolution theory, specifically the optimal coordination hypothesis, which predicts that the electron-transport (Aj) and Rubisco-limited (Ac) rates of photosynthesis converge on the time scale of physiological acclimation. This coordination hypothesis is based on the idea that resources would be wasted if overcapacity of one process is maintained over the other. Many studies have found that coordination theory is consistent with leaf to canopy level measurements (e.g. Chen et al., 1993; Haxeltine & Prentice, 1996; Maire et al., 2012; Quebbeman & Ramirez, 2016; Wang et al., 2017). Harrison et al. (2021) provides a thorough review of the eco-evolutionary optimality theory for plant and vegetation processes. Given that the coordination of Aj and Ac happens in the order of a few weeks, the coordination theory can be readily applied to our monthly GPP estimates. Additionally, this theory allows us to model LUE sensitivity to  $CO_2$  based on Ai, which has a lower response to CO2 than Ac. Therefore, our CFE-Hybrid model provides a conservative estimate of the direct CFE. Lastly, note that we did not use Function A2 to directly model GPP: rather these are used to derive the direct CFE. Representations of GPP response to climatic and environmental factors, including the indirect CFE and water stress, are directly estimated by the machine learning models based on global eddy covariance data.

- Chen, J.-L., Reynolds, J. F., Harley, P. C., and Tenhunen, J. D.: Coordination theory of leaf nitrogen distribution in a canopy, Oecologia, 93, 63–69, https://doi.org/10.1007/BF00321192, 1993.
- Haxeltine, A. and Prentice, I. C.: A General Model for the Light-Use Efficiency of Primary Production, Functional Ecology, 10, 551–561, https://doi.org/10.2307/2390165, 1996.
- Maire, V., Martre, P., Kattge, J., Gastal, F., Esser, G., Fontaine, S., and Soussana, J.-F.: The Coordination of Leaf Photosynthesis Links C and N Fluxes in C3 Plant Species, PLOS ONE, 7, e38345, https://doi.org/10.1371/journal.pone.0038345, 2012.
- Quebbeman, J. A. and Ramirez, J. A.: Optimal allocation of leaf-level nitrogen: Implications for covariation of Vcmax and Jmax and photosynthetic downregulation, Journal of Geophysical Research: Biogeosciences, 121, 2464–2475, https://doi.org/10.1002/2016JG003473, 2016.
- Wang, H., Prentice, I. C., Keenan, T. F., Davis, T. W., Wright, I. J., Cornwell, W. K., Evans, B. J., and Peng, C.: Towards a universal model for carbon dioxide uptake by plants, Nature Plants, 3, 734–741, https://doi.org/10.1038/s41477-017-0006-8, 2017.
- Harrison, S. P., Cramer, W., Franklin, O., Prentice, I. C., Wang, H., Brännström, Å., de Boer, H., Dieckmann, U., Joshi, J., Keenan, T. F., Lavergne, A., Manzoni, S., Mengoli, G., Morfopoulos, C., Peñuelas, J., Pietsch, S., Rebel, K. T., Ryu, Y., Smith, N. G., Stocker, B. D., and Wright, I. J.: Eco-evolutionary optimality as a means to improve vegetation and land-surface models, New Phytologist, https://doi.org/10.1111/nph.17558, 2021.

[**Reviewer Comment**] Besides, since the most important improvement of CEDAR-GPP to other machine learning based GPP products is, they consider the CO2 in GPP modelling. However, the authors hidden the validation results at tropics, the most important contributor to the global GPP, I highly suspect they have a wrong validation. Therefore, I cannot accept the CEDAR-GPP at the current stage.

**Response**: We recognize the critical contribution of tropical regions to global ecosystem productivity and would like to provide further clarifications regarding our validation in these regions.

As detailed in Section 3.1.2, Figure 4, and Table S3, we have thoroughly validated monthly, seasonal, interannual, and cross-site GPP variations for tropical sites. Our models show reasonable accuracies in this area aligning with previous studies. However, the validation of GPP long-term trends in tropical areas is constrained by data availability. Across the FLUXNET2015, AmeriFlux OneFlux, and ICOS data that we compiled, only three tropical sites have more than three years of data, and none of them exceed six years (Figure R1 below). Given the strong interannual variabilities, robust detection of long-term trends is not feasible for these sites without longer data records. Figure R1 below shows the GPP time series at these sites from eddy covariance data and our Baseline, CFE-Hybrid, and CFE-ML models (short-term models). Note that none of the sites exhibit statistically significant trends. In the manuscript, we provided validation for individual sites with more than 10 years of data all tropical sites do not meet these criteria (Figure S4, attached below).

---

## Author Response (AR3)

**Response Letter**

**CEDAR-GPP: spatiotemporally upscaled estimates of gross primary productivity incorporating CO2 fertilization**

**Referee #1**

**Dear reviewer,**

We are grateful for your thorough and constructive feedback, which has helped us improve our manuscript. We have carefully addressed all concerns, including clarifying our methodology, providing additional context for the limitations of our dataset particularly in tropical regions in the revised manuscript. Below we provide our point-to-point response to reviewers' comments, with revisions highlighted in red. For revisions, we provide line numbers from the track-change revised manuscript.

[Reviewer Comment 1] The authors just added some results to make their results more convincing. However, I have pointed out the dataset itself is not robustness since the GPP trend at tropics is highly uncertain. Moreover, they didn't revise the dataset itself since the first time of submission. As a reviewer, my duty is to find out the potential shortcoming of this dataset and make this dataset reliable to the public. Till now, the dataset is not fully convincing. I am not doubting the innovation for this dataset, but the current version of dataset is not convincing.

**Response**: Thank you for your feedback. We understand your desire to ensure the dataset is as robust as possible, but respectfully disagree with the point that our dataset is "not convincing" due to high uncertainties in tropical GPP. Quantifying GPP trends, particularly in tropics, is a known challenge across the field. Current state-of-the-art datasets, such as FLUXCOM, do not capture these trends from site to global levels, partly because they overlook CO2 fertilization. By incorporating this effect, our models (CFE-ML and CFE-Hybrid) significantly improved trend estimates, compared to the Baseline (which represents state-of-the-art approaches). This improvement demonstrates the robustness and significance of our work and how it advances over currently available approaches.

We acknowledge that estimation of GPP trends in tropics remains highly uncertain, a limit shared by all upscaling studies due to limitations in eddy covariance data availability and remote sensing data quality in tropical regions. We have quantified these uncertainties (Fig. 12) to ensure transparency. Additionally, we have expanded our discussion in the revised manuscript to emphasize the limitations and challenges associated with validation. This aspect is highlighted in the abstract and conclusion (see text provided below).

We want to emphasize that the reviewer is correct, in that GPP trends in tropical regions are highly uncertain (as our results clearly show (Fig. 12)), but that this is an issue that affects our entire field, not just our paper, and the improvements we introduce greatly advance over previous efforts regardless.

**Revisions:** Line 857 – 865: Finally, direct validation of GPP trends is limited, particularly in tropical regions, constrained by the availability of long-term records. Detecting and evaluating trends is challenging and typically requires long monitoring records (e.g. over 10 to 15 years), since long-term changes, such as those induced by  $CO_2$ , are very small relative to large interannual variations. Evaluating aggregated GPP trends across multiple sites presents an alternative approach; however, there were still insufficient sites in tropical and evergreen broadleaf areas to robustly validate our estimates for those ecosystems (Figure 5). Partly due to data limitation, uncertainties in GPP estimated from bootstrapped samples are very high in tropical areas (Figure 12). Thus, trend estimates in these areas should be interpreted in the context of associated uncertainties and limitations.

Line 37 - 39: Estimating and validating GPP trends in data-scarce regions, such as the tropics, remains challenging, underscoring the importance of ongoing ground-based monitoring and advancements in modeling techniques.

Line 955 – 958: However, trend estimation and validation remain particularly challenging in data-scarce regions, such as the tropics, emphasizing the need for enhanced data availability and methodological advancements.

**[Reviewer Comment 2]** The CO2 fertilization effect indeed affects the global trend of GPP, but current dataset just explain 51% of the global trend, which is quite low.

**Response**: This comment appears to misinterpret our statement in the abstract: "Incorporation of the direct  $CO_2$  effects substantially enhanced the predicted long-term trend in GPP across global flux towers by up to 51%, aligning much closer to a strong positive trend from eddy covariance data"

Our finding indicates that the  $CO_2$  effects improved the model-predicted GPP trends at eddy covariance sites, when compared to models without considering  $CO_2$ . We have revised this sentence to improve clarity and avoid confusion.

**Revisions:** Line 28 – 32: After incorporating the direct  $CO_2$  effects, predicted long-term GPP trend across global flux towers substantially increased from 3.1 gCm-2year-1 to 4.5 – 5.4 gCm-2year-1, which aligns more closely with the 7.7 gCm-2year-1 trend detected from eddy covariance data.

**[Reviewer Comment 3]** More importantly, the GPP trend in CEDAR-GPP at tropics is not convincing (based on the limited results in the revised manuscript), so I cannot recommend this study for publication. I need to remind the authors should revise and answer the questions point to point, but not omit some key questions in the last round of revision. Therefore, a rejection but an invitation for resubmission is appropriate.

**Response**: Thank you for your feedback. We agree that estimating and validating GPP trends in tropics remain a major challenge in the field, due to data availability. However, we conducted thorough validation in regions with sufficient data, demonstrating the robustness of our dataset (Figure 5b,c). Additionally, we provided detailed uncertainty quantification, transparently outlining the limitations of our estimates (Figure 12). In the revised manuscript, we have expanded our discussion to emphasize these limitations.

**[Reviewer Comment 4]** 1 Validation 1.1 'Moreover, intercomparisons with other RSbased datasets, including FLUXCOM, FLUXSAT, and MODIS, confirm strong consistency inglobal patterns of annual mean GPP, interannual variability, and mean seasonal cycles'.

Comment: I think this part of result is convincing, no need to clarify it again.

**Response**: Thanks for the feedback.

[Reviewer Comment 5] 1.2' Our estimates of long-term trends agree with eddy covariance data across global sites. Notably, the Baseline model, which does not consider the direct CFE, underestimates GPP trends at flux towers. In contrast, the CFE-ML and CFE-Hybrid models show significant improvements, underscoring the need to consider both direct and indirect CFE. In our last revision, we performed additional validation of GPP trend by climate zones and plant functional types (Figure 5b, 5c). We also provided comparisons of estimated and observed trends at long-term sites (Figure S4).'

Comment: I think this validation is weak, since in Figure 5b,c, the author still not show the results at tropics.

**Response**: Thanks for the feedback! Tropics were not included because there were insufficient sites to support a reliable analysis, as noted in the manuscript (line 354 – 355). Specifically, at least five sites with five years of continuous observations (with no gaps more than one month per year) were required. For upscaling purposes, we applied strict quality control, discarding records with more than 20% missing or low-quality gap-filled data – a standard procedure in upscaling studies - which further limited data availability. In the revised manuscript, we expanded our discussion to emphasize the limitations of

our dataset in tropical and data scarce conditions, and we noted that the estimated uncertainty could be used as a reference of data quality. Line 354 – 355: Categories with less than six long-term sites available were excluded

from the analysis, which includes

**355 EBF and Tropics.**

In figure S4, what I can see is, half of the CFE-hybrid and CFE-ML cannot capture the trend in EC sites. So this validation is not convincing.

**Response**: We respectfully disagree with the statement that our "validation is not convincing". CEDAR-GPP greatly improved the quantification of trends when compared to existing state-of-the-art approaches. We established the Baseline model as a reference to the current methods ensuring a fair comparison. In Figure S4 (now S7), the CFE-Hybrid and CFE-ML models showed significantly better performance than the Baseline (current state-of-the-art) in most sites, demonstrating the effectiveness of our methods.

Note that directly benchmarking existing datasets (such as FLUXCOM) against eddy covariance data is not an apples-to-apples comparison, as the datasets were not developed using the same sites. To ensure fairness, we established the Baseline model to represent current methods used by datasets like FLUXCOM and FluxSat.

We agree that substantial differences exist between estimated (even with CO2 effects) and EC-based GPP trends in Figure S4. It is important to consider factors other than CO2 that can cause long-term GPP changes, such as nitrogen deposition, forest aging, succession, changes in surface roughness, or natural and manmade disturbances. These factors may be underrepresented in our models, contributing to the underestimation of trends. Robustly reconstructing trends in individual sites across the globe remains challenging, given the current limitations in eddy covariance and remote sensing inputs. Despite the challenges, our incorporation of the CO2 effects marks a significant improvement over current approaches and a meaningful advancement to the field.

In the revised manuscript, we have expanded our discussion on the limitations of our datasets in representing  $CO_2$  fertilization and trends and highlighted this aspect in the abstract and conclusion. We have introduced the following changes to highlight these points:

**Revisions:** Line 830 – 850: Several limitations should be noted regarding GPP trend estimation and validation. First, the CFE-ML model may not fully capture the intricate mechanisms of plant physiological responses to CO2. For example, eddy covariance towers, especially long-term sites, are typically located in homogeneous and undisturbed ecosystems, not representative of the full diversity of ecosystems globally. Thus, interactions between CO2 and natural or human-induced disturbance, as well as many other stresses, are likely underrepresented in the models. Ultimately, the model's capacity to robustly quantify CO2 fertilization is constrained by the scope and diversity of the eddy covariance data. Additionally, the use of spatially invariant CO2 data may not fully represent the actual CO2 variations that plants experience across different environments.

Secondly,  $CO_2$  effects inferred by the CFE-ML models may be confounded by other factors that correlate with  $CO_2$  over time. Industrialization-induced nitrogen

deposition could synergistically boost GPP alongside CO2 (O'Sullivan et al., 2019). Technological and management improvements in agriculture that contribute to a global enhancement of crop photosynthesis (Zeng et al., 2014), might also be indirectly reflected in the model estimates. Moreover, interactions with the other input features that exhibit long-term trends, such as those induced by non-biological factors (e.g. sensor orbital drifts), also affect the CO2 effects inference. Additionally, other factors that could lead to long-term GPP trends (e.g. forest aging, disturbances) might also be underrepresented in our models.

Line 37 - 39: However, estimating and validating GPP trends in data-scarce regions, such as the tropics, remains challenging, underscoring the importance of ongoing ground-based monitoring and advancements in modeling techniques.

Line 955 – 958: However, trend estimation and validation remain particularly challenging in data-scarce regions, such as the tropics, emphasizing the need for enhanced data availability and methodological advancements.

**[Reviewer Comment 6]** 'we show that CEDARGPP exhibits higher consistency with TRENDY models after incorporating the direct CFE'

Comment: The TRENDY models cannot be the validation! The TRENDY models are results from simulations but not the ground measurement. So this could just be cross-validation at the model world but the observations at the real world.

**Response**: We fully agree with the reviewer that the TRENDY model comparison should not be considered validation. The TRENDY intercomparison was used to illustrate that previous inconsistencies between satellite-based GPP and TRENDY may be induced by an omission of the direct CO2 effect in satellite estimates.

**[Reviewer Comment 7]** I think the authors didn't pay enough attention to my comments. Table S1 should provide the validated and training year for each flux site as I mentioned in the first time.

**Response**: We appreciate the feedback but noted a misinterpretation of our approach. We split data by sites rather than years. In our five-fold cross-validation, each site (including all data years) was randomly assigned to one of five groups, with each iteration training on four groups and testing on one. This scheme assesses model performance on unseen sites, which is more applicable for upscaling. Splitting sites by years would risk overfitting and deflating error metrics.

In the revised manuscript, we have added each site's time span and IGBP class to Table S1 and relocated it to the Appendix, so individual site citations can be integrated to the main text.

**[Reviewer Comment 8]** To my knowledge, there are at least 7 EC sites with long term GPP observation are at tropics:

(1) AU-Rob
https://ozflux.org.au/monitoringsites/robsoncreek/index.html
(2) GF-Guy
https://fluxnet.org/sites/siteinfo/GF-Guy
(3) BR-Sa1
https://ameriflux.lbl.gov/sites/siteinfo/BR-Sa1
(4) AU-ASM
https://ozflux.org.au/monitoringsites/alicesprings/index.html
(5) AU-LIF
https://ozflux.org.au/monitoringsites/litchfield/index.html
(6) CN-Xishuangbanna
https://www.scidb.cn/en/detail?dataSetId=db9bf2dde00746f7a40cfc3dbad324b2
(7) CG-YPS
https://www.congo-biogeochem.com/congoflux
All of these sites contain more than 8 years of observation.

**Response**: Thank you for sharing the list of EC sites with long-term observations. While increased data availability could benefit model robustness, adding a few sites is likely still insufficient for comprehensive inference and validation. As we noted earlier, evaluating  $CO_2$  effects and benchmarking trends in a single site can be problematic, due to the impacts of other factors that may not be fully represented in the models.

We have reviewed these data sources and found that they have limited years of publicly available data insufficient for robust evaluation (e.g. AU-Rob has only 2014 available in FLUXNET2015; CN-Xishuangbana, only has data from 2011 to 2015). Some sites were included in our analysis but lacked long records due to quality issues (e.g. GF-Guy and BR-Sa1).

Given that adding a few sites would not significantly alter our results, we did not include them in this manuscript. But we are committed to continue improving our methods and datasets as more data becomes available, such as the upcoming FLUXNET dataset.

**[Reviewer Comment 9] 2 Model**

As the second reviewer also concerned using a constant  $\chi$  value for the generation of long-term GPP. So if my understanding is right, the constant  $\chi$  value can only reproduce the spatial difference of different PFT, but it cannot reproduce the trend of VPD and soil moisture control to  $\chi$ .

Following the assumption of FvCB model, a constant  $\chi$  value indicated that the GPP is driven by atmospheric CO2 concertation and air temperature but not related to the water

stress such as VPD and soil moisture control. So this also lead to the CEDAR-GPP is leak of robustness.

**Response**: There appears to be a misinterpretation of our methods here. All our models account for GPP responses to VPD, soil moisture, and other environmental factors. The optimality theory that the reviewer refers to, which assumes a constant  $\chi$  value, only simulates the response of LUE/GPP to CO2 concentration. The machine learning models were designed to capture the effects of all other factors on  $\chi$ /LUE/GPP. The CFE-Hybrid model combines the two components (machine learning and optimality theory) to fully represent GPP dynamics.

Our choice of a constant  $\chi$  value is based on rigorous sensitivity analysis at eddy covariance towers, where we found that the CO2 fertilization effects were highly consistent between the constant and dynamic  $\chi$  approaches with  $r^2 > 0.99$  (Figure S14). Given that the dynamic approach did not result in meaningful differences, and that quantification of VPD effects on stomatal conductance remains an active research area with high uncertainties, we decide to use the constant method for the main results. We therefore respectfully disagree with the reviewer's statements here, and hope these clarifications highlight why they are a misinterpretation of our methods and results.

**[Reviewer Comment 10]** Once again, it is not doubt that adding the direct CO2 fertilization effect to machine learning based GPP modelling is important. But I think the high uncertainties in the trend of GPP especially at tropics make this dataset untrusted. The seasonal trend, IAV, and mean spatial distribution of annual GPP are good, but the GPP trend is still not convincing and it is the most important outcome for direct CFE.

Providing a robust dataset for the science community is important, since sometime the wrong results caused the misleading to climate change feedback evaluation.

**Response**: We thank the reviewer for highlighting the importance of incorporating direct CO2 effect and agree on the need for robust datasets. However, we respectfully disagree on the statement that "the high uncertainties in the trend of GPP especially at tropics make this dataset untrusted."

We have demonstrated significant improvements in trend estimates in eddy covariance sites against existing state-of-the-art datasets, particularly in arid, temperate, and cold regions. We acknowledge that estimation and validation of trends in tropics remain highly uncertain due to data limitation. To ensure transparency, we provided detailed uncertainty quantification and emphasized in the manuscript that our datasets should be used and interpreted in the context of uncertainties and limitations.

We concur that robust estimation and uncertainty quantification are important for accurate assessment of ecosystem changes and climate feedback. To further emphasize this, we have added a clarifying note in the revised manuscript.

Revisions: Line 929 – 936: Finally, like other upscaled or remote sensing-based GPP datasets, CEDAR-GPP should not be regarded as "observations" but rather as model estimates informed by remote sensing and ground-based data. The extent of assumptions or structural constraints varies across such datasets. CEDAR-GPP, particularly in its CFE-Baseline and CFE-ML configurations, is entirely data-driven and incorporates no explicit assumptions regarding the biological and environmental processes underlying photosynthesis, apart from the generic assumptions inherent in machine learning models. Consequently, the usage and interpretation of this dataset should be carefully framed within the context of the input eddy covariance and environmental data as well as their limitations.

**Referee #4**

**Dear Dr. Nelson,**

Thank you for your thorough evaluation and thoughtful suggestions for our manuscript. We have carefully revised the manuscript in response to your feedback, and these revisions have significantly enhanced its quality. Below is a summary of the major changes:

1. We expanded the discussion on limitations related to the quantification of CO2 fertilization.

2. We ensured that all ten CEDAR-GPP model results were included in the cross-validation and intercomparison analyses.

3. We revised the cross-validation sampling protocol to account for co-located sites and updated the associated figures, tables, and text.

We also observed that your review was likely based on an earlier version of our manuscript (the preprint available in ESSD discussion). Our most recently submitted version differed from this earlier version in that we had implemented revisions in response to previous reviewers' comments, which addressed some of your points. This is noted in our point-by-point response below.

We hope our responses satisfactorily address your concerns and suggestions. We are looking forward to any further feedback or questions that you may have.

Below we provide our point-to-point response, highlighting revisions in red and line numbers from the track-change revised manuscript.

[Reviewer Comment 1] The manuscript "CEDAR-GPP: spatiotemporally up-scaled estimates of gross primary productivity incorporating CO2 fertilization" outlines an up-scaling approached based on eddy covariance estimates, which differentiates itself from other similar exercises by incorporating CO2 effects on global GPP. The methodology is benchmarked both in cross-validation and comparison to existing products, and shows similar performances. Overall, the evaluation is well done as a benchmark to introduce the new dataset, with a good framing of the motivation and discussion on the general limitations to such up-scaling exercises. The inclusion of an ensemble evaluation with some metrics of uncertainty is also well outlined and a welcome addition.

**Response**: Thank you for your positive feedback on our approach and evaluation of the CEDAR-GPP product.

**[Reviewer Comment 2]** One key aspect that I think needs to be improved is, given that the key addition compared to existing products relates to the CO2 fertilization effects, the manuscript needs further discussion of the potential limitations and implications of how these effects are introduced to the method and how users should interpret them. Some aspects, such as the fact that what is interpreted here as impacts of CO2 fertilization could include any factor with temporal trends, are mentioned only briefly in the discussion. Many other factors, including non-biological

factors such as developments in eddy covariance techniques, the bias in the fact that long term eddy covariance towers are generally placed in locations that are relatively undisturbed and protected, and potential trends in the feature sets such as due to sensor drifts, could be interpreted as CO2 fertilization effects. I do not think these represent a fundamental issue with the CEDAR-GPP products, as I would say they represent a very interesting hypothesis about how CO2 effects can be incorporated into a data driven product, and the set-up described is well evaluated. However, as data driven products tend to be seen as an "observation", it is especially important to highlight these issues to advise on the potential limitations to their use and interpretation.

**Response**: Thank you for this insightful suggestion. In the revised manuscript, we have expanded the discussion on the limitations of ML-based  $CO_2$  fertilization quantification, addressing potential biases stemming from eddy covariance tower representation and the influence of non-biological trends. We have also emphasized the challenges in isolating  $CO_2$  fertilization effects, noting the constraints due to the limited availability of long-term observations, as well as the confounding interactions with other environmental factors.

In addition, we have previously introduced a new section offering detailed guidance on data usage, specifically discussing considerations related to CO2 fertilization, during previous reviews. In this revision, we have further emphasized the "modeled" nature of CEDAR-GPP products, providing additional cautionary notes regarding their interpretation and application.

Revisions: Line 830 - 872: Several limitations should be noted regarding GPP trend estimation and validation. First, the CFE-ML model may not fully capture the intricate mechanisms of plant physiological responses to CO2. For example, eddy covariance towers, especially long-term sites, are typically located in homogeneous and undisturbed ecosystems, not representative of the full diversity of ecosystems globally. Thus, interactions between CO2 and natural or humaninduced disturbance, as well as many other stresses, are likely underrepresented in the models. Ultimately, the model's capacity to robustly quantify CO2 fertilization is constrained by the scope and diversity of the eddy covariance data. Additionally, the use of spatially invariant CO2 data may not fully represent the actual CO2 variations that plants experience across different environments.

Secondly, CO2 effects inferred by the CFE-ML models may be confounded by other factors that correlate with CO2 over time. Industrialization-induced nitrogen deposition could synergistically boost GPP alongside CO2 (O'Sullivan et al., 2019). Technological and management improvements in agriculture that contribute to a global enhancement of crop photosynthesis (Zeng et al., 2014), might also be indirectly reflected in the model estimates. Moreover, interactions with the other input features that exhibit long-term trends, such as those induced by non-biological factors (e.g. sensor orbital drifts), also affect the CO2 effects inference. Additionally, other factors that could lead to long-term GPP trends (e.g. forest aging, disturbances) might also be underrepresented in our models.

Finally, direct validation of GPP trends is limited, particularly in tropical regions, constrained by the availability of long-term records. Detecting and evaluating trends is challenging and typically requires long monitoring records (e.g. over 10 to 15 years), since long-term changes, such as those induced by CO2, are very small relative to large interannual variations. Evaluating aggregated GPP trends across multiple sites presents an alternative approach; however, there were still insufficient sites in tropical and evergreen broadleaf areas to

robustly validate our estimates for those ecosystems (Figure 5). Partly due to data limitations, uncertainties in GPP estimated from bootstrapped samples are very high in tropical areas (Figure 12). Thus, trend estimates in these areas should be interpreted in the context of associated uncertainties and limitations.

Line 886 – 894: Lastly, quantifications of GPP trends and their causes remain highly uncertain from site to global scales. Trend detection is often complicated by data noises and interannual variabilities, thus requiring long-term records which are limited in certain areas, biomes, and environmental conditions, such as tropics, polar regions, wetlands, as well as ecosystems with regular or anthropogenic disturbances (Baldocchi et al., 2018; Zhan et al., 2022). Moreover, isolating the effect of CO2 is challenging, as it is confounded by other factors, such as forest regrowth, land cover change, and disturbances, which also significantly impacts long-term GPP variations. To this end, continued efforts in expanding ecosystem flux measurements and standardizing data processing present new opportunities to assess ecosystem productivity responses to changing climate conditions (Delwiche et al., 2024; Pastorello et al., 2020). Future research could also leverage novel machine learning techniques, such as knowledge-guided machine learning (Liu et al., 2024) and hybrid modeling that combines process-based and machine learning approaches (Kraft et al., 2022; Reichstein et al., 2019).

Line 915 - 924: CO2 Fertilization Effect (CFE) configurations: the CFE-Hybrid and CFE-ML setups are preferable when assessing temporal GPP dynamics, especially long-term trends. The CFE-Hybrid setup includes a hypothetical trend from the direct CO2 effect, while CFE-ML is purely data-driven and does not make any specific assumption about the sensitivity of photosynthesis to CO2. Averaging the CFE-Hybrid and CFE-ML estimates is acceptable, with the difference between them reflecting the uncertainty surrounding the direct CO2 effect. Note that the Baseline setup should not be used to study long-term GPP dynamics, especially those induced by elevated CO2. The Baseline setup may be useful to compare with other remote sensing-derived GPP datasets that do not consider the direct CO2 effect. Differences between these setups regarding mean GPP spatial patterns, seasonal and interannual variations are considered to be minor.

Line 929 – 936: Finally, like other upscaled or remote sensing-based GPP datasets, CEDAR-GPP should not be regarded as "observations" but rather as model estimates informed by remote sensing and ground-based data. The extent of assumptions or structural constraints varies across such datasets. CEDAR-GPP, particularly in its CFE-Baseline and CFE-ML configurations, is entirely data-driven and incorporates no explicit assumptions regarding the biological and environmental processes underlying photosynthesis, apart from the generic assumptions inherent in machine learning models. Consequently, the usage and interpretation of this dataset should be carefully framed within the context of the input eddy covariance and environmental data as well as their limitations.

**[Reviewer Comment 3]** Furthermore, in the comparisons between CEDAR GPP and the reference datasets, it would be important to always include the three variants (baseline and ML), at least in the supplement. Each up-scaling product has many differences due to small choices (e.g. QC, version of eddy covariance data, feature set and processing, etc...), so referencing how the three variants of CEDAR GPP differ, where the underlying methodology is most similar, will help differentiate the effects of CO2 and other methodological choices.

**Response**: We completely agree! During previous rounds of reviews, we have included all the CEDAR-GPP model variants in the cross-validation and intercomparison analyses, supporting a more thorough interpretation of the results. In the main text, we noted the major differences and general consistencies between the models. Please refer to our responses to your specific comments for detailed changes.

**Revisions:** Figures S1, S4, S5, S6, S8, S9, S11, S12

Besides the main point of discussing the limitations to interpreting the CO2 fertilization effects, there are a number of clarifications to the methodology that are described in the specific points below that should be addressed before publication.

**[Reviewer Comment 4]** L120 - I guess here high-quality would be based on the ONEFLUX flags as measured or high quality gap filling? Best to be explicit.

Response: Thank you for pointing this out. We have clarified in the revised manuscript

**Revisions:** Line 133 – 135: High-quality data refers to GPP derived from measured or highquality gap-filled Net Ecosystem Exchange (NEE) data.

**[Reviewer Comment 5]** L123 - The classification of C3/C4 at all eddy covariance sites can be a difficult task, especially at crop sites with rotations. It could be useful to add this information somewhere, either as a supplement or as a referenced dataset.

**Response**: Thank you for the suggestion. We have added the data in the supplement.

Revisions: Line 139: This classification information is included in Supplementary Text S1.

**[Reviewer Comment 6]** L125 - A list of all sites used should be included, ideally with the corresponding DOI where available.

**Response**: Thanks for the suggestion. We included this table in the supplement during our first revision. In this round of revision, we have moved the table to Appendix A to ensure the individual site DOIs are cited in the main text references.

Revisions: Appendix A.

**[Reviewer Comment 7]** Table 1 - Having the information on which datasets were used in which model would be useful to show here (e.g. combining with Table S1), and would make a nice overall summary of the set-up, particularly as there is a lot of empty space in this table.

**Response**: Thanks for the suggestion. We have revised Table 1 to incorporate specific usage of each dataset in the model setups. Table S1 was kept to list original and extracted variables from each dataset.

Revisions: Table 1.

**[Reviewer Comment 8]** L213 - Here it says all data was aggregated to 0.05° resolution, but many datasets are at courser resolutions. How was the resampling done? linear interpolation? Also, I did not see a description of how the gridded data was matched to the eddy covariance towers, I guess the nearest 0.05° pixel was used?

**Response**: Following your suggestion, we have included a description of the resampling and the nearest pixel approach in the revised manuscript.

**Revisions:** Line 243 – 247: Finally, all the datasets were aggregated to a monthly time step and 0.05-degree spatial resolution. We employed the conservative resampling approach using the xESMF python package (Zhuang et al., 2023). To generate the machine learning model training data, we extracted values from the nearest 0.05 degree pixel relative to the site locations within the gridded dataset.

**[Reviewer Comment 8]** L270 - Was there a significant difference between the baseline and the reference models?

**Response**: Great question! The Baseline and CFE-REF models show no difference in long-term trends and their spatial patterns (Figure R1), as indicated by a close to 1 slope and near-zero bias term in the regression line (Figure R3a, c). This consistency suggests that the CO2 effect has been effectively removed from the CFE-ML models.

The key difference between the Baseline and CFE-REF models lies in the magnitude of predicted GPP (Figure R2, R3). Global annual GPP from the CFE-REF models is systematically lower by 1.2 - 1.4% compared to the Baseline models (Figure R2, R3). Despite the bias, GPP predictions from both models agree well with an R2 over 99% and a regression slope of 1. The difference in GPP magnitudes stems from the models' assumptions. The Baseline model, without accounting for CO2 changes over time, essentially assumes a fixed CO2 level, corresponding to an "average" level from 2001 to 2020. However, the CFE-REF models remove CO2 effects by fixing it to the minimum level of year 2001. Therefore, the CFE-REF predicts lower GPP values, though GPP temporal dynamics are consistent between the models.

**Revisions:** Line 308 – 309: Long-term trends from the reference and the Baseline models are consistent.

---

## Author Response (AR4)

**Response Letter**

CEDAR-GPP: spatiotemporally upscaled estimates of gross primary productivity incorporating CO2 fertilization

**Referee #1**

Dear reviewer,

We are grateful for your thorough and constructive feedback, which has helped us improve our manuscript. We have carefully considered each comment and made the following key changes to address your suggestions:

- 1) We conducted additional site-level evaluation using 11 independent sites in tropical and arid areas, which were previously not involved in model development.
- 2) We expanded the intercomparison analysis for global and regional GPP trends using three process-based models forced with remote sensing data.
- 3) We developed further analysis of the direct CO2 effect on GPP trends
- 4) We added new illustrations and discussion on uncertainties associated with the estimated GPP trends.

Below we provide our point-to-point response to reviewers' comments, with revisions highlighted in **red**. For revisions, we provide **line numbers** from the track-change revised manuscript.

As I pointed out, the trend of GPP is the most important result for model effectiveness, so validation is very important. Besides, I have provided several sites (comment #8) for model data validation in the last round of revision, however, the new version of the manuscript has nothing new about it. Because the site validation is the most effective way for the direct validation for GPP trend, and the ESSD also highlight the robustness for dataset especially in the model validation. So the validation has to be done.

So I think four possible ways to prove the results are robustness especially the trend of GPP

[Reviewer Comment 1] 1、Site validation in the EBF at tropics.

The authors should use at least 6 or more sites to validate other results at EBF at tropics, because the sites in the EBF tropics contributes a very high amount of global GPP and now the sites in this study cannot reproduce the spatial representativeness. So please follow the [comment#8] from the last round of revision and validate it.

**Response**: Thank you for your suggestion of enhancing site validation in EBF and tropics. In response, we expanded our validation to include the new OzFlux FluxNet dataset

(https://data.ozflux.org.au/portal/pub/viewColDetails.jspx?collection.id=1882723&collecti

on.owner.id=450&viewType=anonymous), which contains 11 Australian sites, previously not involved in our model training. This dataset contains three tropical sites (savannas) and two non-tropical EBF sites (*Table S3*). Notably, it includes two sites you previously suggested—AU-Lit (Litchfield) and AU-ASM (Alice Springs). Regarding the other five suggested sites, we note that long-term high-quality data necessary for trend evaluation were not available. A detailed explanation of each site's data limitation are provided at the end of the response.

In the validation based on the OzFlux FluxNet dataset, we evaluated annual trends in seven sites with at least four years of good-quality data (*Figure S9*). Of these, only two sites - AU-Cpr (Tropical) and AU-Stp (Arid) exhibited a trend with p-value < 0.3. The CEDAR-GPP model estimates align closely with the observed trend. The other sites showed strong interannual variabilities, which CEDAR-GPP captured effectively, particularly in AU-ASM (savanna) and AU-Cum (EBF) (*Figure S9*, *Figure S10*, *Table S4*).

Within OzFlux and our existing data from FLUXNEXT2015, ICOS, and AmeriFlux datasets, we have a total of 5 tropical and 5 EBF sites (*Figure R1*). However, high-quality data spanning over five years are available only in 1 tropic and 3 EBF sites. This still limits our ability to assess aggregated trends for tropical areas and the effect of  $CO_2$  fertilization.

Lastly, we carefully assessed the sites suggested by the reviewer. AU-Rob, GF-Guy, and BR-Sa1 were part of FLUXNEXT2015, thus already included in our analysis. Unfortunately, AU-Rob had only one year of high-quality data in FLUXNET2015. BR-Sa1 had frequent missing values for at least three months per year from 2002 to 2011. GF-Guy was located in a coastal area and did not contain GPP estimates from the night-time partitioning method. Moreover, while both the Xishuangbanna and CongoFlux sites are valuable for enhancing representation in Africa and Asia, Xishuangbanna only has five years of data with large interannual variation, and CongoFlux data collection started after 2020, thus lacking sufficient long-term observations for trend evaluation.

Nevertheless, the most robust assessment of GPP trends and CO2 fertilization between observations and models remains the comparison of aggregated trends, for which we have included the maximum amount of data that met analysis criteria and results indicate that models which incorporate direct CO2 effects improve the match with observed GPP trends compared to models that do not include direct CO2 effects

| Site ID       | IGBP | Koppen zone | Data range | No. of site-months |  |
|---------------|------|-------------|------------|--------------------|--|
| AU-ASM        | SAV  | Arid        | 2010-2019  | 111                |  |
| AU-Adr        | SAV  | Tropical    | 2007-2009  | 19                 |  |
| AU-Boy        | SAV  | Temperate   | 2017-2019  | 24                 |  |
| AU-Cpr        | SAV  | Arid        | 2011-2019  | 104                |  |
| AU-Cum        | EBF  | Temperate   | 2014-2019  | 71                 |  |
| AU-Dry        | WSA  | Tropical    | 2010-2019  | 90                 |  |
| AU-GWW | SAV  | Arid        | 2013-2019  | 83                 |  |
| AU-Lit        | SAV  | Tropical    | 2015-2019  | 53                 |  |
| AU-Rgf        | CRO  | Temperate   | 2016-2019  | 39                 |  |
| AU-Stp        | GRA  | Arid        | 2009-2019  | 114                |  |
| AU-War        | EBF  | Temperate   | 2013-2019  | 53                 |  |

Table S3. Sites from the OzFlux FluxNet dataset used for independent validation.

**Figure S9**. Standardized annual GPP anomalies from eddy covariance data and estimated by CEDAR-GPP for seven independent sites fomr the OzFlux FluxNet dataset. The results compare three CEDAR-GPP model setups – ST\_Baseline\_NT, ST\_CFE-Hybrid\_NT, and ST\_CFE-ML\_NT. Eddy covariance GPP was partitioned using the Night-time (NT) approach. The bottom right inset table lists the annual GPP trends based on Sen's slopes and the Mann-Kendall test.

|                | R 2 |      |          |        | RMSE  |      |          |        |  |
|----------------|----------------|------|----------|--------|-------|------|----------|--------|--|
| Model Setup    | Overa          | MS   | Anomalie | Cross- | Overa | MS   | Anomalie | Cross- |  |
|                | 11             | С    | S        | site   | 11    | С    | S        | site   |  |
| ST_Baseline_NT | 0.75           | 0.77 | 0.33     | 0.77   | 1.27  | 1.23 | 0.77     | 1.00   |  |
| ST_CFE-        | 0.75           | 0.77 | 0.22     | 0.77   | 1 07  | 1 22 | 0.77     | 0.00   |  |
| Hybrid_NT      | 0.75           | 0.77 | 0.55     | 0.77   | 1.27  | 1.23 | 0.77     | 0.99   |  |
| ST_CFE-ML_NT   | 0.75           | 0.77 | 0.33     | 0.76   | 1.27  | 1.24 | 0.77     | 1.01   |  |
| LT_Baseline_NT | 0.74           | 0.80 | 0.26     | 0.77   | 1.29  | 1.15 | 0.81     | 0.98   |  |
| LT_CFE-        | 0.74           | 0.70 | 0.26     | 0.76   | 1 28  | 1 16 | 0.81     | 1.00   |  |
| Hybrid_NT      | 0.74           | 0.79 | 0.20     | 0.70   | 1.20  | 1.10 | 0.01     | 1.00   |  |
| ST_Baseline_DT | 0.73           | 0.74 | 0.50     | 0.69   | 1.40  | 1.40 | 0.67     | 1.17   |  |
| ST_CFE-        | 0.73           | 0.74 | 0.50     | 0.60   | 1 30  | 1 40 | 0.67     | 1 16   |  |
| Hybrid_DT      | 0.75           | 0.74 | 0.50     | 0.09   | 1.59  | 1.40 | 0.07     | 1.10   |  |
| ST_CFE-ML_DT   | 0.74           | 0.74 | 0.50     | 0.69   | 1.38  | 1.39 | 0.67     | 1.16   |  |
| LT_Baseline_DT | 0.74           | 0.78 | 0.43     | 0.72   | 1.37  | 1.29 | 0.71     | 1.11   |  |
| LT_CFE-        | 0.74           | 0.77 | 0.43     | 0.71   | 1 38  | 1 30 | 0 71     | 1 13   |  |
| Hybrid DT      | 0.74           | 0.11 | 0.40     | 0.71   | 1.50  | 1.50 | 0.71     | 1.15   |  |

**Table S4**. CEDAR-GPP model performance based on independent data from the OzFlux FluxNet dataset

---

## Author Response (AR5)

**Response Letter**

**CEDAR-GPP: spatiotemporally upscaled estimates of gross primary productivity incorporating CO$_2$ fertilization**

Dear editor,

Thank you very much for the opportunity to revise our manuscript further. We have incorporated changes to the figures according to Referee #1's excellent suggestions. We're grateful for all four reviewers' feedback throughout the review process, which has significantly enhanced the manuscript's quality.

Sincerely,
Yanghui Kang
On behalf of all co-authors

**Referee #1**

Thanks to the authors for their great efforts.

Dear reviewer,

Thank you very much again for your thorough review and thoughtful feedback. We have revised the figures following your suggestions. Please find our point-to-point responses below.

I just have two minor suggestions.
1 Please combine Fig.S17 into the Fig. 10(a) in the main text. Also, you could put the Fig. S18(a) into the Fig 11(a) in the main text. Thus, we could have a clear overview of how the annual GPP varying among different GPP products.

Response: Thank you! Following your suggestions, we have combined the former Fig. S17 with Fig. 10. Given the increased number of maps, we have split the updated figure into two: Fig. 10 now presents the trend evaluation for 2001 – 2018), while Fig. 11 presents the trend evaluation for 1982 – 2018.

We have also incorporated the contents from the former Fig. S18 into the former Fig 11. However, for Fig 11a, we found that adding more datasets (i.e. time series lines) significantly reduced readability due to overlapping lines. Therefore, we retained Fig. S18 and instead modified Fig. 11 (old) for better clarity. Specifically, we have split it to two separate figures: Fig. 12 and Fig. 13, showing the annual time series and overall trends comparison respectively. BESS was included in the long-term time series in Fig. 12b. Fig. 12a still shows the original datasets, but references Fig. S18 (now Fig. S16) for comparison with BESS, BEPS, and PML. In Fig. 13, global trends from BESS, BEPS, and PML were shown along with all other datasets.

2 I think the new Fig.S20 is very clear according to my comment#4 at the last round of revision. I highly recommend the authors use Fig. S20 instead of Fig. 12 in the main text. Maybe you could just show the NT or DT series in Fig.S20 for a clear presentation.

Response: Thank you for the suggestion. We have moved the former Fig. S20 to the main text (now Fig. 15), retaining Fig. 12 (now Fig. 14) as it shows additional context on uncertainty distribution across climate zones. Fig. 15 includes the NT datasets only, and DT results were now provided in Fig. S19.